# Beyond Structural Symmetries:
# Linear Mode Connectivity via Neuron Identifiability

Vincent Bürgin [* 1]   Daniel Herbst [* 1]   Ya-Wei Eileen Lin [1]   Stefanie Jegelka [1 2]

## Abstract

Many striking phenomena in deep learning, such as linear mode connectivity and the structured behavior of training dynamics, are closely tied to parameter symmetries: transformations that leave the realized function unchanged. Despite growing attention to parameter symmetries, the exact interplay between parameters, data, and representations remains underexplored. To investigate this, we develop a theoretical framework of *effective function classes*, i.e., the set of functions a neuron can realize on its input support, and the norm cost of realizing them. We then formalize *effective symmetry breaking* via neuron identifiability across independent training runs. Our analysis shows that neural networks can admit large families of approximately equivalent solutions even in *structurally asymmetric* models. We further show that neuron identifiability enables representation merging *without prior alignment*, and characterize when such merging admits a linear low-loss path. These findings highlight the role of effective function classes in affecting the loss landscape.

## 1. Introduction

Modern neural networks are typically overparameterized and have highly nonconvex loss landscapes, yet gradient-based training often converges to models with similar predictions and generalization performance (Goodfellow et al., 2015; Nguyen et al., 2019; Zhang et al., 2017; Li et al., 2018). This occurs in part because many parameters in weight space implement the same function, yielding large equivalence classes of models that are functionally iden-

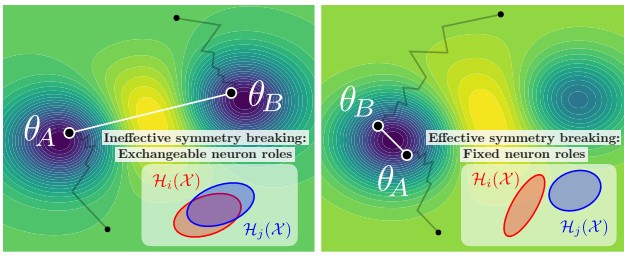

*Figure 1.* Illustration of neuron identifiability. *(Left)* Structural parameter symmetry broken but functions remain indistinguishable on data $\mathcal{X}$. *(Right)* Neurons identifiable, effective symmetry breaking enables merging representations without alignment.

tical despite differing in their parameters. Formally, common architectures admit large parameter symmetry groups (Hecht-Nielsen, 1990; Zhao et al., 2026), shaping optimization geometry (Zhao et al., 2023) and neuron-level interpretability (Godfrey et al., 2022). Parameter symmetries also govern how we analyze or merge weights, and thus act as the data symmetries in the growing area of weight space learning (Andrychowicz et al., 2016; Eilertsen et al., 2020; Unterthiner et al., 2020; Schürholt et al., 2022).

Among parameter symmetries, permutation symmetries of hidden units (Simsek et al., 2021; Zhao et al., 2022; 2024) are among the most ubiquitous and consequential. Widely used weight initializations are invariant and gradient-based training is equivariant under such permutations. Consequently, independent training runs naturally explore different orbit representatives. Most importantly, permutation symmetries have been connected to Linear Mode Connectivity (LMC), that is, the property that independently trained neural network solutions can be linearly interpolated in weight space while retaining similar performance (Frankle et al., 2020). LMC has been conjectured and empirically shown to hold in many settings *when permutations are accounted for* (Entezari et al., 2022), which can be achieved via post-hoc alignment of hidden units (and, analogously, channels in convolutional neural networks or heads in transformers) (Singh & Jaggi, 2020; Ainsworth et al., 2023).

Besides *structural* symmetries, neural networks can also exhibit *approximate*, *data-dependent*, and *local* symmetries: transformations that need not preserve the realized function for all inputs, but approximately preserve it on the data or

*Equal contribution [1] Technical University of Munich; School of CIT, MCML, MDSI [2] MIT; Dept. of EECS, CSAIL. Correspondence to: Vincent Bürgin <vincent.buergin@tum.de>, Daniel Herbst <daniel.herbst@tum.de>.
Code: github.com/vuenc/neuron-identifiability.

representations acting as input to a given layer. This setting is natural as both data and learned representations are often close to low-rank (Ansuini et al., 2019; Pope et al., 2021; Feng et al., 2022; Huh et al., 2023). In such low-rank regimes, parameter transformations can induce functions that agree closely on the inputs actually encountered by the network, while differing substantially elsewhere. Here, we connect this phenomenon to LMC. For example, recent work proposed methods to *break* structural parameter symmetries (Lim et al., 2024b; Ziyin et al., 2025), and observed that such interventions can yield *unaligned* LMC, which indicates the absence of permutation symmetry: the assignment of features to neurons is fixed. But not all such interventions lead to unaligned LMC. We argue that this outcome may indicate additional symmetries: even if a structural symmetry is broken, approximate or data-dependent symmetries may remain intact on the inputs the network actually sees. Hence, we use symmetry breaking and LMC as tools to study the interplay of data, representation geometry and neural symmetries. Specifically, we ask:

> When does a given parameter symmetry breaking mechanism select a consistent assignment of features to neurons across runs, and how do the data and representation distribution control its effectiveness?

**Our Contributions.** We develop a unifying theoretical framework for *neuron identifiability*: the consistent assignment of features to neurons across random training seeds. In each layer, we view neurons functionally on their input support. Given a symmetry breaking mechanism, we characterize the corresponding *effective function classes* of neurons (i.e., which functions they can implement on their input support), and evaluate *realization costs* of functions (the minimum weight norm required to implement them) w.r.t. different neurons' function classes. This allows us to derive conditions for neuron identifiability.

Our analysis highlights that effective symmetry breaking is controlled by which architectural perturbations are observable on the input support. In particular, symmetries can be broken in raw parameter space, yet remain effectively available on the input support (see Fig. 1). We further identify conditions under which hidden-layer representations can be merged without alignment, i.e., unaligned LMC. Our findings systematically explain previously observed empirical phenomena. Specifically, we show that symmetry breaking is not binary, but governed by the interaction between architecture, data geometry, and effective function classes.

## 2. Related Work

**Linear Mode Connectivity.** A growing body of work argues that independently trained neural networks are often

not separated by high-loss barriers, suggesting that good solutions form connected regions (Garipov et al., 2018; Draxler et al., 2018). Linear mode connectivity (LMC) strengthens this view by asking when the straight-line interpolation between two trained solutions stays within a low-loss region (Frankle et al., 2020). Layer-wise connectivity (Adilova et al., 2024) further proposed to analyze by aligning networks at the layer level. A major practical barrier to LMC is parameter symmetry, which can make functionally similar models appear misaligned in weight space (Entezari et al., 2022). These symmetries also govern how we analyze or merge weights, and thus act as the data symmetries in the growing area of weight space learning (Andrychowicz et al., 2016; Zhou et al., 2023a; Manor et al., 2026; Han et al., 2026; Kahana et al., 2025).

**Weight Space Alignment.** Weight-space alignment methods explicitly compute neuron-to-neuron correspondences between independently trained networks before any interpolation or parameter merging, typically by solving an assignment problem that makes the two weight tensors comparable under the symmetries of the parameterization (e.g., hidden-unit permutations) (Ainsworth et al., 2023; Singh & Jaggi, 2020; Peña et al., 2023). Recent work shifts from solving a discrete matching instance per pair of models to learning the alignment map itself (Navon et al., 2024; Shamsian et al., 2024). Subsequent work explores weight merging through explicit alignment pipelines, e.g., representation-based matching with interpolation repair (Li et al., 2020; Jordan et al., 2023), and extends them to additional architectures (e.g., transformers) and richer reparameterization families beyond simple permutations (Imfeld et al., 2024; Verma & Elbayad, 2024; Theus et al., 2025).

## 3. Preliminaries

We first briefly introduce parameter symmetries, and then describe a simple architectural mechanism for breaking them.

### 3.1. Weight Space and Parameter Symmetries

Let a neural network be specified by a parameter vector $\boldsymbol{\theta}$ in a parameter space $\Theta$. Denote $f_{\boldsymbol{\theta}} : \mathcal{X} \to \mathcal{Y}$ the function implemented by the network with parameters $\boldsymbol{\theta}$, mapping the input space $\mathcal{X}$ to the output space $\mathcal{Y}$. A parameter symmetry is a transformation $\varphi : \Theta \to \Theta$ s.t. $f_{\varphi(\boldsymbol{\theta})} = f_{\boldsymbol{\theta}} \; \forall \boldsymbol{\theta} \in \Theta$.

**Permutation Symmetries.** Consider a 2-layer MLP parameterized by $\boldsymbol{\theta} = (\boldsymbol{W}_2, \boldsymbol{W}_1) \in \Theta := \mathbb{R}^{d_{\text{out}} \times m} \times \mathbb{R}^{m \times d_{\text{in}}}$:

$$f_{\boldsymbol{\theta}}(\boldsymbol{x}) := \boldsymbol{W}_2 \eta(\boldsymbol{W}_1 \boldsymbol{x}), \qquad (1)$$

where $\eta$ is an elementwise nonlinearity. For any permutation $\pi \in S_m$ with permutation matrix $\boldsymbol{P}_\pi \in \{0,1\}^{m \times m}$, the mapping $(\boldsymbol{W}_2, \boldsymbol{W}_1) \mapsto (\boldsymbol{W}_2 \boldsymbol{P}_\pi^\top, \boldsymbol{P}_\pi \boldsymbol{W}_1)$ is a parameter symmetry, since $\boldsymbol{W}_2 \boldsymbol{P}_\pi^\top \eta(\boldsymbol{P}_\pi \boldsymbol{W}_1 \boldsymbol{x}) = \boldsymbol{W}_2 \boldsymbol{P}_\pi^\top \boldsymbol{P}_\pi \eta(\boldsymbol{W}_1 \boldsymbol{x}) = \boldsymbol{W}_2 \eta(\boldsymbol{W}_1 \boldsymbol{x})$ for $\boldsymbol{x} \in \mathbb{R}^{d_{\text{in}}}$. In other

words, hidden units are interchangeable, i.e., permuting them and compensating in the adjacent layer does not change the computed function. Depending on $\eta$, this group can also be significantly larger (e.g., diagonal rescalings for positively homogeneous $\eta$ or invertible matrices for linear $\eta$). Similar symmetries appear in modern models, e.g., channels in CNNs, heads in multi-head attention, or through any computation graph automorphisms (Lim et al., 2024a;b).

**Symmetry Breaking.** Symmetry breaking refers to any modification of a network's forward pass that reduces the effective parameter symmetry group so that fewer distinct parameters represent the same function. Equivalently, it aims to make the map $\boldsymbol{\theta} \mapsto f_{\boldsymbol{\theta}}$ closer to injective. In this work, we study an architectural intervention that replaces each trainable weight matrix $\boldsymbol{W}$ with an effective matrix

$$\boldsymbol{W}_{\text{eff}} = \mathbf{F} + \mathbf{D} \odot \boldsymbol{W}, \tag{2}$$

where $\mathbf{F}, \mathbf{D} \in \mathbb{R}^{m \times d}$ are considered fixed parts of the architecture, $\odot$ denotes elementwise multiplication, and $\boldsymbol{W} \in \mathbb{R}^{m \times d}$ contains the trainable parameters that are randomly initialized independently across runs and then optimized. $\mathbf{F}, \mathbf{D} \in \mathbb{R}^{m \times d}$ can be fixed upfront or sampled once per architecture. As summarized in Table 1, this subsumes several existing parameter symmetry breaking schemes.

*Table 1.* Symmetry breaking schemes covered by (2).

| | $\mathbf{F}$ | $\mathbf{D}$ |
|---|---|---|
| **$W$-asym.** (Lim et al., 2024b) | $\mathbf{M} \odot \mathbf{B}$ where $\mathbf{B} \sim \mathcal{N}(0, \sigma_{\mathbf{F}}^2)$ i.i.d., $\mathbf{M} \in \{0,1\}^{m \times d}$ | $\mathbf{1}\mathbf{1}^\top - \mathbf{M}$ |
| *syre* (Ziyin et al., 2025) | $\mathcal{N}(0, \sigma_{\mathbf{F}}^2)$ i.i.d. | $\text{Unif}(1-\varepsilon, 1+\varepsilon)^{-1/2}$ i.i.d. |
| Linear residual | $\boldsymbol{I}_d$ | $\mathbf{1}\mathbf{1}^\top$ |
| Sparse network | $0$ | $\mathbf{M} \in \{0,1\}^{m \times d}$ |

For *$W$-asymmetric networks* (Lim et al., 2024b), a binary mask $\mathbf{M}$ selects a subset of coordinates that $\mathbf{F} := \mathbf{M} \odot \mathbf{B}$ fixes to constant random values, while unmasked coordinates remain trainable. Formally, Lim et al. (2024b) show that if $\mathbf{M}$ has pairwise distinct nonzero rows, all architecture-induced symmetries of a neural DAG (in particular, permutations of hidden units) are broken. In *syre*, Ziyin et al. (2025) directly draw $\mathbf{F}$ and $\mathbf{D}$ i.i.d., and show that, almost surely, more general reflection symmetries of the loss under $\ell^2$ weight decay are removed. Linear residual connections and sparse networks, both of which are known to break hidden-unit permutation symmetries (Lim et al., 2024b; Zhao et al., 2025), can likewise be interpreted in our framework.

### 3.2. Setup and Assumptions

In our work, we will consider an asymmetric (i.e., symmetry-broken) layer with the intervention in the form of (2) in isolation, since permutation symmetries and their breaking act locally at the level of neurons. Concretely, we study the layer map $\boldsymbol{H} : \mathbb{R}^{m \times d} \times \mathbb{R}^d \to \mathbb{R}^m$,

$$\boldsymbol{H}(\boldsymbol{W}; \boldsymbol{x}) := \eta((\mathbf{F} + \mathbf{D} \odot \boldsymbol{W})\boldsymbol{x}), \tag{3}$$

where $\mathbf{F}, \mathbf{D}$ are fixed as in (2) and $\eta : \mathbb{R} \to \mathbb{R}$ acts elementwise. Let $\mathbf{f}_i, \mathbf{d}_i$, and $\boldsymbol{w}_i$ denote the $i$-th rows of $\mathbf{F}, \mathbf{D}$, and $\boldsymbol{W}$. The layer's $i$-th neuron in (3) can be written as $\boldsymbol{h}_i(\boldsymbol{w}_i; \boldsymbol{x}) := \eta((\mathbf{f}_i + \mathbf{d}_i \odot \boldsymbol{w}_i)^\top \boldsymbol{x})$. Let the input $\boldsymbol{x}$ (data or representations) be supported on a measurable subset $\mathcal{X} \subset \mathbb{R}^d$ and drawn from a distribution $\mathbb{P}$ on $\mathcal{X}$. Define

$$\mathcal{H}(\mathcal{X}) := \{\boldsymbol{H}(\boldsymbol{W}; \cdot) \mid \boldsymbol{W} \in \mathbb{R}^{m \times d}\} \subset \{\mathcal{X} \to \mathbb{R}^m\}, \tag{4}$$

and analogously $\mathcal{H}_i(\mathcal{X}) \subset \{\mathcal{X} \to \mathbb{R}\}$ as the set of functions a layer (or single neuron) can implement. As $\mathcal{X}$ is typically far from filling the ambient space $\mathbb{R}^d$, the input support strongly influences symmetries. For our theoretical analysis, we model this via a linear subspace assumption:

**Assumption 3.1** (Subspace support model). The distribution $\mathbb{P}$ is supported on a $k$-dimensional subspace $\mathcal{U} \subset \mathbb{R}^d$, i.e., $\mathbb{P}_{\boldsymbol{x}}(\boldsymbol{x} \in \mathcal{U}) = 1$. Throughout, we fix an orthonormal basis $\boldsymbol{U} \in \mathbb{R}^{d \times k}$ of $\mathcal{U}$, i.e., $\boldsymbol{U}^\top \boldsymbol{U} = \boldsymbol{I}_k$ and $\text{im}(\boldsymbol{U}) = \mathcal{U}$.

Assumption 3.1 is a standard low-dimensional structure hypothesis: although inputs live in a high-dimensional ambient space, their variability is often well-approximated by a low-dimensional set (Ansuini et al., 2019; Pope et al., 2021; Feng et al., 2022; Huh et al., 2023; Papyan et al., 2020). In our setting, the subspace model enables exact projections onto $\mathcal{U}$ and provides explicit coordinates via $\boldsymbol{U}$.

## 4. Neuron Identifiability

In this section, we relate the effectiveness of a symmetry breaking intervention according to (2) to its ability to provide neurons with distinguishable identities that lead to consistent assignments across independent training runs. We show that effective symmetry breaking incurs high weight norm cost to realize non-identity permutations, and quantify realization costs depending on $\mathbf{F}, \mathbf{D}$.

Within one layer of a neural network, each neuron can be viewed as implementing a particular feature. Across independent training runs, a layer often realizes essentially the same *set* of features, and runs differ mainly by a permutation that matches neurons with approximately the same functionality. In a symmetric layer, this assignment is determined by the initialization. By *effective symmetry breaking* we mean that, instead, architectural asymmetries bias training towards a consistent assignment. We analyze this through each neuron's *effective function class*, i.e., the functions it can realize on the input support together with their *realization costs*, measured by the minimum weight norm required to implement them. This viewpoint is well-motivated under explicit weight decay and the implicit norm bias of gradient-based optimization. In an asymmetric layer, two neurons can both be capable of implementing a feature, yet one can require a much larger norm. We quantify symmetry breaking via a *permutation sensitivity*, i.e., the norm cost change from forcing a reassignment of features to different neurons. The underlying premise is that uniformly large reassign-

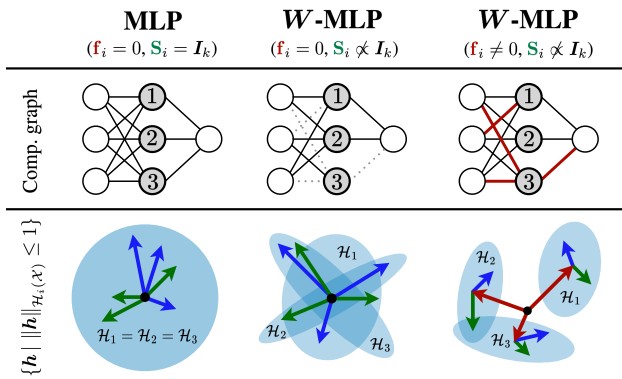

| **MLP** | $W$**-MLP** | $W$**-MLP** |
|---|---|---|
| $(\mathbf{f}_i = 0, \mathbf{S}_i = \mathbf{I}_k)$ | $(\mathbf{f}_i = 0, \mathbf{S}_i \not\propto \mathbf{I}_k)$ | $(\mathbf{f}_i \neq 0, \mathbf{S}_i \not\propto \mathbf{I}_k)$ |

*Figure 2.* Effective function classes and learned features (**run A**/**run B**) with varying levels of symmetry. *Col. 1:* In a fully symmetric MLP, each neuron can implement the same functions on $\mathcal{X}$. *Col. 2:* Pruning weights affects $\mathbf{S}_i$ and can introduce anisotropy. *Col. 3:* Fixed weights via $\mathbf{F}$ introduce functional biases.

ment costs over nontrivial permutations are equivalent to there being a unique minimum-complexity assignment of the feature set to neurons. For this analysis, we impose the following assumption on the input support.

**Assumption 4.1** (Non-degeneracy)**.** *Under* Assumption 3.1, *additionally assume the following. If $\eta$ is injective (e.g., $\tanh$, LeakyReLU), assume $\mathrm{span}(\mathrm{supp}(\mathbb{P})) = \mathcal{U}$. If $s \mapsto (\eta(s), \eta(-s))$ is injective (e.g., ReLU, GELU), assume there is $T \subseteq \mathrm{supp}(\mathbb{P})$ s.t. $T = -T$ and $\mathrm{span}(T) = \mathcal{U}$.*

The condition is relatively mild in the injective case and intentionally stronger for non-injective activations. While we expect the results in this section to extend to weaker distributional assumptions, it allows us to derive our results in terms of linear algebra via the following proposition.

**Proposition 4.2.** *Define $\mathbf{v}_i$ as the projection of the fixed center of the $i$-th asymmetric neuron to the input subspace, and $\mathbf{M}_i$ as its projected diagonal operator, i.e.,*

$$\mathbf{v}_i := \boldsymbol{U}^\top \mathbf{f}_i \in \mathbb{R}^k, \quad \mathbf{M}_i := \boldsymbol{U}^\top \mathrm{Diag}(\mathbf{d}_i) \in \mathbb{R}^{k \times d}. \quad (5)$$

*Under* Assumptions 3.1 *and* 4.1, *$h \in \mathcal{H}_i(\mathcal{X})$ belonging to weight $\boldsymbol{w}$ can be written as $h(\boldsymbol{x}) = \eta((\boldsymbol{U}\boldsymbol{a})^\top \boldsymbol{x})$, where*

$$\boldsymbol{a} := \boldsymbol{U}^\top (\mathbf{f}_i + \mathrm{Diag}(\mathbf{d}_i)\boldsymbol{w}) = \mathbf{v}_i + \mathbf{M}_i \boldsymbol{w} \in \mathbb{R}^k. \quad (6)$$

*The mapping $h \mapsto \boldsymbol{a}$ is a bijection that identifies $\mathcal{H}_i(\mathcal{X})$ with $\mathbf{v}_i + \mathrm{im}(\mathbf{M}_i) \subseteq \mathbb{R}^k$. In particular, $\mathcal{H}_i(\mathcal{X})$ is an affine subspace of $\mathbb{R}^k$ of dimension $\mathrm{rank}(\mathbf{M}_i) \leq k$.*

The proof of Prop. 4.2 is in § C.1. Prop. 4.2 provides a geometric comparison of neuron function classes via the induced subspace preactivation coefficient $\boldsymbol{a}$. It especially implies that if all $\mathbf{M}_i$ have full rank, all neurons can represent exactly the same functions:

$$\mathcal{H}_1(\mathcal{X}) = \cdots = \mathcal{H}_m(\mathcal{X}) \cong \mathbb{R}^k. \quad (7)$$

Thus, even neurons receiving incomplete information (e.g., some inputs masked out) may still be able to realize all functions on the input because of the low effective dimension of the input space. For example, the intensity of a masked-out

input pixel might be recoverable by reading out values of correlated pixels that have not been masked out. In the sequel, we mainly focus on the case (7), and show that even when different neurons are exchangeable in terms of the functions they can represent, some of these functions may still come at a higher weight cost than others. Consequently, even if $\mathcal{H}_i = \mathcal{H}_j$, the $i$-th and $j$-th neuron may still be assigned consistent roles across training runs, by virtue of the optimizer favoring solutions of small weight norm.

### 4.1. Realization Costs

Having characterized *which* functions a neuron can represent, we now ask to quantify *at which cost*. This motivates a complexity measure given by the minimal parameter norm needed for neuron $i$ to realize a target function. We define the *realization cost*[1] of $h : \mathcal{X} \to \mathbb{R}$ w.r.t. $\mathcal{H}_i(\mathcal{X})$ as

$$\|\boldsymbol{h}\|_{\mathcal{H}_i(\mathcal{X})} := \inf \{ \|\boldsymbol{w}\|_2 \mid \boldsymbol{h} = \boldsymbol{h}_i(\boldsymbol{w}; \cdot) \}, \quad (8)$$

with the convention $\|\boldsymbol{h}\|_{\mathcal{H}_i(\mathcal{X})} := +\infty$ for $\boldsymbol{h} \notin \mathcal{H}_i(\mathcal{X})$. Define the global version $\|\cdot\|_{\mathcal{H}(\mathcal{X})}$ analogously w.r.t. the Frobenius norm $\|\cdot\|_F$.[2]

By Assumptions 3.1 and 4.1, the realization cost w.r.t. the $i$-th neuron in (8) can be computed in a closed form as a Mahalanobis seminorm. Namely, define the Gram matrix

$$\mathbf{S}_i := \mathbf{M}_i \mathbf{M}_i^\top = \boldsymbol{U}^\top \mathrm{Diag}(\mathbf{d}_i)^2 \boldsymbol{U} \in \mathbb{R}^{k \times k}. \quad (9)$$

Then, for a function $\boldsymbol{h} \in \mathcal{H}_i(\mathcal{X})$ expressible by the $i$-th neuron, $\boldsymbol{h} = \eta((\boldsymbol{U}\boldsymbol{a})^\top \cdot)$ (Prop. 4.2), the realization cost is

$$\left\| \eta((\boldsymbol{U}\boldsymbol{a})^\top \cdot) \right\|_{\mathcal{H}_i(\mathcal{X})} = \|\boldsymbol{a} - \mathbf{v}_i\|_{\mathbf{S}_i^\dagger}. \quad (10)$$

Here, $\|\boldsymbol{x}\|_{\boldsymbol{A}} := \sqrt{\boldsymbol{x}^\top \boldsymbol{A} \boldsymbol{x}}$ denotes the Mahalanobis norm and $\mathbf{S}_i^\dagger$ the Moore-Penrose pseudoinverse of $\mathbf{S}_i$.

This demonstrates that the realization costs w.r.t. neuron $i$ depend crucially on the anisotropy of its Gram matrix. Intuitively, $\mathbf{S}_i$ acts like a direction-dependent cost map: the realization cost grows the further $\boldsymbol{a}$ is from the classes' projected centers $\mathbf{v}_i$, but possibly more so in some directions than others (see Fig. 2 for an illustration). Along an eigenvector of $\mathbf{S}_i$ with eigenvalue $\lambda$, the cost scales like $1/\sqrt{\lambda}$ (i.e., small $\lambda$ is costly), and directions outside $\mathrm{im}(\mathbf{S}_i)$ ($= \mathrm{im}(\mathbf{M}_i)$) are not realizable at all (i.e., have infinite cost). If $\mathbf{S}_i \approx c\,\boldsymbol{I}_k$, the cost is near-Euclidean, whereas anisotropy makes some features much more expensive than others. This is illustrated in the example below.

**Example 4.3.** Consider a $W$-asym. layer with two neurons and 4D inputs that live on a 2D subspace $\mathcal{U}_\varepsilon = \mathrm{span}\{\boldsymbol{u}_1, \boldsymbol{u}_2\}$, with $\boldsymbol{u}_1 = (1, \varepsilon, 0, 0)^\top, \boldsymbol{u}_2 = (0, 0, 1, \varepsilon)^\top$, for some $\varepsilon \in [0, 1]$. Let $\mathbf{F} = 0, \mathbf{d}_1 = (1, 0, 0, 1)^\top, \mathbf{d}_2 = $

---

[1]This is the *representation cost* of Dai et al. (2021); Gunasekar et al. (2018); Savarese et al. (2019). We call it *realization cost* to avoid confusion with *representations* in a layer.

[2]In general, $\|\cdot\|_{\mathcal{H}(\mathcal{X})}$ is not a norm (e.g., $\boldsymbol{H}(0; \cdot) = \eta(\mathbf{F} \cdot)$ can be nonzero while having cost 0).

$(0, 1, 1, 0)^\top$. If $\varepsilon$ is small, neuron 1 can pick up on information in the $\boldsymbol{u}_2$ direction only at high weight cost, while neuron 2 can pick up on it cheaply; vice-versa, neuron 1 can pick up on the $\boldsymbol{u}_1$ direction cheaply and neuron 2 cannot. This is reflected in anisotropic $\mathbf{S}_1 \propto \mathrm{Diag}(1, \varepsilon^2)$ and $\mathbf{S}_2 \propto \mathrm{Diag}(\varepsilon^2, 1)$. The realization cost $\|\boldsymbol{h}_2\|_{\mathcal{H}_1}$ of neuron 2's function w.r.t. neuron 1's class becomes arbitrarily large as $\varepsilon \to 0$, and $\boldsymbol{h}_2$ cannot be represented by $\mathcal{H}_1$ if $\varepsilon = 0$ (unless $\boldsymbol{h}_2$ does not vary in the direction of $\boldsymbol{u}_2$).

The issue of certain directions of the input being masked out is closely connected to the *subspace coherence*

$$\nu(\mathcal{U}) := \max_{\ell \in [d]} (\boldsymbol{U}\boldsymbol{U}^\top)_{\ell\ell} \in [k/d, 1] \quad (11)$$

(Candes & Tao, 2005; Candes & Recht, 2009).[3] The subspace coherence $\nu(\mathcal{U})$ quantifies how much of the $k$-dimensional variability of the input support can be seen from any single ambient coordinate. It is low ($\approx k/d$) when the intrinsic signal directions are spread uniformly across the raw coordinates, and high ($\approx 1$) when few coordinates dominate the input variation. High coherence makes the effect of coordinate masking on symmetry breaking particularly pronounced, while low coherence can diffuse it across ambient dimensions. In Example 4.3, $\nu(\mathcal{U}_\varepsilon) = (1 + \varepsilon^2)^{-1}$, which tends to 1 as $\varepsilon \to 0$, exactly in the regime where masking ambient coordinates makes certain directions essentially inaccessible and, thus, realization costs highly anisotropic. The following theorem bounds the anisotropy of $\mathbf{S}_i$ in terms of $\nu(\mathcal{U})$ if the diagonal asymmetry $\mathbf{d}_i$ is i.i.d. sampled.

**Theorem 4.4** (Spectral concentration of $\mathbf{S}_i$). *Let $i \in [m]$ and sample the entries of $\mathbf{d}_i$ i.i.d. with $(\mathbf{d}_i)_\ell^2$ having mean $\mu_{\mathbf{D}}$, variance $\sigma_{\mathbf{D}}^2$, and a.s. deviation $b_{\mathbf{D}}$ from its mean (all $< \infty$). For any $\delta \in (0, 1)$, with probability at least $1 - \delta$,*

$$\|\mathbf{S}_i - \mu_{\mathbf{D}}\boldsymbol{I}_k\|_{\mathrm{op}} \leq C \left\{ \sqrt{\sigma_{\mathbf{D}}^2 \nu(\mathcal{U})\Lambda} + b_{\mathbf{D}}\nu(\mathcal{U})\Lambda \right\}, \quad (12)$$

*where $\Lambda := \log(2k/\delta)$, $\nu(\mathcal{U}) := \max_{\ell \in [d]} (\boldsymbol{U}\boldsymbol{U}^\top)_{\ell\ell}$ is the subspace coherence, and $C$ is a universal constant.*[4]

The proof of Thm. 4.4 can be found in § C.1. Note that if $\|\mathbf{S}_i - \mu_{\mathbf{D}}\boldsymbol{I}_k\|_{\mathrm{op}} < \mu_{\mathbf{D}}$, $\mathbf{M}_i$ has full rank, so every other neuron can be represented by $\mathcal{H}_i(\mathcal{X})$. More generally, the interpretation of Thm. 4.4 is that while $\mathbf{D}$ on its own can restrict which neurons can represent which functions, its impact (under sampling of $\mathbf{D}$) is usually not severe if the input subspace is sufficiently incoherent, which one would expect on real-world input data or hidden-layer representations (Pope et al., 2021; Ansuini et al., 2019). Hence, one should not automatically expect $\mathbf{D}$ alone to effectively break symmetries. We discuss in the next sections the interplay of $\mathbf{D}$ with the second component, the fixed weights $\mathbf{F}$.

---

[3]Note that $\nu(\mathcal{U})$ is defined in terms of the orthogonal projection $\boldsymbol{U}\boldsymbol{U}^\top$ on $\mathcal{U}$ and does not depend on the chosen basis.

[4]One can also control $\max_i \|\mathbf{S}_i - \mu_{\mathbf{D}}\boldsymbol{I}_k\|_{\mathrm{op}}$ via a union bound at the cost of replacing $\log(2k/\delta)$ by $\log(2mk/\delta)$.

## 4.2. Realization Cost Sensitivity

To quantify the degree of permutation symmetry breaking, we now ask: How much realization cost does it incur if we reassign which neuron implements which function? While this cost is invariant to permutations in a fully symmetric layer, the same set of functions can have different realization cost after reassignment in an asymmetric layer.

For any permutation $\pi \in S_m$, write $\boldsymbol{P}_\pi \in \{0, 1\}^{m \times m}$ for its corresponding permutation matrix. For a layer $\boldsymbol{H} : \mathbb{R}^d \to \mathbb{R}^m$, define a group action of $(\pi, \tau) \in S_m \times S_d$, where $\tau$ permutes the inputs and $\pi$ permutes the outputs (neurons):

$$\big((\pi, \tau) \cdot \boldsymbol{H}\big)_i := \boldsymbol{h}_{\pi(i)}(\boldsymbol{P}_\tau^\top \cdot). \quad (13)$$

Consider an MLP $\boldsymbol{H}^{1:L} = \boldsymbol{H}^L \circ \cdots \circ \boldsymbol{H}^1$ and let $\pi_\ell \in S_{m_\ell}$ denote a permutation of the $\ell$-th layer's neurons. The product $\prod_{\ell=0}^{L} \pi_\ell$ acts on $\boldsymbol{H}^{1:L}$ by applying $(\pi_\ell, \pi_{\ell-1})$ to each individual layer $\boldsymbol{H}^\ell$ (where $\pi_0 = \pi_L := \mathrm{id}$). We define the *(squared) realization cost sensitivity* $\Delta_{\pi,\tau}(\boldsymbol{H}; \mathcal{X})$ of a layer $\boldsymbol{H}$ to an output-input permutation pair $(\pi, \tau)$ as

$$\Delta_{\pi,\tau}(\boldsymbol{H}; \mathcal{X}) := \|(\pi, \tau) \cdot \boldsymbol{H}\|_{\mathcal{H}(\boldsymbol{P}_\tau \mathcal{X})}^2 - \|\boldsymbol{H}\|_{\mathcal{H}(\mathcal{X})}^2 \quad (14)$$

(we will often omit $\mathcal{X}$ for brevity). It decomposes as

$$\Delta_{\pi,\tau}(\boldsymbol{H}; \mathcal{X}) = \underbrace{\Delta_{\pi,\mathrm{id}}\big((\mathrm{id}, \tau)\cdot\boldsymbol{H}; \boldsymbol{P}_\tau\mathcal{X}\big)}_{=:\Delta_\pi^{\mathrm{out}}} + \underbrace{\Delta_{\mathrm{id},\tau}(\boldsymbol{H}; \mathcal{X})}_{=:\Delta_\tau^{\mathrm{in}}}. \quad (15)$$

This allows us to separately consider the effects of permuting the inputs (via $\tau$) and outputs (via $\pi$) of a layer, for which we will write $\Delta_\tau^{\mathrm{in}}$ and $\Delta_\pi^{\mathrm{out}}$, respectively. In other words, for an MLP, a single $\pi_\ell$ contributes to the total cost difference via both $\Delta_\pi^{\mathrm{out}}(\boldsymbol{H}^\ell)$, and $\Delta_\pi^{\mathrm{in}}(\boldsymbol{H}^{\ell+1})$. We first quantify the sensitivity to neuron reassignments $\pi$ with input coordinates fixed in § 4.3, and then discuss input coordinate permutations $\tau$ in § 4.4. Results for joint permutations $(\pi, \tau)$ can be obtained by applying the results from § 4.3 to $\boldsymbol{P}_\tau\mathcal{X}$.

## 4.3. Sensitivity to Neuron Reassignments

We first focus on the effect of output permutations $\pi$. For this, we assume that all $\mathbf{M}_i$, $i \in [m]$, have full rank, i.e., all neurons' function classes coincide on the input support: $\mathcal{H}_1(\mathcal{X}) = \cdots = \mathcal{H}_m(\mathcal{X})$ (by Prop. 4.2). Given a layer $\boldsymbol{H}$ with components $\boldsymbol{h}_i = \eta((\boldsymbol{U}\boldsymbol{a}_i)^\top \cdot)$, it is convenient to recenter by defining $\boldsymbol{r}_i := \boldsymbol{a}_i - \mathbf{v}_i$ to be the projected trainable part of neuron $i$. We decompose $\Delta_\pi^{\mathrm{out}}(\boldsymbol{H})$ for the special case where $\pi = (ij) \in S_m$, $i \neq j$ is a transposition (the cost to *swap* neurons $i$ and $j$):

$$\Delta_{(ij)}^{\mathrm{out}}(\boldsymbol{H}) = \underbrace{\|\mathbf{v}_j - \mathbf{v}_i\|_{\mathbf{S}_i^{-1} + \mathbf{S}_j^{-1}}^2}_{(i)\ \text{center term}} + \underbrace{2\langle \mathbf{v}_j - \mathbf{v}_i, \mathbf{S}_i^{-1}\boldsymbol{r}_j - \mathbf{S}_j^{-1}\boldsymbol{r}_i \rangle}_{(ii)\ \text{cross term}}$$

$$+ \underbrace{\|\boldsymbol{r}_j\|_{\mathbf{S}_i^{-1} - \mathbf{S}_j^{-1}}^2 + \|\boldsymbol{r}_i\|_{\mathbf{S}_j^{-1} - \mathbf{S}_i^{-1}}^2}_{(iii)\ \text{metric mismatch term}}. \quad (16)$$

The decomposition in (16) isolates three sources of non-invariance: *(i)* The *center* term penalizes swapping two neu-

rons whose fixed centers differ on $\mathcal{X}$. It is large when $\mathbf{v}_i$ and $\mathbf{v}_j$ are far apart in the cost geometry (i.e., Mahalanobis metric) induced by the two neurons. *(ii)* The *cross* term captures how the trainable $\boldsymbol{r}_i, \boldsymbol{r}_j$ compensate or amplify this center mismatch. *(iii)* The *metric mismatch* term compares $\mathbf{S}_i^{-1}$ and $\mathbf{S}_j^{-1}$ on the remainders, and is nonzero only when the two neurons induce different cost geometries.

We now show that in a setting where the fixed centers $\mathbf{F}$ are on a larger scale than the trainable weights, the center term *(i)* dominates in (16), which we call the *center-dominated regime*. This is relevant as both $\boldsymbol{W}$-asymmetric (Lim et al., 2024b) and *syre* networks (Ziyin et al., 2025) rely on large fixed weights $\mathbf{F}$ to obtain unaligned LMC. To this end, we introduce notations to jointly capture the differences of projected centers incurred by a permutation. First, define

$$\boldsymbol{\delta}_\pi^{\text{out}} := (\boldsymbol{P}_\pi - \boldsymbol{I}_m)\mathbf{F}\boldsymbol{U} \in \mathbb{R}^{m \times k}, \qquad (17)$$

which for $\pi = (ij)$ is simply $(\boldsymbol{\delta}_\pi^{\text{out}})_{i,:} = \mathbf{v}_j - \mathbf{v}_i$, $(\boldsymbol{\delta}_\pi^{\text{out}})_{j,:} = \mathbf{v}_i - \mathbf{v}_j$, and $(\boldsymbol{\delta}_\pi^{\text{out}})_{i',:} = 0$ for $i' \neq i, j$. Next, we need an assumption on the spectral concentration of $\mathbf{S}_i$ (by Thm. 4.4) to control the cross and metric mismatch terms *(ii)* and *(iii)* in (16). Fix $\varepsilon \in (0, \frac{\mu_{\mathbf{D}}}{2})$, and assume

$$\max_{i \in [m]} \|\mathbf{S}_i - \mu_{\mathbf{D}}\boldsymbol{I}_k\|_{\text{op}} \leq \varepsilon. \qquad (18)$$

**Theorem 4.5** (Global output sensitivity). *Consider a layer* $\boldsymbol{H} = \eta\big((\mathbf{F} + \mathbf{D} \odot \boldsymbol{W})\cdot\big)$*, and assume (18) holds. Set* $\boldsymbol{R} := (\mathbf{D} \odot \boldsymbol{W})\boldsymbol{U} \in \mathbb{R}^{m \times k}$ *(proj. trainable parts). Let* $\pi \in S_m$ *and set* $\boldsymbol{M}_\pi := \boldsymbol{I}_m - \text{Diag}\big(\text{diag}(\boldsymbol{P}_\pi)\big)$ *to be the projection onto the coordinates affected by* $\pi$*. Assume that* $\boldsymbol{\delta}_\pi^{\text{out}} \neq 0$*,* $\rho_\pi^{\text{out}} := \|\boldsymbol{M}_\pi\boldsymbol{R}\|_F / \|\boldsymbol{\delta}_\pi^{\text{out}}\|_F \leq 1$*. Then,*

$$\Delta_\pi^{\text{out}}(\boldsymbol{H}) = \Big(1 \pm \mathcal{O}(\rho_\pi^{\text{out}} + \varepsilon\mu_{\mathbf{D}}^{-1})\Big) \cdot \mu_{\mathbf{D}}^{-1}\|\boldsymbol{\delta}_\pi^{\text{out}}\|_F^2. \quad (19)$$

The proof of Thm. 4.5 is in § C.2. Thm. 4.5 shows that if the scale of fixed center differences dominates the scale of trainable parts affected by $\pi$, $\Delta_\pi^{\text{out}}$ scales primarily as $\mu_{\mathbf{D}}^{-1}\|\boldsymbol{\delta}_\pi^{\text{out}}\|_F^2$. Further, if $\rho^{\text{out}} := \sup_{i \neq j} \rho_{(ij)}^{\text{out}} \leq 1$, it yields

$$\min_{\pi \neq \text{id}} \Delta_\pi^{\text{out}}(\boldsymbol{H}) = \Big(1 \pm \mathcal{O}(\rho^{\text{out}} + \varepsilon\mu_{\mathbf{D}}^{-1})\Big) \cdot \mu_{\mathbf{D}}^{-1}\gamma_{\text{out}}^2, \quad (20)$$

where $\gamma_{\text{out}} := \min_{i \neq j}\|\boldsymbol{\delta}_{(ij)}^{\text{out}}\|_F = \sqrt{2}\min_{i \neq j}\|\mathbf{v}_j - \mathbf{v}_i\|_2 = \min_{\pi \neq \text{id}}\|\boldsymbol{\delta}_\pi^{\text{out}}\|_F$ is the smallest gap between proj. centers. If $\gamma_{\text{out}}$ is low, neuron swaps become accessible at low weight cost, and symmetry breaking may become ineffective. However, in high dimensions, many vectors of similar norms can be packed without any pair becoming close. We now clarify how $\gamma_{\text{out}}$ scales when $\mathbf{F}$ is sampled, depending on the number of neurons $m$ and the intrinsic dimension $k$ (i.e., number of vectors being packed, and their dimension).

**Theorem 4.6** (Asymptotics of projected center gap $\gamma_{\text{out}}$). *Let* $\mathbf{F}$ *have i.i.d. entries of the form* $B \cdot \sigma_{\mathbf{F}}Y_0$*, where* $B \sim$ *Bernoulli*$(p)$*,* $p \in (0, 1]$*,* $Y_0$ *is independent of* $B$ *with a Lebesgue density bounded by* $M_0 < \infty$*, and* $\sigma_{\mathbf{F}} > 0$*. Set* $q := 2p - p^2$ *and fix* $\delta \in (0, 1)$*. Let* $m \geq 2$ *and assume*

$$q \geq C\big\{\sqrt{q\nu(\mathcal{U})\Lambda} + \nu(\mathcal{U})\Lambda\big\}, \quad \Lambda := \log(4km^2/\delta). \quad (21)$$

*Then, with probability at least* $1 - \delta$*,*

$$\gamma_{\text{out}} \geq c\frac{\sqrt{q}}{M_0}\sigma_{\mathbf{F}}V_k^{-1/k}\Big(\frac{\delta}{m^2}\Big)^{1/k} = \Theta(\sigma_{\mathbf{F}}\sqrt{k}m^{-2/k}), \quad (22)$$

*where* $V_k$ *denotes the volume of the* $k$*-dim. unit ball,* $V_k^{-1/k} = \Theta(\sqrt{k})$*, and* $c, C > 0$ *are absolute constants.*

The proof of Thm. 4.6 can be found in § C.2. Thm. 4.6 gives a lower bound on the separation between the projected centers. Importantly, if $k$ is reasonably large, $m^{-2/k}$ decays very slowly in $m$. Hence, $\gamma_{\text{out}}$ becomes small if $k$ is small or the number of neurons $m$ is very large; otherwise, a symmetry breaking method in the center-dominated regime of Thm. 4.5 can become effective. In contrast, consider the extreme case where the data only varies in one intrinsic dimension, then the offsets are forced onto a line, and $\gamma_{\text{out}}$ vanishes quickly as $m$ grows, so there are many approximate symmetries in wide layers. As an aside, the same would happen if we were to use a symmetry breaking mechanism where $\mathbf{F}$ varies in only one intrinsic dimension, which likely also explains the empirical observation of Lim et al. (2024b) that fixing biases does not effectively break symmetries.

### 4.4. Sensitivity to Input Permutations

We now study input permutations $\tau \in S_d$, i.e., reindexings of the $d$ ambient coordinates of the layer input $\boldsymbol{H}$ that may be induced upstream, whose realization cost sensitivity is $\Delta_\tau^{\text{in}}(\boldsymbol{H}) = \|\boldsymbol{H}(\boldsymbol{P}_\tau^\top\cdot)\|_{\mathcal{H}(\boldsymbol{P}_\tau\mathcal{X})}^2 - \|\boldsymbol{H}\|_{\mathcal{H}(\mathcal{X})}^2$. Note that $\tau$ can affect realization costs by *(i)* reindexing how a neuron's fixed part aligns with the data subspace, and *(ii)* changing the quadratic form that measures norm cost. An input reindexing $\tau$ changes the input support $\mathcal{X} \subseteq \mathcal{U}$ to $\boldsymbol{P}_\tau\mathcal{X} \subseteq \boldsymbol{P}_\tau\mathcal{U}$, thus we can use the new orthonormal basis $\boldsymbol{P}_\tau\boldsymbol{U} \in \mathbb{R}^{d \times k}$. Further, the Gram matrices $\mathbf{S}_i$ change to

$$\mathbf{S}_i^\tau := \boldsymbol{U}^\top\boldsymbol{P}_\tau^\top\text{Diag}(\mathbf{d}_i)^2\boldsymbol{P}_\tau\boldsymbol{U}, \quad i \in [m]. \qquad (23)$$

Analogously to (17), write

$$\boldsymbol{\delta}_\tau^{\text{in}} := \mathbf{F}(\boldsymbol{P}_\tau - \boldsymbol{I}_d)\boldsymbol{U} \in \mathbb{R}^{m \times k} \qquad (24)$$

for the difference introduced to the projected centers by an input reindexing $\tau \in S_d$. The corresponding realization cost sensitivity can now be related to $\boldsymbol{\delta}_\tau^{\text{in}}$ as follows.

**Theorem 4.7** (Global input sensitivity). *In the setup of Thm. 4.5, fix an input reindexing* $\tau \in S_d$*. Assume that the concentration event (18) holds both for* $\mathbf{S}_i$ *and* $\mathbf{S}_i^\tau$*.[5] Also assume that* $\boldsymbol{\delta}_\tau^{\text{in}} \neq 0$ *and* $\rho_\tau^{\text{in}} := \|\boldsymbol{R}\|_F / \|\boldsymbol{\delta}_\tau^{\text{in}}\|_F \leq 1$*. Then,*

$$\Delta_\tau^{\text{in}}(\boldsymbol{H}) = \Big(1 \pm \mathcal{O}(\rho_\tau^{\text{in}} + \varepsilon\mu_{\mathbf{D}}^{-1})\Big) \cdot \mu_{\mathbf{D}}^{-1}\|\boldsymbol{\delta}_\tau^{\text{in}}\|_F^2. \quad (25)$$

The proof of Thm. 4.7 can be found in § C.3. Further, if $\rho^{\text{in}} := \sup_{a \neq b} \rho_{(ab)}^{\text{in}} \leq 1$ is uniformly bounded over transpositions $(ab) \in S_d$, in analogy to (20), we can obtain

$$\min_{a \neq b} \Delta_{(ab)}^{\text{in}}(\boldsymbol{H}) = \Big(1 \pm \mathcal{O}(\rho^{\text{in}} + \varepsilon\mu_{\mathbf{D}}^{-1})\Big) \cdot \mu_{\mathbf{D}}^{-1}\gamma_{\text{in}}^2, \quad (26)$$

---

[5]When $\mathbf{D}$ has i.i.d. entries, $\mathbf{S}_i^\tau = \boldsymbol{U}^\top\boldsymbol{P}_\tau^\top\text{Diag}(\mathbf{d}_i)^2\boldsymbol{P}_\tau\boldsymbol{U} \overset{d}{=} \mathbf{S}_i$.

where $\gamma_{\text{in}} := \min_{a \neq b} \|\boldsymbol{\delta}^{\text{in}}_{(ab)}\|_F$.[6] Hence, similar to the neuron reassignment case, the degree of symmetry breaking depends crucially on this quantity. We again make the asymptotic scaling of $\gamma_{\text{in}}$ explicit when $\mathbf{F}$ is sampled as follows.

**Theorem 4.8** (Asymptotics of $\gamma_{\text{in}}$, informal). *Let $\mathbf{F}_{:,\ell} \in \mathbb{R}^m$ denote the $\ell$-th column of $\mathbf{F}$. Let $a \neq b \in [d]$. Then,*

$$\|\boldsymbol{\delta}^{\text{in}}_{(ab)}\|_F = \|\mathbf{F}_{:,b} - \mathbf{F}_{:,a}\|_2 \, \|\boldsymbol{U}_{b,:} - \boldsymbol{U}_{a,:}\|_2. \quad (27)$$

*If further $\mathbf{F}$ has i.i.d. centered subgaussian entries with variance $\sigma^2_{\mathbf{F}}$, subgaussian norm uniformly bounded by $C\sigma_{\mathbf{F}}$, and $m \gtrsim C^4 \log(d^2/\delta)$, then with probability at least $1 - \delta$,*

$$\gamma_{\text{in}} = \Theta(\sigma_{\mathbf{F}}\sqrt{m}) \cdot \min_{a \neq b} \|\boldsymbol{U}_{b,:} - \boldsymbol{U}_{a,:}\|_2. \quad (28)$$

Note that $\|\boldsymbol{U}_{a,:} - \boldsymbol{U}_{b,:}\|_2$ measures how distinguishable coordinates $a$ and $b$ are on the input. If swapping $a$ and $b$ acts approximately as the trivial action on the input support, i.e., $\boldsymbol{P}_{(ab)}\boldsymbol{x} \approx \boldsymbol{x}$ for $\boldsymbol{x} \in \mathcal{X}$, then $\|\boldsymbol{U}_{a,:} - \boldsymbol{U}_{b,:}\|_2 = \|\boldsymbol{U}^\top(\boldsymbol{e}_a - \boldsymbol{e}_b)\|_2 = \frac{1}{\sqrt{2}}\|(\boldsymbol{P}_{(ab)} - \boldsymbol{I}_d)\boldsymbol{U}\|_F \approx 0$, so the induced cost is small even if the scale of $\mathbf{F}$ is large. One always has $\|\boldsymbol{U}_{a,:} - \boldsymbol{U}_{b,:}\|_2 \leq 2\sqrt{\nu(\mathcal{U})}$, so small $\nu(\mathcal{U})$ can mitigate sensitivity to input reindexings. Hence, in this case, asymmetry matters precisely when it aligns with directions that the input varies along. The exact decomposition analogous to the neuron reassignment case (16), as well as a formal statement and proof of Thm. 4.8, can be found in § C.3.

### 4.5. Identifiability vs. Feature Learning

Our analysis suggests a tradeoff between *identifiability* and *feature learning*. To make a neuron assignment stable across runs, an architecture needs to break permutation symmetries *functionally*, i.e., induce a prior over which neuron functions are cheap and which are expensive. While we have seen in § 4.3 that $\mathbf{F}$ can enforce consistent neuron-feature matching and strong effective symmetry breaking, this also implies a restriction of the set of features that can be learned within a given norm budget. Thus, we posit that strong symmetry breaking comes with reduced feature learning. We formalize this in § C.4 by proving a hardness result in the center-dominated regime. Namely, when trainable deviations are small, an asymmetric two-layer ReLU network remains close to its random feature baseline and cannot approximate certain targets well, via a reduction to hardness results for random feature models (Yehudai & Shamir, 2019).

## 5. Linear Mode Connectivity

In this section, we analyze how intermediate representations evolve when we linearly interpolate between two independently trained models, depending on effective symmetry breaking. LMC asks whether the loss stays small along this segment. A natural sufficient condition is that the network's

hidden representations vary approximately affinely along the interpolation path, and recent work has observed that this co-occurs with LMC (Zhou et al., 2023b). We therefore quantify the deviation induced by a single layer along the path, and relate it to an upper bound on the loss barrier. Given $\boldsymbol{W}^A, \boldsymbol{W}^B \in \mathbb{R}^{m \times d}$, we consider the linear path

$$\boldsymbol{W}(\lambda) := (1-\lambda)\boldsymbol{W}^A + \lambda\boldsymbol{W}^B, \; \lambda \in [0,1]. \quad (29)$$

We measure the chord deviation along $\boldsymbol{W}(\lambda)$ by

$$\xi_{\boldsymbol{H}}(\lambda; \boldsymbol{x}) := \quad (30)$$
$$\boldsymbol{H}(\boldsymbol{W}(\lambda); \boldsymbol{x}) - \big((1-\lambda)\boldsymbol{H}(\boldsymbol{W}^A; \boldsymbol{x}) + \lambda\boldsymbol{H}(\boldsymbol{W}^B; \boldsymbol{x})\big).$$

Equivalently, $\xi_{\boldsymbol{H}}(\lambda; \boldsymbol{x})$ is the interpolation error of the map $\boldsymbol{W} \mapsto \boldsymbol{H}(\boldsymbol{W}; \boldsymbol{x})$ when restricted to the line segment between $\boldsymbol{W}^A$ and $\boldsymbol{W}^B$. For $\eta := \text{ReLU}$, the only curvature comes from the kink at zero.

**Theorem 5.1** (Chord deviation in center-dominated regime, ReLU). *Let $\eta = \text{ReLU}$ and $\boldsymbol{x} \sim \mathcal{N}(0, \boldsymbol{\Sigma})$ with $\boldsymbol{\Sigma} \succeq 0$. Fix $\boldsymbol{W}^A, \boldsymbol{W}^B \in \mathbb{R}^{m \times d}$. Suppose there exists $\beta \in [0,1)$ such that for all $i \in [m]$,*

$$\|\mathbf{d}_i \odot \boldsymbol{w}^A_i\|_{\boldsymbol{\Sigma}} \leq \beta \|\mathbf{f}_i\|_{\boldsymbol{\Sigma}}, \;\; \|\mathbf{d}_i \odot \boldsymbol{w}^B_i\|_{\boldsymbol{\Sigma}} \leq \beta \|\mathbf{f}_i\|_{\boldsymbol{\Sigma}}, \quad (31)$$

*where $\|u\|_{\boldsymbol{\Sigma}} := \sqrt{u^\top \boldsymbol{\Sigma} u}$. Then,*

$$\sup_{\lambda \in [0,1]} \|\xi_{\boldsymbol{H}}(\lambda; \cdot)\|_{L^2(\mathbb{P};\mathbb{R}^m)} = \mathcal{O}(\beta^{3/2}) \|\mathbf{F}\boldsymbol{\Sigma}^{1/2}\|_F. \quad (32)$$

See § D.1 for a proof. Thm. 5.1 gives a second-order view on layer-wise merging along the linear path $\boldsymbol{W}(\lambda)$. It can be interpreted threefold. First, the chord deviation depends on the gating stability. Second, the bound is uniform over $\lambda \in [0,1]$. This aligns with the analysis in § 4 that once every neuron is center-dominated at the endpoints, the entire segment inherits the same dominance by convexity of the norm. Third, $\|\mathbf{F}\boldsymbol{\Sigma}^{1/2}\|_F$ corresponds to the total norm of the fixed part under the input covariance. The prefactor tends to zero as $\beta \to 0$. See § D for further discussion and a generalization of Thm. 5.1 to a broader class of activations. With control over chord deviations for outputs (e.g., logits), one can upper bound the loss along the linear path between two independently trained models, and guarantee a small loss barrier along the segment, i.e., the presence of LMC. We demonstrate this in the following proposition, whose proof can be found in § D.1.

**Proposition 5.2** (LMC bound by chord deviation). *Consider a dataset $\mathcal{X} \times \mathcal{Y} \ni (\boldsymbol{x}, \boldsymbol{y}) \sim \mathbb{P}$, a neural network $f_{\boldsymbol{\theta}} : \mathcal{X} \to \mathcal{Y} \subseteq \mathbb{R}^{d_{\text{out}}}$, and define the population loss $\mathcal{L}(\boldsymbol{\theta}) := \mathbb{E}_{(\boldsymbol{x},\boldsymbol{y})\sim\mathbb{P}}[\ell(f_{\boldsymbol{\theta}}(\boldsymbol{x}), \boldsymbol{y})] < \infty$. Assume that for every $\boldsymbol{y}$, $\boldsymbol{z} \mapsto \ell(\boldsymbol{z}, \boldsymbol{y})$ is convex and $L_\ell$-Lipschitz w.r.t. $\|\cdot\|_2$. Fix two parameter vectors $\boldsymbol{\theta}^A, \boldsymbol{\theta}^B$ and define the linear path $\boldsymbol{\theta}(\lambda) := (1-\lambda)\boldsymbol{\theta}^A + \lambda\boldsymbol{\theta}^B, \; \lambda \in [0,1]$. Define further*

$$\xi_f(\lambda; \boldsymbol{x}) := f_{\boldsymbol{\theta}(\lambda)}(\boldsymbol{x}) - \big((1-\lambda)f_{\boldsymbol{\theta}^A}(\boldsymbol{x}) + \lambda f_{\boldsymbol{\theta}^B}(\boldsymbol{x})\big). \quad (33)$$

*Then, for all $\lambda \in [0,1]$, we obtain*

$$\mathcal{L}(\boldsymbol{\theta}(\lambda)) - \big((1-\lambda)\mathcal{L}(\boldsymbol{\theta}^A) + \lambda\mathcal{L}(\boldsymbol{\theta}^B)\big)$$
$$\leq L_\ell \|\xi_f(\lambda; \cdot)\|_{L^2(\mathbb{P}_{\boldsymbol{x}}, \mathbb{R}^{d_{\text{out}}})}. \quad (34)$$

---

[6]Note that unlike in the neuron reassignment case, this expression need not coincide with the minimum over all $\text{id} \neq \tau \in S_d$.

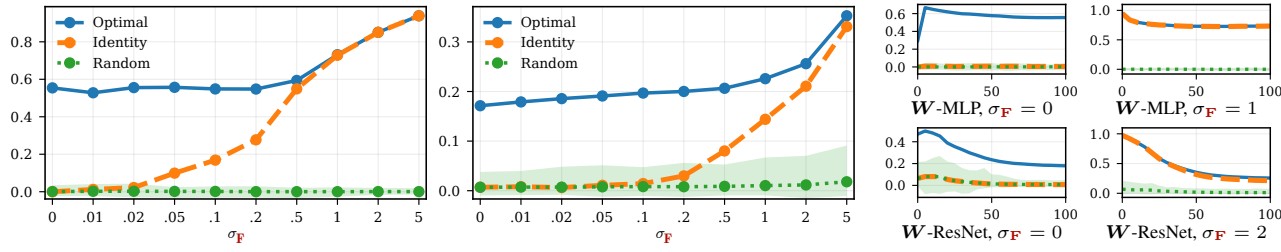

*(a) $\boldsymbol{W}$-MLP, different $\sigma_{\mathbf{F}}$ at epoch 100*     *(b) $\boldsymbol{W}$-ResNet, different $\sigma_{\mathbf{F}}$ at epoch 100*     *(c) $\boldsymbol{W}$-asym. models over different epochs*

*Figure 3.* Activation matching objectives of optimal, identity, and random permutations for networks trained with different values of the fixed weight scale $\sigma_{\mathbf{F}}$. We average the objectives of different layers and use post-norm, post-activation function values.

## 6. Experimental Results

In this section, we empirically investigate the effects that our theory predicts. We check which variables influence the effectiveness of symmetry breaking: in particular, we verify that it significantly depends on properties of the data, and that the structural breaking of weight space symmetries alone is not sufficient to obtain linear mode connectivity.

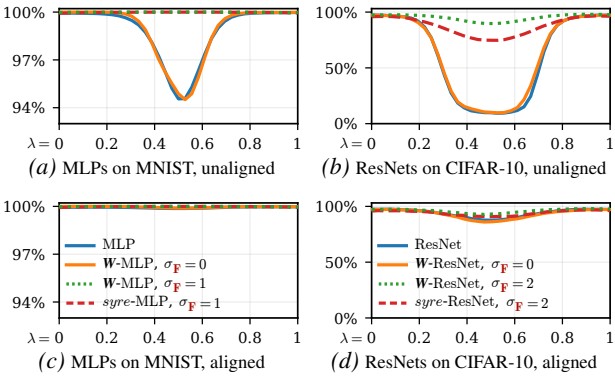

*Figure 4.* Aligned and unaligned training accuracy LMC interpolation for standard models and asymmetric models (average of 8 model pairs each). Alignment using activation matching.

**Q1:** *How do existing symmetry breaking architectures perform in terms of permutation-aligned and unaligned LMC?*

We train standard and asymmetric models on MNIST and CIFAR-10: MLPs, $\boldsymbol{W}$-MLPs (Lim et al., 2024b), and *syre*-MLPs (Ziyin et al., 2025) on MNIST, and ResNets, $\boldsymbol{W}$-ResNets, and *syre*-ResNets on CIFAR-10. For the $\boldsymbol{W}$-asymmetric models, we respectively train versions where $\sigma_{\mathbf{F}} = 0$ (zero fixed weights) vs. $\sigma_{\mathbf{F}} = 1$ for MLPs/$\sigma_{\mathbf{F}} = 2$ for ResNets (center-dominated regime), following the defaults of Lim et al. (2024b). We measure LMC barriers both for *unaligned* LMC (linear interpolation between trained parameters $\boldsymbol{\theta}_A, \boldsymbol{\theta}_B$), as well as *aligned* LMC, interpolating between $\boldsymbol{\theta}_A$ and $\pi\boldsymbol{\theta}_B$ for an alignment permutation $\pi$. [7]

---

[7] Permuting an asymmetric model's weights while preserving functionality would require permuting $\mathbf{F}$. We can nevertheless define alignment by interpreting the effective weights $\boldsymbol{W}_{\text{eff}} = \mathbf{F} + \mathbf{D} \odot \boldsymbol{W}$ as the weights of a standard model.

We obtain $\pi$ via *activation matching* (Singh & Jaggi, 2020; Ainsworth et al., 2023), see **Q2** for details.

Fig. 4 shows the aligned and unaligned LMC results for all architecture variants. The results show that unaligned LMC is achieved for large $\sigma_{\mathbf{F}}$ both in MLPs, and to some degree in ResNets, while the LMC barrier is high in standard MLPs and $\boldsymbol{W}$-MLPs with $\sigma_{\mathbf{F}} = 0$. This supports our findings that structural symmetry breaking does not equal effective symmetry breaking. All models exhibit reasonably low LMC barriers in the aligned case; perhaps surprisingly, alignment reduces the barrier for *syre*-ResNets despite their architectural asymmetry. Furthermore, alignment reduces the LMC barrier of $\boldsymbol{W}$-asym. architectures with $\sigma_{\mathbf{F}} = 0$ similarly well as that of standard architectures. In § G.4 in the appendix, we also compare to a $\boldsymbol{W}$-asymmetric vision transformer variant.

**Q2:** *How large must the fixed weight scale $\sigma_{\mathbf{F}}$ be to make neurons identifiable across independent training runs?*

We answer this by using the activation matching objective (Ainsworth et al., 2023; Singh & Jaggi, 2020). Given $d$-dimensional hidden network activation matrices $\boldsymbol{Z}^A, \boldsymbol{Z}^B \in \mathbb{R}^{n \times d}$ on $n$ data points, produced by two independently trained networks, define a similarity matrix between neurons $\boldsymbol{S}^{\text{act}} \in \mathbb{R}^{d \times d}$, where $\boldsymbol{S}^{\text{act}}_{ij}$ is the Pearson correlation coefficient between the activations of neurons $i$ and $j$. Activation matching finds a permutation matrix $\boldsymbol{P}^*$ that maximizes the Frobenius inner product $\mathcal{L}^{\text{act}}(\boldsymbol{P}) = \langle \boldsymbol{P}, \boldsymbol{S}^{\text{act}} \rangle_F$. In a setting where symmetries are effectively broken on the data, we would expect $\mathcal{L}^{\text{act}}(\boldsymbol{P}^*) \approx \mathcal{L}^{\text{act}}(\boldsymbol{I})$: the identity permutation is nearly optimal. By the results of § 4.3 and § 4.4, we would expect this to be the case for $\sigma_{\mathbf{F}}$ large enough. In Figs. 3a and 3b, we sweep over different values of $\sigma_{\mathbf{F}}$ and compute $\mathcal{L}^{\text{act}}(\boldsymbol{P}^*)$ and $\mathcal{L}^{\text{act}}(\boldsymbol{I})$ and compare them with $\mathcal{L}^{\text{act}}(\tilde{\boldsymbol{P}})$ for random permutations $\tilde{\boldsymbol{P}}$ on pairs of trained networks on MNIST and CIFAR-10. Both $\boldsymbol{W}$-MLPs and $\boldsymbol{W}$-ResNets show that for large enough $\sigma_{\mathbf{F}}$, $\mathcal{L}^{\text{act}}(\boldsymbol{I})$ becomes close to $\mathcal{L}^{\text{act}}(\boldsymbol{P}^*)$, while random permutations remain significantly lower. Fig. 3c shows that this trend holds for all stages of training. In § G.1 in the appendix, we give detailed per-layer breakdowns of the activation matching objective.

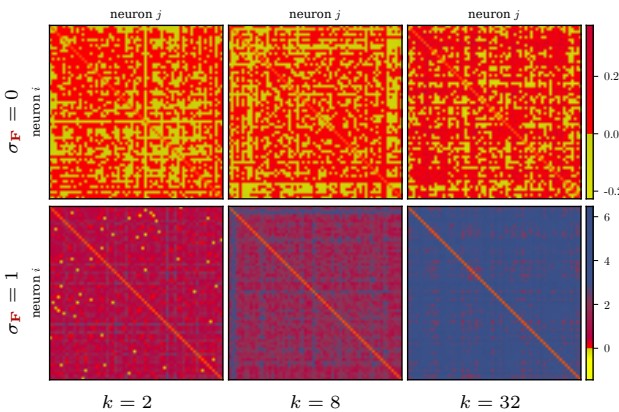

Figure 5. Neuron swap costs $\Delta_{(ij)}^{\mathrm{out}}$ (signed square-root transformed) in synthetic experiments with $\boldsymbol{W}$-MLPs on GMM data with varying intrinsic dim. $k$, showing large/small/negative costs.

**Q3:** *Can we empirically observe the dependence of neuron swap costs $\Delta_{(ij)}^{\mathrm{out}}$ on the fixed weight scale $\sigma_{\mathbf{F}}$, and how does the intrinsic data dimension $k$ influence the swap cost of neuron pairs in trained models? (see Thm. 4.5, Thm. 4.6)*

Our theory in § 4.1 defines neuron realization costs (8), reduces them to a Mahalanobis distance under the subspace support model (10), and uses them to define the realization cost sensitivity $\Delta_{\pi,\tau}(\boldsymbol{H};\mathcal{X})$ in (14). We now estimate realization costs in trained models via (10), using a subspace basis $\boldsymbol{U}$ estimated as the PCA subspace at 90% explained variance. We compute the output realization cost sensitivities for transpositions $\pi = (ij)$, i.e., *neuron swap costs* $\Delta_{(ij)}^{\mathrm{out}}$. We empirically compare to another method to estimate realization cost, via a ridge regression objective (§ E.1), in the appendix (§ G.3).

Thm. 4.5 shows that $\Delta_{\pi}^{\mathrm{out}}$ is approximately proportional to the squared min. neuron center gap $\gamma_{\mathrm{out}}^2$ in the center-dominated regime, and Thm. 4.6 predicts that $\gamma_{\mathrm{out}}$ depends on the intrinsic data dimension $k$ and the number of neurons $m$ (also see § G.2). We train $\boldsymbol{W}$-MLPs with $m = 64$ on a Gaussian mixture dataset, where $k$ can be controlled, and report the measured neuron swap costs for $k \in \{2, 8, 32\}$ and $\sigma_{\mathbf{F}} \in \{0, 1\}$ in Fig. 5. It shows that *(a)*, near zero and even negative pairwise neuron swap costs dominate in the non-center-dominated regime ($\sigma_{\mathbf{F}} = 0$); *(b)* that in the center-dominated regime ($\sigma_{\mathbf{F}} = 1$), there is a strong dependence on the intrinsic dimension $k$: for $k = 2$, many low-cost swaps exist, while only the diagonal remains at low cost for $k = 32$. The experiment demonstrates that for low $k$ and high-enough $m$, neuron swaps again become accessible at lower weight cost despite the symmetry breaking. Unaligned LMC performance in the $\sigma_{\mathbf{F}} = 1$ setting corroborates these results: Over two model pairs each, the train accuracy between endpoints (avg.) and midpoint drops by 46.4 percentage points for $k = 2$, by 15.7 percentage points for $k = 8$, and only by 6.1 percentage points for $k = 32$.

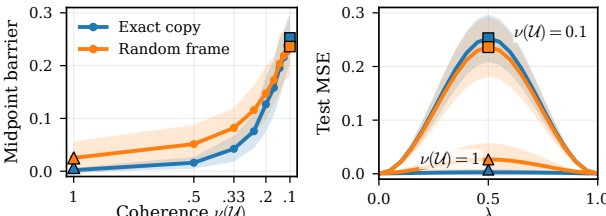

Figure 6. LMC dependence on coherence $\nu(\mathcal{U})$ in $\mathbf{F} = 0$ setting: LMC barriers on a synthetic dataset with varying $\nu(\mathcal{U})$.

**Q4:** *How does the input subspace coherence $\nu(\mathcal{U})$ control the effectiveness of symmetry breaking?*

Thm. 4.4 suggests that on highly coherent data, (i.e., when the data's principal directions are aligned well with the standard basis axes and hence $\nu(\mathcal{U}) \approx 1$), the Gram matrix $\mathbf{S}_i$ can become more anisotropic. In this case, the diagonal operator $\mathbf{D}$ may have a greater influence than observed so far by making certain directions of data variation more expensive to read out than others for some neurons, and hence enable effective symmetry breaking even in the non-center-dominated setting ($\mathbf{F} = 0$).

We consider an intrinsically two-dimensional dataset, where $a, b \sim \mathcal{N}(0, 1)$ are embedded into a 20-dim. ambient space by copying each value $a, b$ to $t \leq 10$ positions, filling up the rest with zeros. Then $\nu(\mathcal{U}) = \frac{1}{t}$ can be varied from minimal ($\frac{1}{10}$) to maximal (1) by varying $t$. We train a 1-hidden layer $\boldsymbol{W}$-MLP ($\mathbf{F} = 0, \mathbf{D} \sim \mathrm{Bernoulli}(0.15)$) to fit the target $\mathrm{ReLU}(a + b)$. The results in Fig. 6 show LMC for high coherence, but high midpoint barrier for low coherence, and the same trend for a variant where the data is randomly rotated inside the first $2t$ coordinates ("random frame").

## 7. Conclusion

In this work, we studied under which conditions parameter symmetry breaking yields neuron identifiability, i.e., consistent assignment of features to neurons across independent training runs, and showed that approximate and data-dependent symmetries play a significant role even when structural symmetries are removed. This clarifies that even when symmetries in raw parameter space are broken, symmetry breaking may remain practically ineffective. Further, we show that identifiability enables representation merging and can yield sufficient conditions for unaligned linear mode connectivity.

Future directions include using neuron identifiability as a tool beyond symmetry breaking, such as for more reliable neuron-level interpretability. Furthermore, it remains to extend our theory to richer symmetry groups, analyze exact training dynamics, and to develop more quantitative diagnostics that predict when neuron identifiability and unaligned LMC should be expected for a given training setup.

## Acknowledgements

We thank the anonymous reviewers for their insightful feedback, and Valerie Engelmayer for providing feedback on an earlier version of this manuscript. VB acknowledges support from a PhD fellowship of the Munich Center for Machine Learning (MCML). DH acknowledges support from the Munich Data Science Institute (MDSI) via the MDSI Doctoral Fellowship. YEL acknowledges support from the Schmidt Futures Israeli Women's Postdoctoral Award. This research was supported by an Alexander von Humboldt Professorship.

## Impact Statement

This paper presents work whose goal is to advance the field of Machine Learning. There are many potential societal consequences of our work, none of which we feel must be specifically highlighted here.

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

# Appendices

# A. Notation

*Table 2.* We list the most important symbols used throughout this work.

| | |
|---|---|
| $\mathbb{N}, \mathbb{N}_0, \mathbb{Q}, \mathbb{R}$ | Natural, non-negative integer, rational, real numbers. |
| $[n]$ | Set $\{1, \ldots, n\}$ for $n \in \mathbb{N}$. |
| $\mathbb{1}_A$ | Indicator function in a set $A$. |
| $\mathcal{O}(\cdot), \Theta(\cdot), \Omega(\cdot)$ | Landau symbols for asymptotic growth of a function. |
| $S_n$ | Symmetric group $\{\pi : [n] \to [n] \mid \pi \text{ bijective}\}$. |
| id, $(ij), \pi$ | Identity, transposition $i \leftrightarrow j$, generic permutation $\in S_n$. |
| $\boldsymbol{P}, \boldsymbol{P}_\pi \in \{0,1\}^{n \times n}$ | Generic permutation matrix, permutation matrix of $\pi \in S_n$. |
| $\odot$ | Elementwise multiplication of matrices. |
| $\boldsymbol{I}_n$ | Identity matrix in $n$ dimensions. |
| $\mathbf{1}_n$ | Vector of all ones in $n$ dimensions. |
| $\boldsymbol{e}_i$ | $i$-th canonical basis vector. |
| $\mathrm{Unif}(\cdot)$ | Uniform distribution on a set. |
| $\mathrm{Bernoulli}(p)$ | Bernoulli distribution with success probability $p$. |
| $\mathcal{N}(\mu, \sigma^2)$ | Normal distribution with mean $\mu$, variance $\sigma^2$. |
| $\mathcal{N}(\boldsymbol{\mu}, \boldsymbol{\Sigma})$ | Multivariate normal distribution with mean vector $\boldsymbol{\mu}$, covariance matrix $\boldsymbol{\Sigma}$. |
| $\|\cdot\|_p, \|\cdot\|_{\mathrm{op}}, \|\cdot\|_F, \|\cdot\|_{\boldsymbol{A}}$ | $\ell^p$ norms ($p \in [1, \infty]$), spectral norm, Frobenius norm, Mahalanobis norm w.r.t. $\boldsymbol{A}$. |
| $\boldsymbol{A}^\dagger$ | Moore-Penrose pseudoinverse of $\boldsymbol{A}$. |
| $\succ, \succeq, \prec, \preceq$ | Löwner partial order for matrices. |
| $L^p(\mu)$ | $p$-integrable functions w.r.t. a measure $\mu$, $p \in [1, \infty]$. |
| $\|\cdot\|_{L^p(\mu)}, \|\cdot\|_{L^p(\mu, \mathbb{R}^m)}$ | $L^p$ norms of functions mapping to $\mathbb{R}, \mathbb{R}^m$, $p \in [1, \infty]$. |
| $f * g$ | Convolution of two functions $f, g$. |
| $\lambda^n$ | $n$-dimensional Lebesgue measure. |
| $B_t^n$ | $n$-dimensional ball in $\mathbb{R}^n$, i.e., $\{\boldsymbol{x} \in \mathbb{R}^n \mid \|\boldsymbol{x}\|_2 < t\}$. |
| $V_n$ | Volume $\lambda^n(B_1^n)$ of the $n$-dimensional unit ball. |
| $\|\cdot\|_{\psi_2}, \|\cdot\|_{\psi_1}$ | Subgaussian and subexponential norms. |
| $\mathbb{S}^{n-1}$ | $n$-dimensional sphere $\{\boldsymbol{x} \in \mathbb{R}^n \mid \|\boldsymbol{x}\|_2 = 1\}$. |
| $\boldsymbol{\theta}, \Theta$ | (Total) parameter vector and parameter space of a neural architecture. |
| $\boldsymbol{W}, \boldsymbol{w}$ | Weight matrix of a layer, weight vector of a single neuron. |
| $d, m$ | Ambient input dimension, output dimension (i.e., width) of a layer. |
| $\eta$ | Activation function. |
| $\mathbf{F}, \mathbf{D} \in \mathbb{R}^{m \times d}$ | Fixed offset and diagonal operator of symmetry breaking intervention. |
| $\mathbf{f}_i, \mathbf{d}_i, \boldsymbol{w}_i \in \mathbb{R}^d$ | $i$-th rows of $\mathbf{F}, \mathbf{D}, \boldsymbol{W}$, $i \in [m]$. |
| $\sigma_{\mathbf{F}}$ | Standard deviation (fixed weight scale) of entry distribution of $\mathbf{F}$. |
| $\mu_{\mathbf{D}}, \sigma_{\mathbf{D}}, b_{\mathbf{D}}$ | Mean, standard deviation, a.s. deviation of entry distribution of $\mathbf{d}_i^2$. |
| $\boldsymbol{x}, \mathcal{X}, \mathbb{P}$ | Input, input support, input distribution of a layer ($\mathcal{X} = \mathrm{supp}(\mathbb{P})$). |
| $\boldsymbol{H}(\boldsymbol{W}, \boldsymbol{x}), \boldsymbol{h}_i(\boldsymbol{w}_i, \boldsymbol{x})$ | Layer map of a symmetry-broken layer, single neuron $i$. |
| $\mathcal{H}, \mathcal{H}_i, \mathcal{H}_i(\mathcal{X})$ | Fct. class of a layer, fct. class of neuron $i$, fct. class of neuron $i$ w/ explicit input supp. $\mathcal{X}$. |
| $k$ | Intrinsic input dimension $\leq d$. |
| $\mathcal{U}, \boldsymbol{U}$ | Input subspace, orthonormal basis of $\mathcal{U}$. |
| $\mathbf{v}_i \in \mathbb{R}^k, \mathbf{M}_i \in \mathbb{R}^{k \times d}$ | $\mathbf{F}_i, \mathrm{Diag}(\mathbf{d}_i)$ in projected $\boldsymbol{U}$ coordinates. |
| $\mathbf{S}_i \in \mathbb{R}^{k \times k}$ | Gram matrix $\mathbf{S}_i = \mathbf{M}_i \mathbf{M}_i^\top$. |
| $\|\boldsymbol{h}\|_{\mathcal{H}_i(\mathcal{X})}$ | Realization cost of $\boldsymbol{h} : \mathcal{X} \to \mathbb{R}$ w.r.t. neuron $i$ on input support $\mathcal{X}$. |
| $\nu(\mathcal{U})$ | Subspace coherence. |
| $\Delta_{\pi, \tau}(\boldsymbol{H}; \mathcal{X})$ | Realization cost sensitivity of $\boldsymbol{H} \in \mathcal{H}(\mathcal{X})$ under output-input permutation pair $(\pi, \tau)$. |
| $\boldsymbol{\delta}_\pi^{\mathrm{out}}, \boldsymbol{\delta}_\tau^{\mathrm{in}} \in \mathbb{R}^{m \times k}$ | Projected center differences under output / input permutation $\pi$ / $\tau$. |
| $\gamma_{\mathrm{out}}, \gamma_{\mathrm{in}}$ | Minimum proj. center distances for output / input. |
| $\xi_{\boldsymbol{H}}(\lambda; \boldsymbol{x})$ | Chord deviation of layer $\boldsymbol{H}$ at interpolation parameter $\lambda \in [0,1]$. |
| $\ell, \mathcal{L}$ | Loss on one data point, population loss. |

# B. Extended Related Work

## B.1. Parameter Symmetries and Mode Connectivity

Mode connectivity of neural networks was introduced by early works which observed that trained neural network solutions can be connected by paths in parameter space where the performance (measured by training or validation loss or accuracy) does not significantly worsen (Garipov et al., 2018; Draxler et al., 2018). The focus soon shifted to linear mode connectivity (LMC) (Frankle et al., 2020), which has been widely interpreted to occur when two solutions are in the same basin of the optimization landscape, which is a desirable property for applications such as model merging. Several settings which admit LMC have been identified: Frankle et al. (2020) found LMC for networks trained with different optimization randomness but starting from the same partially converged solution. Later studies obtained LMC by aligning networks with respect to symmetries (Ainsworth et al., 2023), or removing symmetries (Lim et al., 2024b). Variants of LMC focusing on individual layers have been investigated (Adilova et al., 2024; Zhou et al., 2023b).

Exact structural neural network parameter symmetries, in particular *permutation symmetries*, have been investigated as early as Hecht-Nielsen (1990); Chen et al. (1993), where it was observed that the whole parameter space can be mapped to a small cone via function-preserving permutations. Recent work has also studied hidden, local, and data-dependent symmetries, where equivalences may depend on the activation region, parameter point, or data distribution rather than preserving the realized function globally (Zhao et al., 2023; Grigsby et al., 2023; Xie & Smidt, 2025; Zhao et al., 2026).

Entezari et al. (2022) connected parameter symmetries to LMC and stated the conjecture that pairs of SGD solutions are likely linearly mode connected after applying a suitable permutation. A number of works have proposed permutation alignment algorithms, with the goal of obtaining interpolating paths of low barrier, based either on optimal transport (OT) of activations or weights (*activation* or *weight matching*), or directly optimizing for a permutation that decreases the midpoint barrier (*straight-through estimator* or *learned matching*) (Singh & Jaggi, 2020; Tatro et al., 2020; Pittorino et al., 2022; Ainsworth et al., 2023), alongside non-OT variants (Li et al., 2015; Crisostomi et al., 2024). Alignment methods have been primarily applied to MLPs (permuting neurons) and convolutional architectures such as ResNets (permuting channels), but have recently been successfully applied to Transformers (Imfeld et al., 2024; Verma & Elbayad, 2024; Theus et al., 2025), both for vision and language (where vision task models seem to be easier to align). Theoretical results in Ferbach et al. (2024) show aligned LMC can be obtained for trained wide one-hidden-layer neural networks in the mean-field regime.

There has also been interest in *unaligned LMC*, i.e., conditions under which LMC is given without post-hoc alignment. Recently, Lim et al. (2024b) proposed architectural symmetry breaking methods and obtained unaligned LMC with $W$-*asymmetric models*, where parts of the weights of each neuron are fixed in a way that breaks the usual permutation symmetries. Not primarily motivated by LMC but in a similar fashion, Ziyin et al. (2025) add fixed noise to parameters. Ito et al. (2026) find that increasingly wide models tend to yield LMC without symmetry alignment or removal, and explain this by wide layers exhibiting reciprocally orthogonal null spaces, so that merged models increasingly behave like an ensemble.

## B.2. Fixed Masks, Pruning, and Sketching

When $\mathbf{d}_i \in \{0,1\}^d$, the trainable part $\mathbf{d}_i \odot \boldsymbol{w}_i$ corresponds to a fixed sparse connectivity pattern, which is similar to a pruning-at-initialization or fixed DropConnect-style mask on the incoming weights (Wan et al., 2013). In contrast, standard dropout resamples activation masks during training (Srivastava et al., 2014). Dropout and related feature-noising schemes have been analyzed as induced regularizers (Wager et al., 2013) and, under variational assumptions, as approximate Bayesian inference (Gal & Ghahramani, 2016). The masks in this work, however, are sampled once and kept fixed across the independent training runs.

Prior work on pruning has studied post-training sparsification, lottery-ticket subnetworks (Frankle & Carbin, 2019), dynamic sparse training (Evci et al., 2020), and pruning at initialization (Lee et al., 2019; Wang et al., 2020; Tanaka et al., 2020; Frankle et al., 2021). Recent work has also connected pruning at initialization to randomized sketching (Bar & Giryes, 2025). Our use of this viewpoint is different: we do not use masks primarily to preserve an approximation of a dense network, but to understand when fixed diagonal operators make neurons functionally distinguishable on the input support. The relevant quantity for this distinction is the subspace coherence $\nu(\mathcal{U})$, equivalently the maximum leverage score of the input subspace (Drineas et al., 2012). Coherence is standard in matrix completion, compressed sensing, and randomized numerical linear algebra, where it controls whether uniform coordinate sampling preserves a low-dimensional structure (Candes & Tao, 2005; Candes & Recht, 2009). In our setting, the same quantity controls the concentration of $\mathbf{S}_i = \boldsymbol{U}^\top \mathrm{Diag}(\mathbf{d}_i)^2 \boldsymbol{U}$. I.e., we use coherence not for exact recovery, but to quantify when fixed masks are visible to the effective function classes of neurons.

# C. Neuron Identifiability (§ 4)

## C.1. Proofs from § 4.1–§ 4.2 (Realization Costs)

**Lemma C.1** (Neuron identifiability from preactivations)**.** *Under Assumptions 3.1 and 4.1, any $h_i$, $i \in [m]$, can be identified from its preactivation. Formally,*

$$h_i(w; \cdot) = h_i(w'; \cdot) \Leftrightarrow U^\top \mathrm{Diag}(\mathbf{d}_i)(w - w') = 0, \tag{35}$$

*where the l.h.s. is understood almost surely w.r.t. $\mathbb{P}$.*

*Proof.* Fix $i \in [m]$, $w, w' \in \mathbb{R}^d$, and write the (row-wise) preactivation vectors

$$w_{\mathrm{eff}} := \mathbf{f}_i + \mathbf{d}_i \odot w = \mathbf{f}_i + \mathrm{Diag}(\mathbf{d}_i)w, \qquad w'_{\mathrm{eff}} := \mathbf{f}_i + \mathrm{Diag}(\mathbf{d}_i)w'. \tag{36}$$

By Assumption 3.1, every input $x$ satisfies $x \in \mathcal{U} = \mathrm{im}(U)$, hence we can write $x = Uc$ for some $c \in \mathbb{R}^k$. Define the subspace coefficients $a := U^\top w_{\mathrm{eff}} \in \mathbb{R}^k$ and $a' := U^\top w'_{\mathrm{eff}} \in \mathbb{R}^k$. For any $x = Uc \in \mathcal{U}$, we have the identity

$$w_{\mathrm{eff}}^\top x = w_{\mathrm{eff}}^\top Uc = (U^\top w_{\mathrm{eff}})^\top c = a^\top c, \tag{37}$$

and likewise $(w'_{\mathrm{eff}})^\top x = a'^\top c$. Therefore, the neuron functions restricted to $\mathcal{U}$ can be written as

$$h_i(w; Uc) = \eta(a^\top c), \qquad h_i(w'; Uc) = \eta(a'^\top c). \tag{38}$$

Note that

$$a - a' = U^\top (w_{\mathrm{eff}} - w'_{\mathrm{eff}}) = U^\top \mathrm{Diag}(\mathbf{d}_i)(w - w'). \tag{39}$$

We can now prove the equivalence.

"$\Leftarrow$": If $U^\top \mathrm{Diag}(\mathbf{d}_i)(w - w') = 0$, then (39) and (38) directly yield $h_i(w; \cdot) = h_i(w'; \cdot)$ on $\mathcal{U}$.

"$\Rightarrow$": Assume $h_i(w; \cdot) = h_i(w'; \cdot)$ on $\mathrm{supp}(\mathbb{P}) \subseteq \mathcal{U}$ (by Assumption 3.1). Equivalently,

$$\eta(a^\top c) = \eta(a'^\top c) \qquad \text{for all } c \in U^\top(\mathrm{supp}(\mathbb{P})) \subseteq \mathbb{R}^k. \tag{40}$$

We show this forces $a = a'$ under Assumption 4.1, distinguishing the two activation cases.

**Case ①: $\eta$ injective.** Injectivity gives $a^\top c = a'^\top c$ for all $c \in U^\top(\mathrm{supp}(\mathbb{P}))$, hence $(a - a')^\top c = 0$ for all such $c$. By Assumption 4.1, we have $\mathrm{span}(\mathrm{supp}(\mathbb{P})) = \mathcal{U}$, hence $\mathrm{span}(U^\top \mathrm{supp}(\mathbb{P})) = \mathbb{R}^k$. Therefore, $(a - a')^\top c = 0$ on a spanning set of $\mathbb{R}^k$, which implies $a - a' = U^\top \mathrm{Diag}(\mathbf{d}_i)(w - w') = 0$.

**Case ②: $s \mapsto (\eta(s), \eta(-s))$ injective.** Let $T \subseteq \mathrm{supp}(\mathbb{P})$ be as in Assumption 4.1, so $T = -T$ and $\mathrm{span}(T) = \mathcal{U}$. Set $T_U := U^\top T \subseteq \mathbb{R}^k$. Then $T_U = -T_U$ and $\mathrm{span}(T_U) = \mathbb{R}^k$. Since $\eta(a^\top c) = \eta(a'^\top c)$ for all $c \in U^\top(\mathrm{supp}(\mathbb{P}))$, in particular $\eta(a^\top c) = \eta(a'^\top c)$ for all $c \in T_U$. Moreover, for any $c \in T_U$ we have $-c \in T_U$, hence also $\eta(-a^\top c) = \eta(-a'^\top c)$. Therefore, for all $c \in T_U$, $(\eta(a^\top c), \eta(-a^\top c)) = (\eta(a'^\top c), \eta(-a'^\top c))$, and injectivity yields $a^\top c = a'^\top c$ for all $c \in T_U$, i.e. $(a - a')^\top c = 0$ for all $c \in T_U$. Since $\mathrm{span}(T_U) = \mathbb{R}^k$, we get $a = a'$, which concludes the proof. $\square$

**Proposition 4.2.** *Define $\mathbf{v}_i$ as the projection of the fixed center of the $i$-th asymmetric neuron to the input subspace, and $\mathbf{M}_i$ as its projected diagonal operator, i.e.,*

$$\mathbf{v}_i := U^\top \mathbf{f}_i \in \mathbb{R}^k, \quad \mathbf{M}_i := U^\top \mathrm{Diag}(\mathbf{d}_i) \in \mathbb{R}^{k \times d}. \tag{5}$$

*Under Assumptions 3.1 and 4.1, $h \in \mathcal{H}_i(\mathcal{X})$ belonging to weight $w$ can be written as $h(x) = \eta((Ua)^\top x)$, where*

$$a := U^\top (\mathbf{f}_i + \mathrm{Diag}(\mathbf{d}_i)w) = \mathbf{v}_i + \mathbf{M}_i w \in \mathbb{R}^k. \tag{6}$$

*The mapping $h \mapsto a$ is a bijection that identifies $\mathcal{H}_i(\mathcal{X})$ with $\mathbf{v}_i + \mathrm{im}(\mathbf{M}_i) \subseteq \mathbb{R}^k$. In particular, $\mathcal{H}_i(\mathcal{X})$ is an affine subspace of $\mathbb{R}^k$ of dimension $\mathrm{rank}(\mathbf{M}_i) \leq k$.*

*Proof.* Fix $i \in [m]$. Under Assumption 3.1, every $x \in \mathcal{X}$ satisfies $x \in \mathrm{im}(U)$, so we can write $x = Uc$ with $c = U^\top x \in \mathbb{R}^k$. For any $w \in \mathbb{R}^d$, define the induced subspace preactivation coefficient

$$a(w) := U^\top (\mathbf{f}_i + \mathrm{Diag}(\mathbf{d}_i)w) = \mathbf{v}_i + \mathbf{M}_i w \in \mathbb{R}^k. \tag{41}$$

Then for every $x = Uc \in \mathcal{X}$, $h_i(w; x) = \eta((Ua(w))^\top x)$. This yields

$$\mathcal{H}_i(\mathcal{X}) \subseteq \left\{ \eta((Ua)^\top \cdot) \mid a \in \mathbf{v}_i + \mathrm{im}(\mathbf{M}_i) \right\}. \tag{42}$$

Conversely, let $a \in \mathbf{v}_i + \mathrm{im}(\mathbf{M}_i)$. Then there exists $w \in \mathbb{R}^d$ such that $\mathbf{M}_i w = a - \mathbf{v}_i$, i.e., $a(w) = a$ and $h_i(w; \cdot) = \eta((Ua)^\top \cdot)$ on $\mathcal{X}$, so the reverse inclusion in (42) holds as well.

It remains to argue that $\boldsymbol{a} \mapsto \eta((\boldsymbol{U}\boldsymbol{a})^\top \cdot)$ is injective on $\mathbb{R}^k$ under Assumption 4.1. Let $\boldsymbol{a}, \boldsymbol{a}' \in \mathbb{R}^k$ satisfy $\eta((\boldsymbol{U}\boldsymbol{a})^\top \boldsymbol{x}) = \eta((\boldsymbol{U}\boldsymbol{a}')^\top \boldsymbol{x})$ for all $\boldsymbol{x} \in \mathrm{supp}(\mathbb{P})$. Set $\boldsymbol{d} := \boldsymbol{a} - \boldsymbol{a}'$. If $\eta$ is injective, then $(\boldsymbol{U}\boldsymbol{d})^\top \boldsymbol{x} = 0$ for all $\boldsymbol{x} \in \mathrm{supp}(\mathbb{P})$, hence $\boldsymbol{d}^\top (\boldsymbol{U}^\top \boldsymbol{x}) = 0$ for all $\boldsymbol{x} \in \mathrm{supp}(\mathbb{P})$. Since $\mathrm{span}(\mathrm{supp}(\mathbb{P})) = \mathcal{U} = \mathrm{im}(\boldsymbol{U})$, we obtain $\boldsymbol{d} = 0$, i.e. $\boldsymbol{a} = \boldsymbol{a}'$. If $s \mapsto (\eta(s), \eta(-s))$ is injective, then for all $\boldsymbol{x} \in T \subseteq \mathrm{supp}(\mathbb{P})$, with $T = -T$, we have both

$$\eta((\boldsymbol{U}\boldsymbol{a})^\top \boldsymbol{x}) = \eta((\boldsymbol{U}\boldsymbol{a}')^\top \boldsymbol{x}) \quad \text{and} \quad \eta((\boldsymbol{U}\boldsymbol{a})^\top (-\boldsymbol{x})) = \eta((\boldsymbol{U}\boldsymbol{a}')^\top (-\boldsymbol{x})), \tag{43}$$

so the injectivity of $s \mapsto (\eta(s), \eta(-s))$ yields $(\boldsymbol{U}\boldsymbol{a})^\top \boldsymbol{x} = (\boldsymbol{U}\boldsymbol{a}')^\top \boldsymbol{x}$ for all $\boldsymbol{x} \in T$. Since $\mathrm{span}(T) = \mathcal{U}$, the same argument gives $\boldsymbol{a} = \boldsymbol{a}'$.

Having obtained the injectivity of $\boldsymbol{a} \mapsto \eta((\boldsymbol{U}\boldsymbol{a})^\top \cdot)$, the final claim follows immediately. $\square$

**Lemma C.2** (Realization cost in subspace coefficients)**.** *Let* $i \in [m]$ *and* $\boldsymbol{a} \in \mathbb{R}^k$*. Then,*

$$\left\| \eta((\boldsymbol{U}\boldsymbol{a})^\top \cdot) \right\|_{\mathcal{H}_i(\mathcal{X})} = \sqrt{(\boldsymbol{a} - \mathbf{v}_i)^\top \mathbf{S}_i^\dagger (\boldsymbol{a} - \mathbf{v}_i)} \tag{44}$$

$$=: \left\| \boldsymbol{a} - \mathbf{v}_i \right\|_{\mathbf{S}_i^\dagger} \tag{45}$$

*for* $\boldsymbol{a} - \mathbf{v}_i \in \mathrm{im}(\mathbf{S}_i) = \mathrm{im}(\mathbf{M}_i)$ *(and* $+\infty$ *otherwise). Here,* $\mathbf{S}_i^\dagger$ *denotes the Moore-Penrose pseudoinverse of* $\mathbf{S}_i$*.*

*Proof.* Fix $i \in [m]$ and $\boldsymbol{a} \in \mathbb{R}^k$. By Prop. 4.2, we have

$$\eta((\boldsymbol{U}\boldsymbol{a})^\top \cdot) \in \mathcal{H}_i(\mathcal{X}) \quad \Leftrightarrow \quad \boldsymbol{a} - \mathbf{v}_i \in \mathrm{im}(\mathbf{M}_i), \tag{46}$$

and in that case $\eta((\boldsymbol{U}\boldsymbol{a})^\top \cdot) = \boldsymbol{h}_i(\boldsymbol{w}; \cdot)$ holds iff $\mathbf{M}_i \boldsymbol{w} = \boldsymbol{a} - \mathbf{v}_i$. Therefore, by definition of realization cost (8),

$$\left\| \eta((\boldsymbol{U}\boldsymbol{a})^\top \cdot) \right\|_{\mathcal{H}_i(\mathcal{X})} = \inf\{\|\boldsymbol{w}\|_2 \mid \mathbf{M}_i \boldsymbol{w} = \boldsymbol{a} - \mathbf{v}_i\}, \tag{47}$$

with the convention $+\infty$ if $\boldsymbol{a} - \mathbf{v}_i \notin \mathrm{im}(\mathbf{M}_i)$. Assume now that $\boldsymbol{a} - \mathbf{v}_i \in \mathrm{im}(\mathbf{M}_i)$. Recalling $\mathbf{S}_i = \mathbf{M}_i \mathbf{M}_i^\top$, the (unique) minimum $\ell^2$ norm solution is

$$\boldsymbol{w}^\star = \mathbf{M}_i^\top (\mathbf{M}_i \mathbf{M}_i^\top)^\dagger (\boldsymbol{a} - \mathbf{v}_i) = \mathbf{M}_i^\top \mathbf{S}_i^\dagger (\boldsymbol{a} - \mathbf{v}_i), \tag{48}$$

and every solution can be written as $\boldsymbol{w} = \boldsymbol{w}^\star + \boldsymbol{w}_\perp$ with $\boldsymbol{w}_\perp \in \ker(\mathbf{M}_i)$, which implies $\|\boldsymbol{w}\|_2^2 = \|\boldsymbol{w}^\star\|_2^2 + \|\boldsymbol{w}_\perp\|_2^2 \geq \|\boldsymbol{w}^\star\|_2^2$. Hence, $\inf\{\|\boldsymbol{w}\|_2 \mid \mathbf{M}_i \boldsymbol{w} = \boldsymbol{a} - \mathbf{v}_i\} = \|\boldsymbol{w}^\star\|_2$. Moreover,

$$\|\boldsymbol{w}^\star\|_2^2 = (\boldsymbol{a} - \mathbf{v}_i)^\top \mathbf{S}_i^\dagger \mathbf{S}_i \mathbf{S}_i^\dagger (\boldsymbol{a} - \mathbf{v}_i) \tag{49}$$

$$= (\boldsymbol{a} - \mathbf{v}_i)^\top \mathbf{S}_i^\dagger (\boldsymbol{a} - \mathbf{v}_i), \tag{50}$$

where we used $\mathbf{S}_i^\dagger \mathbf{S}_i \mathbf{S}_i^\dagger = \mathbf{S}_i^\dagger$. This yields

$$\left\| \eta((\boldsymbol{U}\boldsymbol{a})^\top \cdot) \right\|_{\mathcal{H}_i(\mathcal{X})} = \sqrt{(\boldsymbol{a} - \mathbf{v}_i)^\top \mathbf{S}_i^\dagger (\boldsymbol{a} - \mathbf{v}_i)}, \tag{51}$$

which holds for $\boldsymbol{a} - \mathbf{v}_i \in \mathrm{im}(\mathbf{S}_i) = \mathrm{im}(\mathbf{M}_i)$ (and equals $+\infty$ otherwise). $\square$

**Theorem 4.4** (Spectral concentration of $\mathbf{S}_i$)**.** *Let* $i \in [m]$ *and sample the entries of* $\mathbf{d}_i$ *i.i.d. with* $(\mathbf{d}_i)_\ell^2$ *having mean* $\mu_{\mathbf{D}}$*, variance* $\sigma_{\mathbf{D}}^2$*, and a.s. deviation* $b_{\mathbf{D}}$ *from its mean (all* $< \infty$*). For any* $\delta \in (0, 1)$*, with probability at least* $1 - \delta$*,*

$$\|\mathbf{S}_i - \mu_{\mathbf{D}} \boldsymbol{I}_k\|_{\mathrm{op}} \leq C \left\{ \sqrt{\sigma_{\mathbf{D}}^2 \nu(\mathcal{U}) \Lambda} + b_{\mathbf{D}} \nu(\mathcal{U}) \Lambda \right\}, \tag{12}$$

*where* $\Lambda := \log(2k/\delta)$*,* $\nu(\mathcal{U}) := \max_{\ell \in [d]} (\boldsymbol{U}\boldsymbol{U}^\top)_{\ell\ell}$ *is the subspace coherence, and* $C$ *is a universal constant.*[8]

The proof of this theorem relies on an application of the matrix Bernstein inequality.

**Lemma C.3** (Matrix Bernstein Inequality; Theorem 1.4 of Tropp (2015))**.** *Let* $\boldsymbol{A}_1, \ldots, \boldsymbol{A}_d \in \mathbb{R}^{k \times k}$ *be independent, centered random symmetric matrices. Assume a uniform bound* $\|\boldsymbol{A}_l\|_{\mathrm{op}} \leq R$ *almost surely for each addend* $l \in [d]$*. Let*

$$\sigma^2 := \left\| \sum_{l=1}^d \mathbb{E}[\boldsymbol{A}_l^2] \right\|_{\mathrm{op}}. \tag{52}$$

*Then, for all* $t \geq 0$*,*

$$\mathbb{P}\left( \left\| \sum_{l=1}^d \boldsymbol{A}_l \right\|_{\mathrm{op}} \geq t \right) \leq 2k \exp\left( \frac{-t^2/2}{\sigma^2 + Rt/3} \right). \tag{53}$$

---

[8]One can also control $\max_i \|\mathbf{S}_i - \mu_{\mathbf{D}} \boldsymbol{I}_k\|_{\mathrm{op}}$ via a union bound at the cost of replacing $\log(2k/\delta)$ by $\log(2mk/\delta)$.

*Proof of Thm. 4.4.* Denote the $l$-th row of $\boldsymbol{U}$ by $\boldsymbol{u}_l \in \mathbb{R}^k$. Observe that

$$(\boldsymbol{U}\boldsymbol{U}^\top)_{ll} = \boldsymbol{e}_l^\top \boldsymbol{U}\boldsymbol{U}^\top \boldsymbol{e}_l = \|\boldsymbol{U}^\top \boldsymbol{e}_l\|_2^2 = \|\boldsymbol{u}_l\|_2^2, \tag{54}$$

where $\boldsymbol{e}_l$ denotes the $l$-th canonical basis vector in $\mathbb{R}^d$. The fluctuation of the Gram matrix $\mathbf{S}_i$ around its mean

$$\mathbb{E}[\mathbf{S}_i] = \boldsymbol{U}^\top \mathbb{E}[\mathrm{diag}(\mathbf{d}_i)^2]\boldsymbol{U} = \boldsymbol{U}^\top (\mu_{\mathbf{D}}\boldsymbol{I}_d)\boldsymbol{U} = \mu_{\mathbf{D}}\boldsymbol{I}_k \tag{55}$$

can be written as a sum of independent rank 1 matrices

$$\mathbf{S}_i - \mu_{\mathbf{D}}\boldsymbol{I}_k = \sum_{l=1}^{d} \underbrace{((\mathbf{d}_i)_l^2 - \mu_{\mathbf{D}})\boldsymbol{u}_l\boldsymbol{u}_l^\top}_{=:\boldsymbol{A}_l} = \sum_{l=1}^{d} \boldsymbol{A}_l. \tag{56}$$

Clearly, the addends $\boldsymbol{A}_l$ are centered and symmetric. Furthermore, we can bound

$$\|\boldsymbol{A}_l\|_{\mathrm{op}} = |(\mathbf{d}_i)_l^2 - \mu_{\mathbf{D}}| \cdot \|\boldsymbol{u}_l\boldsymbol{u}_l^\top\|_{\mathrm{op}} \leq b_{\mathbf{D}}\|\boldsymbol{u}_l\|_2^2 \leq b_{\mathbf{D}}\nu(\mathcal{U}) =: R. \tag{57}$$

Next, we compute the variance parameter $\sigma^2$. Since

$$\boldsymbol{A}_l^2 = ((\mathbf{d}_i)_l^2 - \mu_{\mathbf{D}})^2(\boldsymbol{u}_l\boldsymbol{u}_l^\top)^2 = ((\mathbf{d}_i)_l^2 - \mu_{\mathbf{D}})^2\|\boldsymbol{u}_l\|_2^2(\boldsymbol{u}_l\boldsymbol{u}_l^\top) \tag{58}$$

and $\mathbb{E}[((\mathbf{d}_i)_l^2 - \mu_{\mathbf{D}})^2] = \sigma_{\mathbf{D}}^2$ by definition, we have

$$\sigma^2 = \left\|\sum_{l=1}^{d} \sigma_{\mathbf{D}}^2\|\boldsymbol{u}_l\|_2^2(\boldsymbol{u}_l\boldsymbol{u}_l^\top)\right\|_{\mathrm{op}} \tag{59}$$

$$\overset{(*)}{\leq} \sigma_{\mathbf{D}}^2 \underbrace{\left(\max_k \|\boldsymbol{u}_k\|_2^2\right)}_{=\nu(\mathcal{U})}\left\|\sum_{l=1}^{d} \boldsymbol{u}_l\boldsymbol{u}_l^\top\right\|_{\mathrm{op}} \tag{60}$$

$$= \sigma_{\mathbf{D}}^2\nu(\mathcal{U})\|\underbrace{\boldsymbol{U}^\top \boldsymbol{U}}_{=\boldsymbol{I}_k}\|_{\mathrm{op}} \tag{61}$$

$$= \sigma_{\mathbf{D}}^2\nu(\mathcal{U}). \tag{62}$$

In $(*)$, we have utilized that for positive semidefinite matrices $\boldsymbol{B}_l \succeq 0$ and scalars $c_l \in [0, C]$, one has $\|\sum_l c_l\boldsymbol{B}_l\|_{\mathrm{op}} \leq C\|\sum_l \boldsymbol{B}_l\|_{\mathrm{op}}$. We are now ready to apply Lem. C.3 to $\sum_{l=1}^{d} \boldsymbol{A}_l$. We set the target failure probability to $\delta \in (0, 1)$. We seek a threshold $t$ such that the failure probability is at most $\delta$. This holds if

$$2k\exp\left(\frac{-t^2/2}{\sigma^2 + Rt/3}\right) \leq \delta \quad \Leftrightarrow \quad \frac{t^2}{2} \geq \left(\sigma^2 + \frac{Rt}{3}\right)\log\left(\frac{2k}{\delta}\right). \tag{63}$$

Let $L := \log(2k/\delta)$. The inequality becomes $t^2 - \frac{2}{3}RLt - 2\sigma^2 L \geq 0$. Solving for $t$ yields

$$t \geq \frac{RL}{3} + \sqrt{\left(\frac{RL}{3}\right)^2 + 2\sigma^2 L} \overset{(**)}{\leq} \frac{2}{3}RL + \sigma\sqrt{2L}, \tag{64}$$

where $(**)$ uses $\sqrt{a+b} \leq \sqrt{a} + \sqrt{b}$ for $a, b \geq 0$ (setting $t$ to this upper bound suffices). Finally, we can plug in $R = b_{\mathbf{D}}\nu(\mathcal{U})$ ((57)) and $\sigma^2 \leq \sigma_{\mathbf{D}}^2\nu(\mathcal{U})$ ((62)), and absorbing absolute constants into $C$, we obtain

$$t \geq C\left\{\sqrt{\sigma_{\mathbf{D}}^2\nu(\mathcal{U})\log(2k/\delta)} + b_{\mathbf{D}}\nu(\mathcal{U})\log(2k/\delta)\right\}, \tag{65}$$

which completes the proof. $\qquad\square$

**Lemma C.4.** *Let $\boldsymbol{H} \in \mathcal{H}(\mathcal{X})$ with $\boldsymbol{h}_i = \eta((\boldsymbol{U}\boldsymbol{a}_i)^\top \cdot)$ and $\boldsymbol{r}_i := \boldsymbol{a}_i - \mathbf{v}_i$. Then,*

$$\Delta_{(ij)}^{\mathrm{out}}(\boldsymbol{H}) = \underbrace{\|\mathbf{v}_j - \mathbf{v}_i\|_{\mathbf{S}_i^{-1}+\mathbf{S}_j^{-1}}^2}_{(i)\ \textit{center term}} + \underbrace{2\langle \mathbf{v}_j - \mathbf{v}_i, \mathbf{S}_i^{-1}\boldsymbol{r}_j - \mathbf{S}_j^{-1}\boldsymbol{r}_i\rangle}_{(ii)\ \textit{cross term}} + \underbrace{\|\boldsymbol{r}_j\|_{\mathbf{S}_i^{-1}-\mathbf{S}_j^{-1}}^2 + \|\boldsymbol{r}_i\|_{\mathbf{S}_j^{-1}-\mathbf{S}_i^{-1}}^2}_{(iii)\ \textit{metric mismatch term}}. \tag{66}$$

*Proof.* Fix $i \neq j$ and write $\mathbf{v}_{ij} := \mathbf{v}_j - \mathbf{v}_i$. For a transposition $(ij)$, only the $i$- and $j$-addends change, hence

$$\Delta_{(ij)}^{\mathrm{out}}(\boldsymbol{H}) = \|\mathbf{v}_j - \mathbf{v}_i + \boldsymbol{r}_j\|_{\mathbf{S}_i^{-1}}^2 + \|\mathbf{v}_i - \mathbf{v}_j + \boldsymbol{r}_i\|_{\mathbf{S}_j^{-1}}^2 - \|\boldsymbol{r}_i\|_{\mathbf{S}_i^{-1}}^2 - \|\boldsymbol{r}_j\|_{\mathbf{S}_j^{-1}}^2$$

$$= \|\mathbf{v}_{ij} + \boldsymbol{r}_j\|_{\mathbf{S}_i^{-1}}^2 + \|-\mathbf{v}_{ij} + \boldsymbol{r}_i\|_{\mathbf{S}_j^{-1}}^2 - \|\boldsymbol{r}_i\|_{\mathbf{S}_i^{-1}}^2 - \|\boldsymbol{r}_j\|_{\mathbf{S}_j^{-1}}^2. \tag{67}$$

Using $\|x\|_A^2 = x^\top A x$ and expanding the quadratic forms in (67) yields

$$\|\mathbf{v}_{ij} + \mathbf{r}_j\|_{\mathbf{S}_i^{-1}}^2 \;=\; \mathbf{v}_{ij}^\top \mathbf{S}_i^{-1} \mathbf{v}_{ij} + 2\, \mathbf{v}_{ij}^\top \mathbf{S}_i^{-1} \mathbf{r}_j + \mathbf{r}_j^\top \mathbf{S}_i^{-1} \mathbf{r}_j, \tag{68}$$

$$\|-\mathbf{v}_{ij} + \mathbf{r}_i\|_{\mathbf{S}_j^{-1}}^2 \;=\; \mathbf{v}_{ij}^\top \mathbf{S}_j^{-1} \mathbf{v}_{ij} - 2\, \mathbf{v}_{ij}^\top \mathbf{S}_j^{-1} \mathbf{r}_i + \mathbf{r}_i^\top \mathbf{S}_j^{-1} \mathbf{r}_i. \tag{69}$$

Substituting (68) and (69) into (67) and regrouping gives

$$\begin{aligned}
\Delta_{(ij)}^{\mathrm{out}}(\boldsymbol{H}) = \;& \mathbf{v}_{ij}^\top(\mathbf{S}_i^{-1} + \mathbf{S}_j^{-1})\mathbf{v}_{ij} + 2\, \mathbf{v}_{ij}^\top(\mathbf{S}_i^{-1}\mathbf{r}_j - \mathbf{S}_j^{-1}\mathbf{r}_i) \\
& + \mathbf{r}_j^\top(\mathbf{S}_i^{-1} - \mathbf{S}_j^{-1})\mathbf{r}_j + \mathbf{r}_i^\top(\mathbf{S}_j^{-1} - \mathbf{S}_i^{-1})\mathbf{r}_i,
\end{aligned} \tag{70}$$

which is precisely (16) / (66). $\qquad\qquad\square$

## C.2. Proofs from § 4.3 (Neuron Reassignments)

**Theorem 4.5** (Global output sensitivity). *Consider a layer $\boldsymbol{H} = \eta\big((\mathbf{F} + \mathbf{D} \odot \boldsymbol{W})\cdot\big)$, and assume (18) holds. Set $\boldsymbol{R} := (\mathbf{D} \odot \boldsymbol{W})\boldsymbol{U} \in \mathbb{R}^{m \times k}$ (proj. trainable parts). Let $\pi \in S_m$ and set $\boldsymbol{M}_\pi := \boldsymbol{I}_m - \mathrm{Diag}\big(\mathrm{diag}(\boldsymbol{P}_\pi)\big)$ to be the projection onto the coordinates affected by $\pi$. Assume that $\boldsymbol{\delta}_\pi^{\mathrm{out}} \neq 0$, $\rho_\pi^{\mathrm{out}} := \|\boldsymbol{M}_\pi \boldsymbol{R}\|_F / \|\boldsymbol{\delta}_\pi^{\mathrm{out}}\|_F \leq 1$. Then,*

$$\Delta_\pi^{\mathrm{out}}(\boldsymbol{H}) \;=\; \Big(1 \pm \mathcal{O}(\rho_\pi^{\mathrm{out}} + \varepsilon \mu_{\mathbf{D}}^{-1})\Big) \cdot \mu_{\mathbf{D}}^{-1} \|\boldsymbol{\delta}_\pi^{\mathrm{out}}\|_F^2. \tag{19}$$

*Proof.* First, note that by definition,

$$\Delta_\pi^{\mathrm{out}}(\boldsymbol{H}, \mathcal{X}) \;=\; \sum_{i=1}^{m} \Big( \big\|(\boldsymbol{\delta}_\pi^{\mathrm{out}})_{i,:} + (\boldsymbol{P}_\pi \boldsymbol{R})_{i,:}\big\|_{\mathbf{S}_i^{-1}}^2 - \|\boldsymbol{R}_{i,:}\|_{\mathbf{S}_i^{-1}}^2 \Big). \tag{71}$$

Now if $\pi(i) = i$, then $(\boldsymbol{\delta}_\pi^{\mathrm{out}})_{i,:} = 0$ and $(\boldsymbol{P}_\pi \boldsymbol{R})_{i,:} = \boldsymbol{R}_{i,:}$, so the $i$-th summand in (71) vanishes. Since $\boldsymbol{M}_\pi$ is the (diagonal) projection onto the rows moved by $\pi$, we may write

$$\Delta_\pi^{\mathrm{out}}(\boldsymbol{H}, \mathcal{X}) \;=\; \sum_{i=1}^{m} \Big( \big\|\big(\boldsymbol{M}_\pi(\boldsymbol{\delta}_\pi^{\mathrm{out}} + \boldsymbol{P}_\pi \boldsymbol{R})\big)_{i,:}\big\|_{\mathbf{S}_i^{-1}}^2 - \|(\boldsymbol{M}_\pi \boldsymbol{R})_{i,:}\|_{\mathbf{S}_i^{-1}}^2 \Big). \tag{72}$$

Since $\varepsilon < \mu_{\mathbf{D}}$, the uniform bound $\|\mathbf{S}_i - \mu_{\mathbf{D}} \boldsymbol{I}_k\|_{\mathrm{op}} \leq \varepsilon$ in $i$ yields $(\mu_{\mathbf{D}} - \varepsilon)\boldsymbol{I}_k \preceq \mathbf{S}_i \preceq (\mu_{\mathbf{D}} + \varepsilon)\boldsymbol{I}_k$. Hence $\mathbf{S}_i$ is invertible, and

$$\big\|\mathbf{S}_i^{-1} - \mu_{\mathbf{D}}^{-1} \boldsymbol{I}_k\big\|_{\mathrm{op}} \;\leq\; \frac{\varepsilon}{\mu_{\mathbf{D}}(\mu_{\mathbf{D}} - \varepsilon)} \;=:\; \beta. \tag{73}$$

Thus, for every $\boldsymbol{a} \in \mathbb{R}^k$,

$$\left| \|\boldsymbol{a}\|_{\mathbf{S}_i^{-1}}^2 - \mu_{\mathbf{D}}^{-1}\|\boldsymbol{a}\|_2^2 \right| \;\leq\; \beta \|\boldsymbol{a}\|_2^2. \tag{74}$$

Applying (74) to each term in (72) yields

$$\Delta_\pi^{\mathrm{out}}(\boldsymbol{H}, \mathcal{X}) \;=\; \mu_{\mathbf{D}}^{-1}\big(\|\boldsymbol{M}_\pi(\boldsymbol{\delta}_\pi^{\mathrm{out}} + \boldsymbol{P}_\pi \boldsymbol{R})\|_F^2 - \|\boldsymbol{M}_\pi \boldsymbol{R}\|_F^2\big) + E_\pi, \tag{75}$$

with

$$|E_\pi| \;\leq\; \beta\big(\|\boldsymbol{M}_\pi(\boldsymbol{\delta}_\pi^{\mathrm{out}} + \boldsymbol{P}_\pi \boldsymbol{R})\|_F^2 + \|\boldsymbol{M}_\pi \boldsymbol{R}\|_F^2\big). \tag{76}$$

Because $\boldsymbol{M}_\pi \boldsymbol{\delta}_\pi^{\mathrm{out}} = \boldsymbol{\delta}_\pi^{\mathrm{out}}$ and $\boldsymbol{M}_\pi \boldsymbol{P}_\pi = \boldsymbol{P}_\pi \boldsymbol{M}_\pi$ implies $\|\boldsymbol{M}_\pi \boldsymbol{P}_\pi \boldsymbol{R}\|_F = \|\boldsymbol{P}_\pi \boldsymbol{M}_\pi \boldsymbol{R}\|_F = \|\boldsymbol{M}_\pi \boldsymbol{R}\|_F$, we can apply Cauchy-Schwarz to bound (76) as

$$|E_\pi| \;\leq\; \beta\Big(\|\boldsymbol{\delta}_\pi^{\mathrm{out}}\|_F^2 + 2\|\boldsymbol{\delta}_\pi^{\mathrm{out}}\|_F\|\boldsymbol{M}_\pi \boldsymbol{R}\|_F + 2\|\boldsymbol{M}_\pi \boldsymbol{R}\|_F^2\Big). \tag{77}$$

It remains to simplify the isotropic term in (75), for which one can obtain

$$\|\boldsymbol{M}_\pi(\boldsymbol{\delta}_\pi^{\mathrm{out}} + \boldsymbol{P}_\pi \boldsymbol{R})\|_F^2 - \|\boldsymbol{M}_\pi \boldsymbol{R}\|_F^2 \;=\; \|\boldsymbol{\delta}_\pi^{\mathrm{out}}\|_F^2 + 2\langle \boldsymbol{\delta}_\pi^{\mathrm{out}}, \boldsymbol{M}_\pi \boldsymbol{P}_\pi \boldsymbol{R}\rangle_F = \|\boldsymbol{\delta}_\pi^{\mathrm{out}}\|_F^2 + 2\langle \boldsymbol{\delta}_\pi^{\mathrm{out}}, \boldsymbol{P}_\pi \boldsymbol{R}\rangle_F, \tag{78}$$

where the last equality holds since $\boldsymbol{M}_\pi^\top = \boldsymbol{M}_\pi$ and $\boldsymbol{M}_\pi \boldsymbol{\delta}_\pi^{\mathrm{out}} = \boldsymbol{\delta}_\pi^{\mathrm{out}}$. Define

$$\boldsymbol{Q}_\pi \;:=\; \boldsymbol{I}_m - \mathrm{ord}(\pi)^{-1} \sum_{t=0}^{\mathrm{ord}(\pi)-1} \boldsymbol{P}_\pi^t. \tag{79}$$

Since $\boldsymbol{P}_\pi^{\mathrm{ord}(\pi)} = \boldsymbol{I}_m$,

$$(\boldsymbol{I}_m - \boldsymbol{Q}_\pi)\boldsymbol{P}_\pi \;=\; \boldsymbol{P}_\pi(\boldsymbol{I}_m - \boldsymbol{Q}_\pi) \;=\; \boldsymbol{I}_m - \boldsymbol{Q}_\pi. \tag{80}$$

Thus, $I_m - Q_\pi$ is the orthogonal projection onto the fixed subspace of $P_\pi$, and $Q_\pi$ is the orthogonal projection onto its orthogonal complement. Moreover,

$$(I_m - Q_\pi)\delta_\pi^{\text{out}} = (I_m - Q_\pi)(P_\pi - I_m)FU = 0. \tag{81}$$

Hence, $Q_\pi \delta_\pi^{\text{out}} = \delta_\pi^{\text{out}}$ and, using $P_\pi(I_m - Q_\pi) = I_m - Q_\pi$,

$$\langle \delta_\pi^{\text{out}}, P_\pi(I_m - Q_\pi)R \rangle_F = \langle \delta_\pi^{\text{out}}, (I_m - Q_\pi)R \rangle_F = 0. \tag{82}$$

Therefore,

$$\langle \delta_\pi^{\text{out}}, P_\pi R \rangle_F = \langle \delta_\pi^{\text{out}}, P_\pi Q_\pi R \rangle_F. \tag{83}$$

By Cauchy–Schwarz and the orthogonality of $P_\pi$,

$$\left| \langle \delta_\pi^{\text{out}}, P_\pi R \rangle_F \right| \leq \|\delta_\pi^{\text{out}}\|_F \|Q_\pi R\|_F. \tag{84}$$

Combining (75), (77), (78), and (84) gives

$$\left| \Delta_\pi^{\text{out}}(H, \mathcal{X}) - \mu_D^{-1}\|\delta_\pi^{\text{out}}\|_F^2 \right| \leq \frac{2}{\mu_D} \|\delta_\pi^{\text{out}}\|_F \|Q_\pi R\|_F$$
$$+ \frac{\varepsilon}{\mu_D(\mu_D - \varepsilon)}\left( \|\delta_\pi^{\text{out}}\|_F^2 + 2\|\delta_\pi^{\text{out}}\|_F \|M_\pi R\|_F + 2\|M_\pi R\|_F^2 \right). \tag{85}$$

For the final statement, note that $Q_\pi = Q_\pi M_\pi$, and as $Q_\pi$ is an orthogonal projection,

$$\|Q_\pi R\|_F = \|Q_\pi M_\pi R\|_F \leq \|M_\pi R\|_F. \tag{86}$$

Dividing (85) by $\mu_D^{-1}\|\delta_\pi^{\text{out}}\|_F^2$ therefore yields

$$\left| \frac{\mu_D \Delta_\pi^{\text{out}}(H, \mathcal{X})}{\|\delta_\pi^{\text{out}}\|_F^2} - 1 \right| \leq 2\rho_\pi^{\text{out}} + \frac{\varepsilon}{\mu_D - \varepsilon}\left( 1 + 2\rho_\pi^{\text{out}} + 2(\rho_\pi^{\text{out}})^2 \right). \tag{87}$$

If $\rho_\pi^{\text{out}} \leq 1$ and $\varepsilon \leq \mu_D/2$, then

$$\frac{\varepsilon}{\mu_D - \varepsilon} \leq \frac{2\varepsilon}{\mu_D}, \qquad 1 + 2\rho_\pi^{\text{out}} + 2(\rho_\pi^{\text{out}})^2 \leq 5. \tag{88}$$

Hence, the r.h.s. is $\mathcal{O}(\rho_\pi^{\text{out}} + \varepsilon\mu_D^{-1})$, and therefore

$$\Delta_\pi^{\text{out}}(H, \mathcal{X}) = \left( 1 \pm \mathcal{O}(\rho_\pi^{\text{out}} + \varepsilon\mu_D^{-1}) \right) \cdot \mu_D^{-1}\|\delta_\pi^{\text{out}}\|_F^2. \tag{89}$$

$\square$

**Corollary C.5** (Minimum output reassignment cost). *In the setup of Thm. 4.5, suppose further that $\rho_{(ij)}^{\text{out}} \leq \rho^{\text{out}} \leq 1$ for some $\rho^{\text{out}}$, uniformly in $i \neq j \in [m]$. Then,*

$$\min_{\text{id} \neq \pi \in S_m} \Delta_\pi^{\text{out}}(H, \mathcal{X}) = \left( 1 \pm \mathcal{O}(\rho^{\text{out}} + \varepsilon\mu_D^{-1}) \right) \cdot \mu_D^{-1}\gamma_{\text{out}}^2. \tag{90}$$

*Proof.* Fix $\text{id} \neq \pi \in S_m$. Then,

$$\|\delta_\pi^{\text{out}}\|_F^2 = \|(P_\pi - I_m)FU\|_F^2 = \sum_{i=1}^m \|v_{\pi(i)} - v_i\|_2^2. \tag{91}$$

Write $\pi$ as a product of disjoint cycles, and consider one nontrivial cycle $(i_1 \ldots i_\ell)$ of length $\ell \geq 2$, and use the convention $i_{\ell+1} := i_1$. Its contribution to the above sum is

$$\sum_{r=1}^\ell \|v_{i_{r+1}} - v_{i_r}\|_2^2 \geq \ell \min_{i \neq j} \|v_j - v_i\|_2^2 \geq 2 \min_{i \neq j} \|v_j - v_i\|_2^2. \tag{92}$$

On the other hand, for a transposition $(ij)$ one has $\|\delta_{(ij)}^{\text{out}}\|_F^2 = 2\|v_j - v_i\|_2^2$. Therefore,

$$\min_{\text{id} \neq \pi \in S_m} \|\delta_\pi^{\text{out}}\|_F = \min_{i \neq j} \|\delta_{(ij)}^{\text{out}}\|_F = \gamma_{\text{out}}. \tag{93}$$

Next, for every $i \neq j$,

$$\|R_{i,:}\|_2^2 + \|R_{j,:}\|_2^2 = \|M_{(ij)}R\|_F^2 = (\rho_{(ij)}^{\text{out}})^2\|\delta_{(ij)}^{\text{out}}\|_F^2 \leq 2(\rho^{\text{out}})^2\|v_j - v_i\|_F^2. \tag{94}$$

Now let $\pi \neq \text{id}$ and decompose its moved indices into $s$ nontrivial disjoint cycles $(i_1^{(s)} \ldots i_{\ell_s}^{(s)})$. Then,

$$2\|M_\pi R\|_F^2 = \sum_s \sum_{r=1}^{\ell_s}\left( \|R_{i_r^{(s)},:}\|_2^2 + \|R_{i_{r+1}^{(s)},:}\|_2^2 \right) \leq 2(\rho^{\text{out}})^2 \sum_s \sum_{r=1}^{\ell_s} \|v_{i_{r+1}^{(s)}} - v_{i_r^{(s)}}\|_2^2 = 2(\rho^{\text{out}})^2\|\delta_\pi^{\text{out}}\|_F^2. \tag{95}$$

Hence, $\rho_\pi^{\text{out}} = \|\boldsymbol{M}_\pi \boldsymbol{R}\|_F / \|\boldsymbol{\delta}_\pi^{\text{out}}\|_F \leq \rho^{\text{out}}$ for all $\pi \neq \text{id}$. By Theorem 4.5, it follows that

$$\Delta_\pi^{\text{out}}(\boldsymbol{H}, \mathcal{X}) \;=\; \left(1 \pm \mathcal{O}(\rho^{\text{out}} + \varepsilon \mu_{\mathbf{D}}^{-1})\right) \cdot \mu_{\mathbf{D}}^{-1} \|\boldsymbol{\delta}_\pi^{\text{out}}\|_F^2 \tag{96}$$

uniformly in $\pi \neq \text{id}$. Using (93) yields

$$\min_{\text{id} \neq \pi \in S_m} \Delta_\pi^{\text{out}}(\boldsymbol{H}, \mathcal{X}) \;=\; \left(1 \pm \mathcal{O}(\rho^{\text{out}} + \varepsilon \mu_{\mathbf{D}}^{-1})\right) \cdot \mu_{\mathbf{D}}^{-1} \gamma_{\text{out}}^2. \tag{97}$$

$\square$

**Theorem 4.6** (Asymptotics of projected center gap $\gamma_{\text{out}}$). *Let $\mathbf{F}$ have i.i.d. entries of the form $B \cdot \sigma_{\mathbf{F}} Y_0$, where $B \sim$ Bernoulli$(p)$, $p \in (0, 1]$, $Y_0$ is independent of $B$ with a Lebesgue density bounded by $M_0 < \infty$, and $\sigma_{\mathbf{F}} > 0$. Set $q := 2p - p^2$ and fix $\delta \in (0, 1)$. Let $m \geq 2$ and assume*

$$q \geq C\{\sqrt{q\,\nu(\mathcal{U})\Lambda} + \nu(\mathcal{U})\Lambda\}, \quad \Lambda := \log(4km^2/\delta). \tag{21}$$

*Then, with probability at least $1 - \delta$,*

$$\gamma_{\text{out}} \geq c\,\frac{\sqrt{q}}{M_0} \sigma_{\mathbf{F}} V_k^{-1/k} \left(\frac{\delta}{m^2}\right)^{1/k} = \Theta(\sigma_{\mathbf{F}} \sqrt{k} m^{-2/k}), \tag{22}$$

*where $V_k$ denotes the volume of the k-dim. unit ball, $V_k^{-1/k} = \Theta(\sqrt{k})$, and $c, C > 0$ are absolute constants.*

*Proof.* Recall that $\mathbf{f}_i \in \mathbb{R}^d$ is the $i$-th row of $\mathbf{F}$ and $\mathbf{v}_i = \boldsymbol{U}^\top \mathbf{f}_i \in \mathbb{R}^k$, further $\gamma_{\text{out}} = \sqrt{2} \min_{i \neq j} \|\mathbf{v}_i - \mathbf{v}_j\|_2$ and $\mathbf{v}_{ij} = \mathbf{v}_i - \mathbf{v}_j = \boldsymbol{U}^\top(\mathbf{f}_i - \mathbf{f}_j) \in \mathbb{R}^k$. Moreover, define $Y := \sigma_{\mathbf{F}} Y_0$, which has a Lebesgue density bounded by $M := \sigma_{\mathbf{F}}^{-1} M_0$.

**Step ①: Random mask concentration.** Fix a pair $(i, j)$. For each $\ell \in [d]$, let

$$\xi_\ell := \mathbb{1}\{(B_{i\ell}, B_{j\ell}) \neq (0, 0)\} \in \{0, 1\}, \qquad \boldsymbol{P} := \boldsymbol{U}^\top \text{Diag}(\xi) \boldsymbol{U} \in \mathbb{R}^{k \times k}. \tag{98}$$

Then, $\xi_\ell$ are i.i.d. Bernoulli variables with

$$\mathbb{E}[\xi_\ell] \;=\; \mathbb{P}(\xi_\ell = 1) \;=\; 1 - (1 - p)^2 \;=\; 2p - p^2 \;=:\; q. \tag{99}$$

Let $\boldsymbol{u}_\ell := \boldsymbol{U}^\top \boldsymbol{e}_\ell \in \mathbb{R}^k$, so that $\|\boldsymbol{u}_\ell\|_2^2 = (\boldsymbol{U}\boldsymbol{U}^\top)_{\ell\ell} \leq \nu(\mathcal{U})$. Then,

$$\boldsymbol{P} - q\boldsymbol{I}_k \;=\; \sum_{\ell=1}^d (\xi_\ell - q)\,\boldsymbol{u}_\ell \boldsymbol{u}_\ell^\top. \tag{100}$$

Set $\boldsymbol{A}_\ell := (\xi_\ell - q)\,\boldsymbol{u}_\ell \boldsymbol{u}_\ell^\top$. Then, $\mathbb{E}[\boldsymbol{A}_\ell] = 0$ and

$$\|\boldsymbol{A}_\ell\|_{\text{op}} \;=\; |\xi_\ell - q|\,\|\boldsymbol{u}_\ell\|_2^2 \;\leq\; \nu(\mathcal{U}) \;=:\; R. \tag{101}$$

Further,

$$\boldsymbol{A}_\ell^2 \;=\; (\xi_\ell - q)^2 (\boldsymbol{u}_\ell \boldsymbol{u}_\ell^\top)^2 \;=\; (\xi_\ell - q)^2 \|\boldsymbol{u}_\ell\|_2^2\,\boldsymbol{u}_\ell \boldsymbol{u}_\ell^\top, \tag{102}$$

so, since $\mathbb{E}[(\xi_\ell - q)^2] = q(1 - q) \leq q$,

$$\mathbb{E}[\boldsymbol{A}_\ell^2] \;\preceq\; q\,\|\boldsymbol{u}_\ell\|_2^2\,\boldsymbol{u}_\ell \boldsymbol{u}_\ell^\top \;\preceq\; q\,\nu(\mathcal{U})\,\boldsymbol{u}_\ell \boldsymbol{u}_\ell^\top. \tag{103}$$

Because $\sum_{\ell=1}^d \boldsymbol{u}_\ell \boldsymbol{u}_\ell^\top = \boldsymbol{U}^\top \boldsymbol{U} = \boldsymbol{I}_k$, it follows that

$$\sum_{\ell=1}^d \mathbb{E}[\boldsymbol{A}_\ell^2] \;\preceq\; q\,\nu(\mathcal{U})\,\boldsymbol{I}_k, \qquad \text{hence} \qquad \sigma^2 := \left\|\sum_{\ell=1}^d \mathbb{E}[\boldsymbol{A}_\ell^2]\right\|_{\text{op}} \;\leq\; q\,\nu(\mathcal{U}). \tag{104}$$

By the matrix Bernstein inequality (Tropp (2015) and Lem. C.3), there exists an absolute constant $C_0 > 0$ such that for all $\eta \in (0, 1)$, with probability at least $1 - \eta$,

$$\|\boldsymbol{P} - q\boldsymbol{I}_k\|_{\text{op}} \;\leq\; C_0\left(\sqrt{\sigma^2 \log(2k/\eta)} + R\log(2k/\eta)\right) \;\leq\; C_0\left(\sqrt{q\,\nu(\mathcal{U}) \log(2k/\eta)} + \nu(\mathcal{U})\log(2k/\eta)\right). \tag{105}$$

Now set $\eta := \delta/(2m^2)$, let

$$\Lambda \;:=\; \log(2k/\eta) \;=\; \log(4km^2/\delta), \tag{106}$$

and define the event

$$\mathcal{G} \;:=\; \left\{\|\boldsymbol{P} - q\boldsymbol{I}_k\|_{\text{op}} \leq \frac{q}{2}\right\}. \tag{107}$$

By (105) and the assumption $q \geq C\left(\sqrt{q\,\nu(\mathcal{U})\Lambda} + \nu(\mathcal{U})\Lambda\right)$, taken with $C := 2C_0$, we have

$$\mathbb{P}(\mathcal{G}) \;\geq\; 1 - \frac{\delta}{2m^2}. \tag{108}$$

On $\mathcal{G}$, we have $\boldsymbol{P} \succeq (q/2)\boldsymbol{I}_k$, and therefore

$$\det(\boldsymbol{P}) \geq (q/2)^k. \tag{109}$$

**Step ②: Uniform conditional density bound for $\mathbf{v}_{ij}$.** Fix $\xi^\star \in \{0, 1\}^d$ such that $\mathbb{P}(\xi = \xi^\star) > 0$ and

$$\left\| \boldsymbol{U}^\top \mathrm{Diag}(\xi^\star)\boldsymbol{U} - q\boldsymbol{I}_k \right\|_{\mathrm{op}} \leq \frac{q}{2}. \tag{110}$$

Let

$$S := \{\ell \in [d] \mid \xi_\ell^\star = 1\} =: \{\ell_1, \ldots, \ell_{|S|}\}, \tag{111}$$

define

$$\zeta := \left((\mathbf{f}_i - \mathbf{f}_j)_{\ell_1}, \ldots, (\mathbf{f}_i - \mathbf{f}_j)_{\ell_{|S|}}\right) \in \mathbb{R}^{|S|}, \tag{112}$$

and let $\boldsymbol{U}_S \in \mathbb{R}^{|S| \times k}$ be the submatrix of $\boldsymbol{U}$ with rows indexed by $S$. Then,

$$\boldsymbol{P}^\star := \boldsymbol{U}^\top \mathrm{Diag}(\xi^\star)\boldsymbol{U} = \boldsymbol{U}_S^\top \boldsymbol{U}_S \succeq \frac{q}{2}\boldsymbol{I}_k. \tag{113}$$

In particular, $\boldsymbol{P}^\star$ is positive definite and $\mathrm{rank}(\boldsymbol{U}_S^\top) = k$. For each $\ell \in [d]$, write $X_\ell := (\mathbf{f}_i - \mathbf{f}_j)_\ell = B_{i\ell}Y_{i\ell} - B_{j\ell}Y_{j\ell}$ and let $\mathcal{E}_\ell := \{\xi_\ell = \xi_\ell^\star\}$. Then, $\{\xi = \xi^\star\} = \bigcap_{\ell=1}^d \mathcal{E}_\ell$, and each $\mathcal{E}_\ell$ is measurable w.r.t. $(B_{i\ell}, B_{j\ell})$. Since the families $(B_{i\ell}, B_{j\ell}, Y_{i\ell}, Y_{j\ell})$ are independent across $\ell \in [d]$, it follows that for arbitrary Borel sets $A_1, \ldots, A_{|S|} \subseteq \mathbb{R}$,

$$\mathbb{P}\left(\bigcap_{r=1}^{|S|}\{X_{\ell_r} \in A_r\} \,\middle|\, \xi = \xi^\star\right) = \frac{\mathbb{P}\left(\bigcap_{r=1}^{|S|}\{X_{\ell_r} \in A_r\} \cap \bigcap_{\ell=1}^d \mathcal{E}_\ell\right)}{\mathbb{P}(\xi = \xi^\star)} \tag{114}$$

$$= \frac{\prod_{r=1}^{|S|} \mathbb{P}(X_{\ell_r} \in A_r, \mathcal{E}_{\ell_r}) \prod_{\ell \notin S} \mathbb{P}(\mathcal{E}_\ell)}{\prod_{r=1}^{|S|} \mathbb{P}(\mathcal{E}_{\ell_r}) \prod_{\ell \notin S} \mathbb{P}(\mathcal{E}_\ell)} \tag{115}$$

$$= \prod_{r=1}^{|S|} \mathbb{P}(X_{\ell_r} \in A_r \mid \mathcal{E}_{\ell_r}). \tag{116}$$

Thus, the coordinates of $\zeta$ are independent under $\mathbb{P}(\cdot|\xi = \xi^\star)$. Now fix $\ell \in S$ for which $\xi_\ell^\star = 1$ and $\mathcal{E}_\ell = \{\xi_\ell = 1\}$. Let $f_Y$ denote the Lebesgue density of $Y$, with $\|f_Y\|_{L^\infty(\lambda)} \leq M$, and let $Y'$ be an independent copy of $Y$. For every Borel set $A \subseteq \mathbb{R}$,

$$\mathbb{P}(X_\ell \in A \mid \mathcal{E}_\ell) = \frac{p(1-p)}{q}\mathbb{P}(Y \in A) + \frac{p(1-p)}{q}\mathbb{P}(-Y \in A) + \frac{p^2}{q}\mathbb{P}(Y - Y' \in A). \tag{117}$$

Hence, under $\mathbb{P}(\cdot|\mathcal{E}_\ell)$, $X_\ell$ has a Lebesgue density which is a convex combination of the densities of $Y$, $-Y$, and $Y - Y'$. $-Y$ has density $x \mapsto f_{-Y}(x) = f_Y(-x)$ and $Y - Y'$ has density $f_{Y-Y'} = f_Y * f_{-Y}$. Thus,

$$\|f_{Y-Y'}\|_{L^\infty(\lambda)} = \|f_Y * f_{-Y}\|_{L^\infty(\lambda)} \leq \underbrace{\|f_Y\|_{L^\infty(\lambda)}}_{\leq M} \underbrace{\|f_{-Y}\|_{L^1(\lambda)}}_{=1} \leq M. \tag{118}$$

Therefore, for every $\ell \in S$, the conditional distribution of $X_\ell$ given $\mathcal{E}_\ell$ admits a Lebesgue density bounded by $M$. Together with (116), this shows that under $\mathbb{P}(\cdot|\xi = \xi^\star)$, $\zeta \in \mathbb{R}^{|S|}$ has independent coordinates, each with Lebesgue density bounded by $M$. Now set $\boldsymbol{A} := \boldsymbol{U}_S^\top \in \mathbb{R}^{k \times |S|}$, so that $\boldsymbol{A}\boldsymbol{A}^\top = \boldsymbol{P}^\star$. Since $\xi_\ell^\star = 0$ implies $B_{i\ell} = B_{j\ell} = 0$, we have $X_\ell = 0$ a.s. under $\mathbb{P}(\cdot|\xi = \xi^\star)$ for every $\ell \notin S$. Consequently,

$$\mathbf{v}_{ij} = \boldsymbol{U}^\top(\mathbf{f}_i - \mathbf{f}_j) = \boldsymbol{U}_S^\top \zeta = \boldsymbol{A}\zeta \qquad \text{a.s. under } \mathbb{P}(\cdot|\xi = \xi^\star). \tag{119}$$

Let $\boldsymbol{\Sigma} := (\boldsymbol{A}\boldsymbol{A}^\top)^{1/2} = (\boldsymbol{P}^\star)^{1/2}$ and $\boldsymbol{Q} := \boldsymbol{\Sigma}^{-1}\boldsymbol{A} \in \mathbb{R}^{k \times |S|}$. Then, $\boldsymbol{Q}\boldsymbol{Q}^\top = \boldsymbol{I}_k$. Writing $\boldsymbol{\Pi} := \boldsymbol{Q}^\top \boldsymbol{Q}$, $\boldsymbol{\Pi}$ is the orthogonal projection onto the $k$-dim. subspace $\mathrm{im}(\boldsymbol{Q}^\top) \subseteq \mathbb{R}^{|S|}$, and $\boldsymbol{Q}\zeta = \boldsymbol{Q}\boldsymbol{\Pi}\zeta$. Let $\lambda_{\mathrm{im}(\boldsymbol{Q}^\top)}$ denote the $k$-dimensional Lebesgue measure on the subspace $\mathrm{im}(\boldsymbol{Q}^\top)$. By Rudelson & Vershynin (2015, Theorem 1.1), applied under the conditional probability measure $\mathbb{P}(\cdot|\xi = \xi^\star)$ to $\zeta$ and $\boldsymbol{\Pi}$, the random vector $\boldsymbol{\Pi}\zeta$ admits a density $f_{\boldsymbol{\Pi}\zeta|\xi=\xi^\star}$ w.r.t. $\lambda_{\mathrm{im}(\boldsymbol{Q}^\top)}$ satisfying

$$\|f_{\boldsymbol{\Pi}\zeta|\xi=\xi^\star}\|_{L^\infty(\lambda_{\mathrm{im}(\boldsymbol{Q}^\top)})} \leq (C_1 M)^k \tag{120}$$

for an absolute constant $C_1 > 0$. Since the restriction $\boldsymbol{Q}\big|_{\mathrm{im}(\boldsymbol{Q}^\top)} : \mathrm{im}(\boldsymbol{Q}^\top) \to \mathbb{R}^k$ is an isometry, it preserves the $k$-dim. Lebesgue measure $\lambda^k$, and thus, $\boldsymbol{Q}\zeta$ admits a Lebesgue density $f_{\boldsymbol{Q}\zeta|\xi=\xi^\star}$ on $\mathbb{R}^k$ with

$$\|f_{\boldsymbol{Q}\zeta|\xi=\xi^\star}\|_{L^\infty(\lambda^k)} \leq (C_1 M)^k. \tag{121}$$

Since $\mathbf{v}_{ij} = \boldsymbol{A}\zeta = \boldsymbol{\Sigma}(\boldsymbol{Q}\zeta)$ under $\mathbb{P}(\cdot\,|\,\xi = \xi^\star)$, a change of variables for the invertible linear map $\boldsymbol{\Sigma} : \mathbb{R}^k \to \mathbb{R}^k$ shows that $\mathbf{v}_{ij}$ admits a Lebesgue density $f_{\mathbf{v}_{ij}|\xi=\xi^\star}$ on $\mathbb{R}^k$ given by

$$f_{\mathbf{v}_{ij}|\xi=\xi^\star}(\boldsymbol{x}) \;=\; \frac{f_{\boldsymbol{Q}\zeta|\xi=\xi^\star}(\boldsymbol{\Sigma}^{-1}\boldsymbol{x})}{|\det\boldsymbol{\Sigma}|}, \qquad \boldsymbol{x} \in \mathbb{R}^k. \tag{122}$$

Therefore,

$$\|f_{\mathbf{v}_{ij}|\xi=\xi^\star}\|_{L^\infty(\lambda^k)} \;\leq\; \frac{\|f_{\boldsymbol{Q}\zeta|\xi=\xi^\star}\|_{L^\infty(\lambda^k)}}{|\det\boldsymbol{\Sigma}|} \;\leq\; \frac{(C_1 M)^k}{\sqrt{\det(\boldsymbol{A}\boldsymbol{A}^\top)}} \;=\; \frac{(C_1 M)^k}{\sqrt{\det(\boldsymbol{P}^\star)}} \;\leq\; (C_1 M)^k \left(\frac{2}{q}\right)^{k/2}. \tag{123}$$

Consequently, for every $t \geq 0$,

$$\mathbb{P}(\|\mathbf{v}_{ij}\|_2 \leq t \,|\, \xi = \xi^\star) \;=\; \int_{B_t^k} f_{\mathbf{v}_{ij}|\xi=\xi^\star} \,\mathrm{d}\lambda^k \tag{124}$$

$$\leq\; \lambda^k(B_t^k)\,\|f_{\mathbf{v}_{ij}|\xi=\xi^\star}\|_{L^\infty(\lambda^k)} \tag{125}$$

$$\leq\; V_k t^k (C_1 M)^k \left(\frac{2}{q}\right)^{k/2} \;=\; \left(C_2 V_k^{1/k} \frac{Mt}{\sqrt{q}}\right)^k, \tag{126}$$

where we set $C_2 := \sqrt{2}\,C_1$.

The bound (126) is uniform over all $\xi^\star \in \{0,1\}^d$ with $\mathbb{P}(\xi = \xi^\star) > 0$ and $\|\boldsymbol{U}^\top \mathrm{Diag}(\xi^\star)\boldsymbol{U} - q\boldsymbol{I}_k\|_{\mathrm{op}} \leq q/2$. Since the event $\mathcal{G}$ is measurable w.r.t. $\xi$, and $\xi$ takes values in the finite set $\{0,1\}^d$, conditioning on $\mathcal{G}$ yields

$$\mathbb{P}(\|\mathbf{v}_{ij}\|_2 \leq t \,|\, \mathcal{G}) \;=\; \sum_{\substack{\xi^\star \in \{0,1\}^d \\ \mathbb{P}(\xi=\xi^\star|\mathcal{G})>0}} \mathbb{P}(\xi = \xi^\star \,|\, \mathcal{G})\,\mathbb{P}(\|\mathbf{v}_{ij}\|_2 \leq t \,|\, \xi = \xi^\star) \;\leq\; \left(C_2 V_k^{1/k} \frac{Mt}{\sqrt{q}}\right)^k. \tag{127}$$

Combining (127) with $\mathbb{P}(\mathcal{G}^\mathsf{c}) \leq \delta/(2m^2)$ from Step ①, we obtain

$$\mathbb{P}(\|\mathbf{v}_{ij}\|_2 \leq t) \;\leq\; \mathbb{P}(\mathcal{G}^\mathsf{c}) + \mathbb{P}(\|\mathbf{v}_{ij}\|_2 \leq t \,|\, \mathcal{G}) \;\leq\; \frac{\delta}{2m^2} + \left(C_2 V_k^{1/k} \frac{Mt}{\sqrt{q}}\right)^k. \tag{128}$$

**Step ③: Union bound over all pairs.** Choose

$$t \;:=\; c\,\frac{\sqrt{q}}{M} V_k^{-1/k} \left(\frac{\delta}{m^2}\right)^{1/k} \tag{129}$$

with $c := (2C_2)^{-1}$. Then,

$$\left(C_2 V_k^{1/k} \frac{Mt}{\sqrt{q}}\right)^k \;=\; (C_2 c)^k \frac{\delta}{m^2} \;=\; \left(\frac{1}{2}\right)^k \frac{\delta}{m^2} \;\leq\; \frac{1}{2}\frac{\delta}{m^2}, \tag{130}$$

so (128) gives for every fixed pair $(i,j)$

$$\mathbb{P}(\|\mathbf{v}_{ij}\|_2 \leq t) \;\leq\; \frac{\delta}{2m^2} + \frac{1}{2}\frac{\delta}{m^2} \;=\; \frac{\delta}{m^2}. \tag{131}$$

By a union bound over the $\binom{m}{2} \leq m^2/2$ pairs,

$$\mathbb{P}\left(\min_{i\neq j}\|\mathbf{v}_{ij}\| \leq t\right) \;\leq\; \sum_{i<j}\mathbb{P}(\|\mathbf{v}_{ij}\|_2 \leq t) \;\leq\; \frac{m^2}{2} \cdot \frac{\delta}{m^2} \;\leq\; \delta. \tag{132}$$

Hence, recalling that $M = \sigma_{\mathbf{F}}^{-1} M_0$,

$$\gamma_{\mathrm{out}} \;=\; \sqrt{2}\min_{i\neq j}\|\mathbf{v}_{ij}\| \;\geq\; \sqrt{2}t \;=\; \sqrt{2}c\,\frac{\sqrt{q}}{M_0}\sigma_{\mathbf{F}} V_k^{-1/k}\left(\frac{\delta}{m^2}\right)^{1/k} \tag{133}$$

with probability at least $1 - \delta$, which is precisely the first inequality of (22), absorbing $\sqrt{2}$ into $c$.

**Step ④: Stirling approximation.** Fixing $\delta, q, M_0$ and using the Stirling formula

$$\Gamma(x+1) \;\sim\; \sqrt{2\pi x}\left(\frac{x}{e}\right)^x \qquad \text{as } x \to \infty, \tag{134}$$

we obtain

$$V_k^{-1/k} = \left(\frac{\Gamma(k/2+1)}{\pi^{k/2}}\right)^{1/k} \sim \left(\frac{\sqrt{\pi k}\left(\frac{k}{2e}\right)^{k/2}}{\pi^{k/2}}\right)^{1/k} = \underbrace{(\pi k)^{1/(2k)}}_{\to 1,\, k\to\infty}\sqrt{\frac{k}{2\pi e}} \sim \sqrt{\frac{k}{2\pi e}}, \tag{135}$$

where $\Gamma(x) := \int_0^\infty t^{x-1}e^{-t}\,\mathrm{d}t$ denotes the gamma function. Hence, $V_k^{-1/k} = \Theta(\sqrt{k})$, and absorbing the constant $(2\pi e)^{-1/2}$ into $c$ yields that the r.h.s. of (133) is $\Theta\bigl(\sigma_{\mathbf{F}}\sqrt{k}\,m^{-2/k}\bigr)$ in $m, k, \sigma_{\mathbf{F}}$. $\qquad\square$

### C.3. Proofs from § 4.4 (Input Reindexings)

**Theorem 4.7** (Global input sensitivity). *In the setup of Thm. 4.5, fix an input reindexing $\tau \in S_d$. Assume that the concentration event (18) holds both for $\mathbf{S}_i$ and $\mathbf{S}_i^\tau$.[9] Also assume that $\boldsymbol{\delta}_\tau^{\mathrm{in}} \neq 0$ and $\rho_\tau^{\mathrm{in}} := \|\mathbf{R}\|_F/\|\boldsymbol{\delta}_\tau^{\mathrm{in}}\|_F \leq 1$. Then,*

$$\Delta_\tau^{\mathrm{in}}(\boldsymbol{H}) = \left(1 \pm \mathcal{O}(\rho_\tau^{\mathrm{in}} + \varepsilon\mu_{\mathbf{D}}^{-1})\right)\cdot\mu_{\mathbf{D}}^{-1}\|\boldsymbol{\delta}_\tau^{\mathrm{in}}\|_F^2. \tag{25}$$

*Proof.* The argument is partly analogous to the proof of Thm. 4.5. For each $i \in [m]$, let $\boldsymbol{a}_i := \mathbf{v}_i + \boldsymbol{R}_{i,:} \in \mathbb{R}^k$ denote the coefficient vector realized by the $i$-th neuron on $\mathcal{X}$. Under the input permutation $\tau$, the transformed function is evaluated on the reindexed support $\boldsymbol{P}_\tau\mathcal{X} \subseteq \boldsymbol{P}_\tau\mathcal{U}$, which has $\boldsymbol{P}_\tau\boldsymbol{U}$ as an orthonormal basis. W.r.t. this basis, the same transformed neuron is still represented by the coefficient vector $\boldsymbol{a}_i$, while the corresponding projected center becomes

$$\mathbf{v}_i^\tau := (\mathbf{F}\boldsymbol{P}_\tau\boldsymbol{U})_{i,:} = \mathbf{v}_i + (\boldsymbol{\delta}_\tau^{\mathrm{in}})_{i,:}. \tag{136}$$

Hence,

$$\Delta_\tau^{\mathrm{in}}(\boldsymbol{H},\mathcal{X}) = \sum_{i=1}^m \left(\left\|\boldsymbol{R}_{i,:} - (\boldsymbol{\delta}_\tau^{\mathrm{in}})_{i,:}\right\|_{(\mathbf{S}_i^\tau)^{-1}}^2 - \|\boldsymbol{R}_{i,:}\|_{\mathbf{S}_i^{-1}}^2\right). \tag{137}$$

Since $\varepsilon < \mu_{\mathbf{D}}$, the assumptions imply

$$(\mu_{\mathbf{D}} - \varepsilon)\boldsymbol{I}_k \preceq \mathbf{S}_i, \mathbf{S}_i^\tau \preceq (\mu_{\mathbf{D}} + \varepsilon)\boldsymbol{I}_k \qquad \text{for all } i \in [m]. \tag{138}$$

Thus, both $\mathbf{S}_i$ and $\mathbf{S}_i^\tau$ are invertible, and

$$\left\|\mathbf{S}_i^{-1} - \mu_{\mathbf{D}}^{-1}\boldsymbol{I}_k\right\|_{\mathrm{op}}, \left\|(\mathbf{S}_i^\tau)^{-1} - \mu_{\mathbf{D}}^{-1}\boldsymbol{I}_k\right\|_{\mathrm{op}} \leq \frac{\varepsilon}{\mu_{\mathbf{D}}(\mu_{\mathbf{D}} - \varepsilon)} =: \beta. \tag{139}$$

Therefore, for every $\boldsymbol{a} \in \mathbb{R}^k$,

$$\left|\|\boldsymbol{a}\|_{\mathbf{S}_i^{-1}}^2 - \mu_{\mathbf{D}}^{-1}\|\boldsymbol{a}\|_2^2\right| \leq \beta\|\boldsymbol{a}\|_2^2, \qquad \left|\|\boldsymbol{a}\|_{(\mathbf{S}_i^\tau)^{-1}}^2 - \mu_{\mathbf{D}}^{-1}\|\boldsymbol{a}\|_2^2\right| \leq \beta\|\boldsymbol{a}\|_2^2. \tag{140}$$

Applying this to each term in (137) yields

$$\Delta_\tau^{\mathrm{in}}(\boldsymbol{H},\mathcal{X}) = \mu_{\mathbf{D}}^{-1}\left(\|\boldsymbol{R} - \boldsymbol{\delta}_\tau^{\mathrm{in}}\|_F^2 - \|\boldsymbol{R}\|_F^2\right) + E_\tau, \tag{141}$$

with

$$|E_\tau| \leq \beta\left(\|\boldsymbol{R} - \boldsymbol{\delta}_\tau^{\mathrm{in}}\|_F^2 + \|\boldsymbol{R}\|_F^2\right). \tag{142}$$

By Cauchy–Schwarz,

$$|E_\tau| \leq \beta\left(\|\boldsymbol{\delta}_\tau^{\mathrm{in}}\|_F^2 + 2\|\boldsymbol{\delta}_\tau^{\mathrm{in}}\|_F\|\boldsymbol{R}\|_F + 2\|\boldsymbol{R}\|_F^2\right). \tag{143}$$

Moreover,

$$\|\boldsymbol{R} - \boldsymbol{\delta}_\tau^{\mathrm{in}}\|_F^2 - \|\boldsymbol{R}\|_F^2 = \|\boldsymbol{\delta}_\tau^{\mathrm{in}}\|_F^2 - 2\left\langle\boldsymbol{\delta}_\tau^{\mathrm{in}}, \boldsymbol{R}\right\rangle_F, \tag{144}$$

and by Cauchy–Schwarz,

$$\left|\left\langle\boldsymbol{\delta}_\tau^{\mathrm{in}}, \boldsymbol{R}\right\rangle_F\right| \leq \|\boldsymbol{\delta}_\tau^{\mathrm{in}}\|_F\|\boldsymbol{R}\|_F. \tag{145}$$

Combining (141), (143), (144), and (145) gives

$$\left|\Delta_\tau^{\mathrm{in}}(\boldsymbol{H},\mathcal{X}) - \mu_{\mathbf{D}}^{-1}\|\boldsymbol{\delta}_\tau^{\mathrm{in}}\|_F^2\right| \leq \frac{2}{\mu_{\mathbf{D}}}\|\boldsymbol{\delta}_\tau^{\mathrm{in}}\|_F\|\boldsymbol{R}\|_F + \frac{\varepsilon}{\mu_{\mathbf{D}}(\mu_{\mathbf{D}} - \varepsilon)}\left(\|\boldsymbol{\delta}_\tau^{\mathrm{in}}\|_F^2 + 2\|\boldsymbol{\delta}_\tau^{\mathrm{in}}\|_F\|\boldsymbol{R}\|_F + 2\|\boldsymbol{R}\|_F^2\right). \tag{146}$$

---

[9]When $\mathbf{D}$ has i.i.d. entries, $\mathbf{S}_i^\tau = \boldsymbol{U}^\top\boldsymbol{P}_\tau^\top\mathrm{Diag}(\mathbf{d}_i)^2\boldsymbol{P}_\tau\boldsymbol{U} \overset{d}{=} \mathbf{S}_i$.

Finally, divide (146) by $\mu_{\mathbf{D}}^{-1}\|\boldsymbol{\delta}_\tau^{\text{in}}\|_F^2$ to obtain

$$\left|\frac{\mu_{\mathbf{D}}\Delta_\tau^{\text{in}}(\boldsymbol{H},\mathcal{X})}{\|\boldsymbol{\delta}_\tau^{\text{in}}\|_F^2} - 1\right| \leq 2\rho_\tau^{\text{in}} + \frac{\varepsilon}{\mu_{\mathbf{D}} - \varepsilon}\left(1 + 2\rho_\tau^{\text{in}} + 2(\rho_\tau^{\text{in}})^2\right). \tag{147}$$

If $\rho_\tau^{\text{in}} \leq 1$ and $\varepsilon \leq \mu_{\mathbf{D}}/2$, then

$$\frac{\varepsilon}{\mu_{\mathbf{D}} - \varepsilon} \leq \frac{2\varepsilon}{\mu_{\mathbf{D}}}, \qquad 1 + 2\rho_\tau^{\text{in}} + 2(\rho_\tau^{\text{in}})^2 \leq 5. \tag{148}$$

Thus the r.h.s. is $\mathcal{O}(\rho_\tau^{\text{in}} + \varepsilon\mu_{\mathbf{D}}^{-1})$, and therefore

$$\Delta_\tau^{\text{in}}(\boldsymbol{H},\mathcal{X}) = \left(1 \pm \mathcal{O}(\rho_\tau^{\text{in}} + \varepsilon\mu_{\mathbf{D}}^{-1})\right) \cdot \mu_{\mathbf{D}}^{-1}\|\boldsymbol{\delta}_\tau^{\text{in}}\|_F^2. \tag{149}$$

$\square$

**Corollary C.6** (Minimum input transposition cost). *Assume that the assumptions of Thm. 4.7 hold for every transposition* $(ab) \in S_d$, *and that* $\rho_{(ab)}^{\text{in}} \leq \rho^{\text{in}} \leq 1$ *for some* $\rho^{\text{in}}$ *uniformly in* $a \neq b$. *Then,*

$$\min_{a \neq b} \Delta_{(ab)}^{\text{in}}(\boldsymbol{H},\mathcal{X}) = \left(1 \pm \mathcal{O}(\rho^{\text{in}} + \varepsilon\mu_{\mathbf{D}}^{-1})\right) \cdot \mu_{\mathbf{D}}^{-1}\gamma_{\text{in}}^2. \tag{150}$$

*Proof.* Since $\rho_{(ab)}^{\text{in}} \leq \rho^{\text{in}}$ for every transposition $(ab)$, by Thm. 4.7,

$$\Delta_{(ab)}^{\text{in}}(\boldsymbol{H},\mathcal{X}) = \left(1 \pm \mathcal{O}(\rho^{\text{in}} + \varepsilon\mu_{\mathbf{D}}^{-1})\right) \cdot \mu_{\mathbf{D}}^{-1}\|\boldsymbol{\delta}_{(ab)}^{\text{in}}\|_F^2 \tag{151}$$

uniformly in $a \neq b$. Taking the minimum over all transpositions gives

$$\min_{a \neq b} \Delta_{(ab)}^{\text{in}}(\boldsymbol{H},\mathcal{X}) = \left(1 \pm \mathcal{O}(\rho^{\text{in}} + \varepsilon\mu_{\mathbf{D}}^{-1})\right) \cdot \mu_{\mathbf{D}}^{-1}\gamma_{\text{in}}^2, \tag{152}$$

which is exactly the claim. $\square$

**Theorem 4.8** (Asymptotics of $\gamma_{\text{in}}$, informal). *Let* $\mathbf{F}_{:,\ell} \in \mathbb{R}^m$ *denote the $\ell$-th column of* $\mathbf{F}$. *Let* $a \neq b \in [d]$. *Then,*

$$\|\boldsymbol{\delta}_{(ab)}^{\text{in}}\|_F = \|\mathbf{F}_{:,b} - \mathbf{F}_{:,a}\|_2 \, \|\boldsymbol{U}_{b,:} - \boldsymbol{U}_{a,:}\|_2. \tag{27}$$

*If further* $\mathbf{F}$ *has i.i.d. centered subgaussian entries with variance* $\sigma_{\mathbf{F}}^2$, *subgaussian norm uniformly bounded by* $C\sigma_{\mathbf{F}}$, *and* $m \gtrsim C^4 \log(d^2/\delta)$, *then with probability at least* $1 - \delta$,

$$\gamma_{\text{in}} = \Theta(\sigma_{\mathbf{F}}\sqrt{m}) \cdot \min_{a \neq b}\|\boldsymbol{U}_{b,:} - \boldsymbol{U}_{a,:}\|_2. \tag{28}$$

*Proof.* **Step ①: Proof of (27).** Fix $a \neq b$. For a transposition $(ab)$, we have for any $\boldsymbol{x} \in \mathbb{R}^d$

$$(\boldsymbol{P}_{(ab)} - \boldsymbol{I}_d)\boldsymbol{x} = (x_b - x_a)(\boldsymbol{e}_a - \boldsymbol{e}_b). \tag{153}$$

Thus,

$$(\boldsymbol{\delta}_{(ab)}^{\text{in}}) = \mathbf{F}(\boldsymbol{P}_{(ab)} - \boldsymbol{I}_d)\boldsymbol{U} = \mathbf{F}(\boldsymbol{e}_a - \boldsymbol{e}_b)(\boldsymbol{U}_{b,:} - \boldsymbol{U}_{a,:}) = (\mathbf{F}_{:,a} - \mathbf{F}_{:,b})(\boldsymbol{U}_{b,:} - \boldsymbol{U}_{a,:}), \tag{154}$$

which is a matrix of rank $\leq 1$. Using $\|\boldsymbol{x}\boldsymbol{y}^\top\|_F = \|\boldsymbol{x}\|_2\|\boldsymbol{y}\|_2$ gives (27).

**Step ②: Proof of (28).** We assume that the entries of $\mathbf{F}$ are i.i.d., centered, have variance $\sigma_{\mathbf{F}}^2$, and subgaussian norm $K := \|\mathbf{F}_{ij}\|_{\psi_2}$. Fix $a < b$ and define $Z_i := \mathbf{F}_{ib} - \mathbf{F}_{ia}$, for $i \in [m]$. Then, $Z_1, \ldots, Z_m$ are independent and centered, with

$$\mathbb{E}Z_i^2 = \mathbb{E}\mathbf{F}_{ib}^2 + \mathbb{E}\mathbf{F}_{ia}^2 = 2\sigma_{\mathbf{F}}^2. \tag{155}$$

Moreover, by the triangle inequality for the subgaussian norm,

$$\|Z_i\|_{\psi_2} \leq \|\mathbf{F}_{ib}\|_{\psi_2} + \|\mathbf{F}_{ia}\|_{\psi_2} = 2K. \tag{156}$$

Therefore, for $Y_i := Z_i^2 - \mathbb{E}Z_i^2$, the random variables $Y_1, \ldots, Y_m$ are independent, centered, and subexponential. In particular, there exists a universal constant $C_0 > 0$ such that

$$\|Y_i\|_{\psi_1} \leq C_0\|Z_i\|_{\psi_2}^2 \leq 4C_0K^2. \tag{157}$$

Hence, by the Bernstein inequality for sums of independent subexponential random variables (e.g., Vershynin (2018, Thm. 2.8.2)), and absorbing the universal constant in the bound on $\|Y_i\|_{\psi_1}$, there exists a universal constant $c_1 > 0$ such that for every $t \geq 0$,

$$\mathbb{P}\left(\left|\sum_{i=1}^m Y_i\right| \geq t\right) \leq 2\exp\left(-c_1\min\left\{\frac{t^2}{mK^4}, \frac{t}{K^2}\right\}\right). \tag{158}$$

Set $L := \log(d^2/\delta)$ and choose

$$t := C_1 K^2 \left( \sqrt{mL} + L \right),$$ (159)

where $C_1 > 0$ is a universal constant yet to be determined. Then,

$$\frac{t}{K^2} = C_1 \left( \sqrt{mL} + L \right) \geq C_1 L, \qquad \frac{t^2}{mK^4} = C_1^2 \frac{(\sqrt{mL} + L)^2}{m} \geq C_1^2 L.$$ (160)

Thus,

$$\min \left\{ \frac{t^2}{mK^4}, \frac{t}{K^2} \right\} \geq \min\{C_1^2, C_1\} L.$$ (161)

Choosing $C_1$ large enough so that $c_1 \min\{C_1^2, C_1\} \geq 1$, we obtain

$$\mathbb{P}\left( \left| \sum_{i=1}^m Y_i \right| \geq t \right) \leq 2 \exp\left( - c_1 \min\{C_1^2, C_1\} \cdot L \right) \leq 2e^{-L} = \frac{2\delta}{d^2}.$$ (162)

Since

$$\sum_{i=1}^m Y_i = \sum_{i=1}^m Z_i^2 - 2m\sigma_{\mathbf{F}}^2 = \|\mathbf{F}_{:,b} - \mathbf{F}_{:,a}\|_2^2 - 2m\sigma_{\mathbf{F}}^2,$$ (163)

we have shown that, for this fixed pair $a < b$,

$$\mathbb{P}\left( \left| \|\mathbf{F}_{:,b} - \mathbf{F}_{:,a}\|_2^2 - 2m\sigma_{\mathbf{F}}^2 \right| \geq t \right) \leq \frac{2\delta}{d^2}.$$ (164)

Taking a union bound over the $\binom{d}{2} \leq d^2/2$ pairs, we conclude that, with probability at least $1 - \delta$, simultaneously for all $a < b$,

$$\left| \|\mathbf{F}_{:,b} - \mathbf{F}_{:,a}\|_2^2 - 2m\sigma_{\mathbf{F}}^2 \right| \leq C_1 K^2 \left( \sqrt{m \log(d^2/\delta)} + \log(d^2/\delta) \right).$$ (165)

On this event, if $t < 2m\sigma_{\mathbf{F}}^2$, then for every $a \neq b$,

$$\sqrt{2m\sigma_{\mathbf{F}}^2 - t} \leq \|\mathbf{F}_{:,b} - \mathbf{F}_{:,a}\|_2 \leq \sqrt{2m\sigma_{\mathbf{F}}^2 + t}.$$ (166)

Combining this with (27) gives

$$\sqrt{2m\sigma_{\mathbf{F}}^2 - t} \, \min_{a \neq b} \|\boldsymbol{U}_{a,:} - \boldsymbol{U}_{b,:}\|_2 \leq \gamma_{\mathrm{in}} \leq \sqrt{2m\sigma_{\mathbf{F}}^2 + t} \, \min_{a \neq b} \|\boldsymbol{U}_{a,:} - \boldsymbol{U}_{b,:}\|_2.$$ (167)

It remains to translate (167) into the asymptotic statement. By assumption, $K = \|\mathbf{F}_{ij}\|_{\psi_2} \leq C\sigma_{\mathbf{F}}$. Increasing $C$ if necessary, we may assume $C \geq 1$. We obtain

$$t \leq C_1 C^2 \sigma_{\mathbf{F}}^2 \left( \sqrt{mL} + L \right).$$ (168)

We now choose a universal constant $C_2 \geq \max\{4C_1^2, 2C_1\}$. If $m \geq C_2 C^4 L$, then

$$C_1 C^2 \sqrt{mL} \leq m/2, \qquad C_1 C^2 L \leq m/2.$$ (169)

Indeed, the first inequality follows from $C_1 C^2 \sqrt{mL} \leq m/2 \Leftrightarrow m \geq 4C_1^2 C^4 L$; the second follows since $C \geq 1$ and $C_2 \geq 2C_1$. Therefore, $t \leq m\sigma_{\mathbf{F}}^2$, and substituting this into (167) gives

$$\sigma_{\mathbf{F}} \sqrt{m} \, \min_{a \neq b} \|\boldsymbol{U}_{b,:} - \boldsymbol{U}_{a,:}\|_2 \leq \gamma_{\mathrm{in}} \leq \sqrt{3} \, \sigma_{\mathbf{F}} \sqrt{m} \, \min_{a \neq b} \|\boldsymbol{U}_{b,:} - \boldsymbol{U}_{a,:}\|_2.$$ (170)

Equivalently,

$$\gamma_{\mathrm{in}} = \Theta(\sigma_{\mathbf{F}} \sqrt{m}) \cdot \min_{a \neq b} \|\boldsymbol{U}_{b,:} - \boldsymbol{U}_{a,:}\|_2$$ (171)

with probability at least $1 - \delta$. $\qquad \square$

### C.4. Theoretical Properties in § 4.5 (Identifiability vs. Feature Learning)

Let $\boldsymbol{x} \sim \mathbb{P} := \mathcal{N}(0, \boldsymbol{I}_d)$ and let $\eta = \mathrm{ReLU}$. For any measurable $h : \mathbb{R}^d \to \mathbb{R}$, write $\|h\|_{L^2(\mathbb{P})}^2 := \mathbb{E}_{\boldsymbol{x}}[h(\boldsymbol{x})^2]$. For the symmetric full feature learning regime, consider a two-layer network of width $m$ as

$$f_{\mathrm{sym}}(\boldsymbol{x}) := \boldsymbol{\alpha}^\top \eta(\boldsymbol{W}\boldsymbol{x} + \boldsymbol{b}),$$ (172)

where $\boldsymbol{\alpha} \in \mathbb{R}^m, \boldsymbol{W} \in \mathbb{R}^{m \times d}$, and $\boldsymbol{b} \in \mathbb{R}^m$ are trainable and $\eta$ acts elementwise. For an asymmetric model, fix centers $\mathbf{F} \in \mathbb{R}^{m \times d}$ and consider

$$f_{\mathrm{asym}}(\boldsymbol{x}) := \boldsymbol{\alpha}^\top \eta((\mathbf{F} + \boldsymbol{W})\boldsymbol{x}),$$ (173)

where $\boldsymbol{\alpha} \in \mathbb{R}^m$ and $\boldsymbol{W} \in \mathbb{R}^{m \times d}$ are trainable.

**Lemma C.7** (Lipschitz stability of ridge features). *Let $\boldsymbol{x} \sim \mathbb{P} = \mathcal{N}(0, \boldsymbol{I}_d)$ and $\eta = \mathrm{ReLU}$. Then for any $\mathbf{f} \in \mathbb{R}^d$ and any perturbation $\boldsymbol{w} \in \mathbb{R}^d$,*

$$\left\| \eta\big((\mathbf{f} + \boldsymbol{w})^\top \boldsymbol{x}\big) - \eta\big(\mathbf{f}^\top \boldsymbol{x}\big) \right\|_{L^2(\mathbb{P})} \leq \|\boldsymbol{w}\|_2. \tag{174}$$

*Proof.* Since $\eta = \mathrm{ReLU}$ is 1-Lipschitz, we have pointwise for all $\boldsymbol{x} \in \mathbb{R}^d$,

$$\left| \eta\big((\mathbf{f} + \boldsymbol{w})^\top \boldsymbol{x}\big) - \eta\big(\mathbf{f}^\top \boldsymbol{x}\big) \right| \leq \left| (\mathbf{f} + \boldsymbol{w})^\top \boldsymbol{x} - \mathbf{f}^\top \boldsymbol{x} \right| = |\boldsymbol{w}^\top \boldsymbol{x}|. \tag{175}$$

Taking the expectation under $\boldsymbol{x}$ yields

$$\left\| \eta\big((\mathbf{f} + \boldsymbol{w})^\top \boldsymbol{x}\big) - \eta\big(\mathbf{f}^\top \boldsymbol{x}\big) \right\|_{L^2(\mathbb{P})}^2 \leq \mathbb{E}_{\boldsymbol{x}}\big[(\boldsymbol{w}^\top \boldsymbol{x})^2\big] = \boldsymbol{w}^\top \mathbb{E}_{\boldsymbol{x}}[\boldsymbol{x}\boldsymbol{x}^\top] \boldsymbol{w} = \|\boldsymbol{w}\|_2^2, \tag{176}$$

since $\mathbb{E}_{\boldsymbol{x}}[\boldsymbol{x}\boldsymbol{x}^\top] = \boldsymbol{I}_d$. $\qquad\square$

**Lemma C.8** (Asymmetric network stability under bounded perturbations). *Let $\boldsymbol{x} \sim \mathbb{P} = \mathcal{N}(0, \boldsymbol{I}_d)$ and $\eta = \mathrm{ReLU}$. Fix $\mathbf{F} \in \mathbb{R}^{m \times d}$ and consider the following two networks with the same output weights $\boldsymbol{\alpha} \in \mathbb{R}^m$*

$$f_{\mathrm{asym}}(\boldsymbol{x}) := \boldsymbol{\alpha}^\top \eta\big((\mathbf{F} + \boldsymbol{W})\boldsymbol{x}\big), \qquad f_0(\boldsymbol{x}) := \boldsymbol{\alpha}^\top \eta(\mathbf{F}\boldsymbol{x}), \tag{177}$$

*where $\boldsymbol{W} \in \mathbb{R}^{m \times d}$ is arbitrary. Then,*

$$\|f_{\mathrm{asym}} - f_0\|_{L^2(\mathbb{P})} \leq \|\boldsymbol{\alpha}\|_2 \|\boldsymbol{W}\|_F. \tag{178}$$

*Proof.* Write $\mathbf{f}_i \in \mathbb{R}^d$ and $\boldsymbol{w}_i \in \mathbb{R}^d$ for the $i$-th rows of $\mathbf{F}$ and $\boldsymbol{W}$, respectively, so that

$$f_{\mathrm{asym}}(\boldsymbol{x}) - f_0(\boldsymbol{x}) = \sum_{i=1}^m \alpha_i \Big( \eta\big((\mathbf{f}_i + \boldsymbol{w}_i)^\top \boldsymbol{x}\big) - \eta\big(\mathbf{f}_i^\top \boldsymbol{x}\big) \Big). \tag{179}$$

Let $\delta_i(\boldsymbol{x}) := \eta\big((\mathbf{f}_i + \boldsymbol{w}_i)^\top \boldsymbol{x}\big) - \eta\big(\mathbf{f}_i^\top \boldsymbol{x}\big)$. By Cauchy–Schwarz applied pointwise in $\boldsymbol{x}$,

$$\left| f_{\mathrm{asym}}(\boldsymbol{x}) - f_0(\boldsymbol{x}) \right| \leq \|\boldsymbol{\alpha}\|_2 \|\boldsymbol{\delta}(\boldsymbol{x})\|_2, \qquad \boldsymbol{\delta}(\boldsymbol{x}) := (\delta_1(\boldsymbol{x}), \ldots, \delta_m(\boldsymbol{x}))^\top. \tag{180}$$

Squaring and taking expectation gives

$$\|f_{\mathrm{asym}} - f_0\|_{L^2(\mathbb{P})}^2 \leq \|\boldsymbol{\alpha}\|_2^2 \mathbb{E}_{\boldsymbol{x}}\big[\|\boldsymbol{\delta}(\boldsymbol{x})\|_2^2\big] = \|\boldsymbol{\alpha}\|_2^2 \sum_{i=1}^m \|\delta_i\|_{L^2(\mathbb{P})}^2. \tag{181}$$

Applying Lem. C.7 to each $(\mathbf{f}_i, \boldsymbol{w}_i)$ yields $\|\delta_i\|_{L^2(\mathbb{P})} \leq \|\boldsymbol{w}_i\|_2$, hence

$$\|f_{\mathrm{asym}} - f_0\|_{L^2(\mathbb{P})}^2 \leq \|\boldsymbol{\alpha}\|_2^2 \sum_{i=1}^m \|\boldsymbol{w}_i\|_2^2 = \|\boldsymbol{\alpha}\|_2^2 \|\boldsymbol{W}\|_F^2. \tag{182}$$

$\qquad\square$

**Theorem C.9** (Random feature hardness for single ReLU neuron (Yehudai & Shamir, 2019), specialization of Theorem 4.2). *There exists a universal constant $c > 0$ such that for all $d > 40$, the following holds: For every $\boldsymbol{w}_\star \in \mathbb{R}^d$ with $\|\boldsymbol{w}_\star\|_2 = d^2$, there exists a bias $b_\star \in \mathbb{R}$ with $|b_\star| \leq 6d^3 + 1$ such that for any coefficients $\alpha_1, \ldots, \alpha_m \in \mathbb{R}$, with probability at least $1 - \exp(-cd)$ over i.i.d. $\boldsymbol{b}_1, \ldots, \boldsymbol{b}_m \sim \mathrm{Unif}(\mathbb{S}^{d-1})$,*

$$\mathbb{E}_{\boldsymbol{x} \sim \mathcal{N}(0, \boldsymbol{I}_d)}\left[ \left( \sum_{i=1}^m \alpha_i \, \eta(\boldsymbol{b}_i^\top \boldsymbol{x}) - \eta(\boldsymbol{w}_\star^\top \boldsymbol{x} + b_\star) \right)^2 \right] \leq \frac{1}{50} \implies m \cdot \max_{i \in [m]} |\alpha_i| \geq \frac{1}{48d^2} \exp(cd). \tag{183}$$

**Theorem C.10** (Hardness in center-dominated regime). *Let $\boldsymbol{x} \sim \mathbb{P} = \mathcal{N}(0, \boldsymbol{I}_d)$, $\eta = \mathrm{ReLU}$, and fix $d > 40$, $m \in \mathbb{N}$. Draw $\mathbf{F} \in \mathbb{R}^{m \times d}$ with i.i.d. $\mathbf{f}_i \sim \mathrm{Unif}(\mathbb{S}^{d-1})$. The following holds for every $\boldsymbol{w}_\star \in \mathbb{R}^d$ with $\|\boldsymbol{w}_\star\|_2 = d^2$: There exists a bias $b_\star \in \mathbb{R}$ with $|b_\star| \leq 6d^3 + 1$ such that for all $\boldsymbol{\alpha} \in \mathbb{R}^m$ and $\boldsymbol{W} \in \mathbb{R}^{m \times d}$ satisfying $\|\boldsymbol{\alpha}\|_\infty < \frac{1}{48md^2} \exp(cd)$ and $\|\boldsymbol{\alpha}\|_2 \|\boldsymbol{W}\|_F \leq \frac{1}{10\sqrt{2}}$, with probability at least $1 - \exp(-cd)$ over sampling $\mathbf{F}$, the asymmetric network $f_{\mathrm{asym}}(\boldsymbol{x}) := \boldsymbol{\alpha}^\top \eta\big((\mathbf{F} + \boldsymbol{W})\boldsymbol{x}\big)$ incurs the population error lower bound*

$$\|f_{\mathrm{asym}} - \eta(\boldsymbol{w}_\star^\top(\cdot) + b_\star)\|_{L^2(\mathbb{P})} > 1/(10\sqrt{2}). \tag{184}$$

*Proof.* Work on the event $\mathcal{E}$ (of probability at least $1 - \exp(-cd)$) on which Thm. C.9 holds for $\{\mathbf{f}_i\}_{i=1}^m$. Fix any $\boldsymbol{w}_\star \in \mathbb{R}^d$ with $\|\boldsymbol{w}_\star\|_2 = d^2$. On $\mathcal{E}$, Thm. C.9 provides a bias $b_\star \in \mathbb{R}$ with $|b_\star| \leq 6d^3 + 1$ such that for every choice of coefficients

$\alpha_1, \ldots, \alpha_m \in \mathbb{R}$,

$$\mathbb{E}_{\boldsymbol{x} \sim \mathbb{P}}\left[\left(\sum_{i=1}^m \alpha_i \, \eta(\mathbf{f}_i^\top \boldsymbol{x}) - \eta(\boldsymbol{w}_\star^\top \boldsymbol{x} + b_\star)\right)^2\right] \leq \frac{1}{50} \implies m \max_{i \in [m]} |\alpha_i| \geq \frac{1}{48d^2} \exp(cd). \tag{185}$$

Now fix arbitrary $\boldsymbol{\alpha} \in \mathbb{R}^m$ and $\boldsymbol{W} \in \mathbb{R}^{m \times d}$ satisfying

$$\|\boldsymbol{\alpha}\|_\infty < \frac{1}{48md^2} \exp(cd), \qquad \|\boldsymbol{\alpha}\|_2 \|\boldsymbol{W}\|_F \leq \frac{1}{10\sqrt{2}}. \tag{186}$$

Define

$$f_{\text{asym}}(\boldsymbol{x}) := \boldsymbol{\alpha}^\top \eta\big((\mathbf{F} + \boldsymbol{W})\boldsymbol{x}\big), \qquad f_0(\boldsymbol{x}) := \boldsymbol{\alpha}^\top \eta(\mathbf{F}\boldsymbol{x}) = \sum_{i=1}^m \alpha_i \, \eta(\mathbf{f}_i^\top \boldsymbol{x}). \tag{187}$$

By Lem. C.8,

$$\|f_{\text{asym}} - f_0\|_{L^2(\mathbb{P})} \leq \|\boldsymbol{\alpha}\|_2 \|\boldsymbol{W}\|_F \leq \frac{1}{10\sqrt{2}}. \tag{188}$$

Next,

$$m \max_{i \in [m]} |\alpha_i| < \frac{1}{48d^2} \exp(cd) \tag{189}$$

as required in the theorem. Therefore, by the contrapositive of (185), we must have

$$\mathbb{E}_{\boldsymbol{x} \sim \mathbb{P}}\big[(f_0(\boldsymbol{x}) - \eta(\boldsymbol{w}_\star^\top \boldsymbol{x} + b_\star))^2\big] > \frac{1}{50}, \qquad \text{i.e.} \qquad \|f_0 - \eta(\boldsymbol{w}_\star^\top \boldsymbol{x} + b_\star)\|_{L^2(\mathbb{P})} > \frac{1}{\sqrt{50}} = \frac{1}{5\sqrt{2}}. \tag{190}$$

Finally, by the triangle inequality and (188),

$$\|f_{\text{asym}} - \eta(\boldsymbol{w}_\star^\top \boldsymbol{x} + b_\star)\|_{L^2(\mathbb{P})} \geq \|f_0 - \eta(\boldsymbol{w}_\star^\top \boldsymbol{x} + b_\star)\|_{L^2(\mathbb{P})} - \|f_{\text{asym}} - f_0\|_{L^2(\mathbb{P})} > \frac{1}{5\sqrt{2}} - \frac{1}{10\sqrt{2}} = \frac{1}{10\sqrt{2}}. \tag{191}$$

$\square$

# D. Linear Mode Connectivity (§ 5)

## D.1. Proofs from § 5

**Theorem 5.1** (Chord deviation in center-dominated regime, ReLU). *Let $\eta = \mathrm{ReLU}$ and $\boldsymbol{x} \sim \mathcal{N}(0, \boldsymbol{\Sigma})$ with $\boldsymbol{\Sigma} \succeq 0$. Fix $\boldsymbol{W}^A, \boldsymbol{W}^B \in \mathbb{R}^{m \times d}$. Suppose there exists $\beta \in [0,1)$ such that for all $i \in [m]$,*

$$\|\mathbf{d}_i \odot \boldsymbol{w}_i^A\|_{\boldsymbol{\Sigma}} \le \beta \|\mathbf{f}_i\|_{\boldsymbol{\Sigma}}, \quad \|\mathbf{d}_i \odot \boldsymbol{w}_i^B\|_{\boldsymbol{\Sigma}} \le \beta \|\mathbf{f}_i\|_{\boldsymbol{\Sigma}}, \tag{31}$$

*where $\|u\|_{\boldsymbol{\Sigma}} := \sqrt{u^\top \boldsymbol{\Sigma} u}$. Then,*

$$\sup_{\lambda \in [0,1]} \big\|\xi_{\boldsymbol{H}}(\lambda; \cdot)\big\|_{L^2(\mathbb{P}; \mathbb{R}^m)} = \mathcal{O}(\beta^{3/2}) \|\mathbf{F}\boldsymbol{\Sigma}^{1/2}\|_F. \tag{32}$$

*Proof.* Fix $i \in [m]$. Write

$$\boldsymbol{w}_{\mathrm{eff},i}^A := \mathbf{f}_i + \mathbf{d}_i \odot \boldsymbol{w}_i^A, \qquad \boldsymbol{w}_{\mathrm{eff},i}^B := \mathbf{f}_i + \mathbf{d}_i \odot \boldsymbol{w}_i^B \tag{192}$$

for the effective weights and

$$Z_i^A := (\boldsymbol{w}_{\mathrm{eff},i}^A)^\top \boldsymbol{x}, \qquad Z_i^B := (\boldsymbol{w}_{\mathrm{eff},i}^B)^\top \boldsymbol{x} \tag{193}$$

for the preactivation random variables under the input distribution. If $\|\mathbf{f}_i\|_{\boldsymbol{\Sigma}} = 0$, then (31) forces $\|\mathbf{d}_i \odot \boldsymbol{w}_i^A\|_{\boldsymbol{\Sigma}} = \|\mathbf{d}_i \odot \boldsymbol{w}_i^B\|_{\boldsymbol{\Sigma}} = 0$, hence $Z_i^A = Z_i^B = 0$ a.s. and $\xi_i(\lambda; \boldsymbol{x}) \equiv 0$; in this case the desired bound holds trivially for neuron $i$. Assume from now on that $\|\mathbf{f}_i\|_{\boldsymbol{\Sigma}} > 0$. Then, $Z_i^A \sim \mathcal{N}(0, (\sigma_i^A)^2)$, $Z_i^B \sim \mathcal{N}(0, (\sigma_i^B)^2)$ with $\sigma_i^A := \|\boldsymbol{w}_{\mathrm{eff},i}^A\|_{\boldsymbol{\Sigma}}$, $\sigma_i^B := \|\boldsymbol{w}_{\mathrm{eff},i}^B\|_{\boldsymbol{\Sigma}}$. Since $\beta < 1$, we have $\sigma_i^A, \sigma_i^B > 0$. Further, the correlation of $Z_i^A$ and $Z_i^B$ is

$$\rho_i := \frac{\langle \boldsymbol{w}_{\mathrm{eff},i}^A, \boldsymbol{w}_{\mathrm{eff},i}^B \rangle_{\boldsymbol{\Sigma}}}{\sigma_i^A \sigma_i^B}, \qquad \theta_i := \arccos(\rho_i). \tag{194}$$

For ReLU, one obtains

$$\xi_i(\lambda; \boldsymbol{x})^2 = \Big(\mathrm{ReLU}\big((1-\lambda)Z_i^A + \lambda Z_i^B\big) - \big((1-\lambda)\mathrm{ReLU}(Z_i^A) + \lambda\mathrm{ReLU}(Z_i^B)\big)\Big)^2 \tag{195}$$

$$= \min\big\{(1-\lambda)^2 (Z_i^A)^2, \lambda^2 (Z_i^B)^2\big\} \mathbb{1}\{Z_i^A Z_i^B \le 0\} \tag{196}$$

$$\le (1-\lambda)^2 (Z_i^A)^2 \mathbb{1}\{Z_i^A Z_i^B \le 0\} + \lambda^2 (Z_i^B)^2 \mathbb{1}\{Z_i^A Z_i^B \le 0\}. \tag{197}$$

For centered bivariate Gaussian random variables with variances $(\sigma_i^A)^2, (\sigma_i^B)^2$ and correlation $\rho_i = \cos(\theta_i)$,

$$\mathbb{E}\big[(Z_i^A)^2 \mathbb{1}\{Z_i^A Z_i^B \le 0\}\big] = (\sigma_i^A)^2 \frac{\theta_i - \sin(\theta_i)\cos(\theta_i)}{\pi}, \tag{198}$$

and likewise

$$\mathbb{E}\big[(Z_i^B)^2 \mathbb{1}\{Z_i^A Z_i^B \le 0\}\big] = (\sigma_i^B)^2 \frac{\theta_i - \sin(\theta_i)\cos(\theta_i)}{\pi}. \tag{199}$$

Indeed, if $(G_1, G_2) \sim \mathcal{N}(0, \boldsymbol{I}_2)$, then

$$(Z_i^A/\sigma_i^A, Z_i^B/\sigma_i^B) \overset{d}{=} (G_1, \cos(\theta_i)G_1 + \sin(\theta_i)G_2), \tag{200}$$

where $\theta_i = \arccos(\rho_i) \in [0, \pi]$. Since (198) only depends on the joint distribution, we may compute it under this representation. Write $(G_1, G_2) = R(\cos\varphi, \sin\varphi)$ in polar coordinates. Since the standard Gaussian distribution is invariant under rotations, $\varphi \sim \mathrm{Unif}\big([0, 2\pi)\big)$ is independent of $R$, and $\mathbb{E}[R^2] = \mathbb{E}[G_1^2 + G_2^2] = 2$. In polar coordinates, we have

$$Z_i^A/\sigma_i^A = R\cos(\varphi), \tag{201}$$

$$Z_i^B/\sigma_i^B = \cos(\theta_i)R\cos(\varphi) + \sin(\theta_i)R\sin(\varphi) = R\cos(\varphi - \theta_i), \tag{202}$$

where the last equality uses $\cos(\varphi - \theta_i) = \cos(\varphi)\cos(\theta_i) + \sin(\varphi)\sin(\theta_i)$. Thus, since $\sigma_i^A, \sigma_i^B > 0$ and $R > 0$ a.s.,

$$Z_i^A Z_i^B \le 0 \iff \sigma_i^A \sigma_i^B R^2 \cos(\varphi)\cos(\varphi - \theta_i) \le 0 \tag{203}$$

$$\iff \cos(\varphi)\cos(\varphi - \theta_i) \le 0. \tag{204}$$

For $\theta_i \in [0, \pi]$, this holds exactly for

$$\varphi \in \left[\frac{\pi}{2}, \frac{\pi}{2} + \theta_i\right] \cup \left[\frac{3\pi}{2}, \frac{3\pi}{2} + \theta_i\right], \tag{205}$$

with endpoints modulo $2\pi$. Therefore,

$$\mathbb{E}\big[(Z_i^A)^2\mathbb{1}\{Z_i^A Z_i^B \leq 0\}\big] = (\sigma_i^A)^2\,\mathbb{E}\big[R^2\cos^2(\varphi)\mathbb{1}\{\cos(\varphi)\cos(\varphi-\theta_i)\leq 0\}\big] \tag{206}$$

$$= (\sigma_i^A)^2\,\underbrace{\mathbb{E}\big[R^2\big]}_{=2}\,\mathbb{E}\big[\cos^2(\varphi)\mathbb{1}\{\cos(\varphi)\cos(\varphi-\theta_i)\leq 0\}\big] \tag{207}$$

$$= (\sigma_i^A)^2\,\frac{2}{2\pi}\int_{\{\cos(\varphi)\cos(\varphi-\theta_i)\leq 0\}}\cos^2(\varphi)\,\mathrm{d}\varphi \tag{208}$$

$$= (\sigma_i^A)^2\,\frac{2}{\pi}\int_{\pi/2}^{\pi/2+\theta_i}\cos^2(\varphi)\,\mathrm{d}\varphi \tag{209}$$

$$= (\sigma_i^A)^2\,\frac{\theta_i - \sin(\theta_i)\cos(\theta_i)}{\pi}, \tag{210}$$

where the last equality follows from $\int\cos^2(\varphi)\,\mathrm{d}\varphi = \varphi/2 + \sin(2\varphi)/4$ and $\sin(2\theta_i) = 2\sin(\theta_i)\cos(\theta_i)$. Hence,

$$\|\xi_i(\lambda;\cdot)\|_{L^2(\mathbb{P})}^2 \leq \Big((1-\lambda)^2(\sigma_i^A)^2 + \lambda^2(\sigma_i^B)^2\Big)\frac{\theta_i - \sin(\theta_i)\cos(\theta_i)}{\pi}. \tag{211}$$

It remains to bound the size and angular terms in (211) using (31). For the size term, write $\boldsymbol{w}_{\mathrm{eff},i}^A = \mathbf{f}_i + \boldsymbol{r}_i^A$ and $\boldsymbol{w}_{\mathrm{eff},i}^B = \mathbf{f}_i + \boldsymbol{r}_i^B$, where $\boldsymbol{r}_i^A := \mathbf{d}_i \odot \boldsymbol{w}_i^A$ and $\boldsymbol{r}_i^B := \mathbf{d}_i \odot \boldsymbol{w}_i^B$. Then

$$\sigma_i^A = \|\mathbf{f}_i + \boldsymbol{r}_i^A\|_{\boldsymbol{\Sigma}} \leq \|\mathbf{f}_i\|_{\boldsymbol{\Sigma}} + \|\boldsymbol{r}_i^A\|_{\boldsymbol{\Sigma}} \leq (1+\beta)\|\mathbf{f}_i\|_{\boldsymbol{\Sigma}}, \tag{212}$$

and similarly $\sigma_i^B \leq (1+\beta)\|\mathbf{f}_i\|_{\boldsymbol{\Sigma}}$. Hence, since $(1-\lambda)^2 + \lambda^2 \leq 1$,

$$(1-\lambda)^2(\sigma_i^A)^2 + \lambda^2(\sigma_i^B)^2 \leq (1+\beta)^2\,\|\mathbf{f}_i\|_{\boldsymbol{\Sigma}}^2. \tag{213}$$

To bound $\theta_i$, define

$$\boldsymbol{u}_i^0 := \frac{\mathbf{f}_i}{\|\mathbf{f}_i\|_{\boldsymbol{\Sigma}}}, \qquad \boldsymbol{u}_i^A := \frac{\mathbf{f}_i + \boldsymbol{r}_i^A}{\|\mathbf{f}_i + \boldsymbol{r}_i^A\|_{\boldsymbol{\Sigma}}}, \qquad \boldsymbol{u}_i^B := \frac{\mathbf{f}_i + \boldsymbol{r}_i^B}{\|\mathbf{f}_i + \boldsymbol{r}_i^B\|_{\boldsymbol{\Sigma}}}. \tag{214}$$

These vectors are normalized in $\|\cdot\|_{\boldsymbol{\Sigma}}$, and $\rho_i = \langle\boldsymbol{u}_i^A,\boldsymbol{u}_i^B\rangle_{\boldsymbol{\Sigma}}$. Set

$$\eta_i^A := \frac{\|\boldsymbol{r}_i^A\|_{\boldsymbol{\Sigma}}}{\|\mathbf{f}_i\|_{\boldsymbol{\Sigma}}}, \qquad s_i^A := \frac{\langle\boldsymbol{r}_i^A,\mathbf{f}_i\rangle_{\boldsymbol{\Sigma}}}{\|\mathbf{f}_i\|_{\boldsymbol{\Sigma}}^2}. \tag{215}$$

Then $\eta_i^A \leq \beta$ and $s_i^A \geq -\eta_i^A$ by Cauchy-Schwarz. Since $\beta < 1$, this gives $1 + s_i^A > 0$, and

$$\langle\boldsymbol{u}_i^A,\boldsymbol{u}_i^0\rangle_{\boldsymbol{\Sigma}} = \frac{1 + s_i^A}{\sqrt{1 + 2s_i^A + (\eta_i^A)^2}}. \tag{216}$$

Moreover,

$$\frac{(1+s_i^A)^2}{1 + 2s_i^A + (\eta_i^A)^2} - \big(1 - (\eta_i^A)^2\big) = \frac{(s_i^A + (\eta_i^A)^2)^2}{1 + 2s_i^A + (\eta_i^A)^2} \geq 0. \tag{217}$$

Thus,

$$\langle\boldsymbol{u}_i^A,\boldsymbol{u}_i^0\rangle_{\boldsymbol{\Sigma}} \geq \sqrt{1 - (\eta_i^A)^2} \geq \sqrt{1 - \beta^2}. \tag{218}$$

The same argument gives $\langle\boldsymbol{u}_i^B,\boldsymbol{u}_i^0\rangle_{\boldsymbol{\Sigma}} \geq \sqrt{1-\beta^2}$. Now write $a_i^A := \langle\boldsymbol{u}_i^A,\boldsymbol{u}_i^0\rangle_{\boldsymbol{\Sigma}}$ and $a_i^B := \langle\boldsymbol{u}_i^B,\boldsymbol{u}_i^0\rangle_{\boldsymbol{\Sigma}}$. Since $\|\boldsymbol{u}_i^A - a_i^A\boldsymbol{u}_i^0\|_{\boldsymbol{\Sigma}}^2 = 1 - (a_i^A)^2$ and similarly for $B$, Cauchy-Schwarz gives

$$\rho_i = \langle\boldsymbol{u}_i^A,\boldsymbol{u}_i^B\rangle_{\boldsymbol{\Sigma}} = a_i^A a_i^B + \big\langle\boldsymbol{u}_i^A - a_i^A\boldsymbol{u}_i^0,\boldsymbol{u}_i^B - a_i^B\boldsymbol{u}_i^0\big\rangle_{\boldsymbol{\Sigma}} \tag{219}$$

$$\geq a_i^A a_i^B - \sqrt{1 - (a_i^A)^2}\sqrt{1 - (a_i^B)^2}. \tag{220}$$

The function

$$(a,b) \mapsto ab - \sqrt{1-a^2}\sqrt{1-b^2} \tag{221}$$

is increasing in each argument on $[0,1]^2$. Since $a_i^A, a_i^B \geq \sqrt{1-\beta^2}$, we obtain

$$\rho_i \geq (1-\beta^2) - \beta^2 = 1 - 2\beta^2. \tag{222}$$

Therefore,

$$\theta_i = \arccos(\rho_i) \leq \arccos(1 - 2\beta^2) = 2\arcsin(\beta) \leq \pi\beta. \tag{223}$$

Thus,

$$\frac{\theta_i - \sin(\theta_i)\cos(\theta_i)}{\pi} = \frac{1}{\pi}\int_0^{\theta_i} 2\underbrace{\sin^2(s)}_{\leq s^2}\, \mathrm{d}s \leq \frac{2}{3\pi}\theta_i^3 \leq \frac{2\pi^2}{3}\beta^3. \tag{224}$$

Combining (211), (213), and (224) yields

$$\|\xi_i(\lambda;\cdot)\|_{L^2(\mathbb{P})}^2 \leq \frac{2\pi^2}{3}(1+\beta)^2\beta^3\,\|\mathbf{f}_i\|_{\mathbf{\Sigma}}^2. \tag{225}$$

Summing over $i$ and using $\|\xi_{\boldsymbol{H}}(\lambda;\cdot)\|_{L^2(\mathbb{P};\mathbb{R}^m)}^2 = \sum_{i=1}^m \|\xi_i(\lambda;\cdot)\|_{L^2(\mathbb{P})}^2$ gives

$$\|\xi_{\boldsymbol{H}}(\lambda;\cdot)\|_{L^2(\mathbb{P};\mathbb{R}^m)}^2 \leq \frac{2\pi^2}{3}(1+\beta)^2\beta^3\,\|\mathbf{F}\mathbf{\Sigma}^{1/2}\|_F^2. \tag{226}$$

Finally, taking $\sqrt{\cdot}$ and the supremum over $\lambda \in [0,1]$ on the l.h.s. proves the claim. $\qquad\square$

**Proposition 5.2** (LMC bound by chord deviation). *Consider a dataset $\mathcal{X} \times \mathcal{Y} \ni (\boldsymbol{x}, \boldsymbol{y}) \sim \mathbb{P}$, a neural network $f_{\boldsymbol{\theta}} : \mathcal{X} \to \mathcal{Y} \subseteq \mathbb{R}^{d_{\mathrm{out}}}$, and define the population loss $\mathcal{L}(\boldsymbol{\theta}) := \mathbb{E}_{(\boldsymbol{x},\boldsymbol{y})\sim\mathbb{P}}[\ell(f_{\boldsymbol{\theta}}(\boldsymbol{x}), \boldsymbol{y})] < \infty$. Assume that for every $\boldsymbol{y}$, $\boldsymbol{z} \mapsto \ell(\boldsymbol{z}, \boldsymbol{y})$ is convex and $L_\ell$-Lipschitz w.r.t. $\|\cdot\|_2$. Fix two parameter vectors $\boldsymbol{\theta}^A, \boldsymbol{\theta}^B$ and define the linear path $\boldsymbol{\theta}(\lambda) := (1-\lambda)\boldsymbol{\theta}^A + \lambda\boldsymbol{\theta}^B$, $\lambda \in [0,1]$. Define further*

$$\xi_f(\lambda;\boldsymbol{x}) := f_{\boldsymbol{\theta}(\lambda)}(\boldsymbol{x}) - \big((1-\lambda)f_{\boldsymbol{\theta}^A}(\boldsymbol{x}) + \lambda f_{\boldsymbol{\theta}^B}(\boldsymbol{x})\big). \tag{33}$$

*Then, for all $\lambda \in [0,1]$, we obtain*

$$\begin{aligned}&\mathcal{L}(\boldsymbol{\theta}(\lambda)) - \big((1-\lambda)\mathcal{L}(\boldsymbol{\theta}^A) + \lambda\mathcal{L}(\boldsymbol{\theta}^B)\big)\\ &\qquad \leq L_\ell\,\|\xi_f(\lambda;\cdot)\|_{L^2(\mathbb{P}_{\boldsymbol{x}},\mathbb{R}^{d_{\mathrm{out}}})}.\end{aligned} \tag{34}$$

*Proof.* Fix $\lambda \in [0,1]$. By convexity of $\boldsymbol{z} \mapsto \ell(\boldsymbol{z}, \boldsymbol{y})$, for every $(\boldsymbol{x}, \boldsymbol{y})$,

$$\ell\big((1-\lambda)f_{\boldsymbol{\theta}^A}(\boldsymbol{x}) + \lambda f_{\boldsymbol{\theta}^B}(\boldsymbol{x}), \boldsymbol{y}\big) \leq (1-\lambda)\ell(f_{\boldsymbol{\theta}^A}(\boldsymbol{x}), \boldsymbol{y}) + \lambda\ell(f_{\boldsymbol{\theta}^B}(\boldsymbol{x}), \boldsymbol{y}). \tag{227}$$

Moreover, by the definition of $\xi_f(\lambda, \boldsymbol{x})$ and since $\ell(\cdot, \boldsymbol{y})$ is $L_\ell$-Lipschitz w.r.t. $\|\cdot\|_2$,

$$\ell(f_{\boldsymbol{\theta}(\lambda)}(\boldsymbol{x}), \boldsymbol{y}) \leq \ell\big((1-\lambda)f_{\boldsymbol{\theta}^A}(\boldsymbol{x}) + \lambda f_{\boldsymbol{\theta}^B}(\boldsymbol{x}), \boldsymbol{y}\big) + L_\ell\,\|\xi_f(\lambda, \boldsymbol{x})\|_2. \tag{228}$$

Combining (227) and (228) and taking expectations w.r.t. $(\boldsymbol{x}, \boldsymbol{y}) \sim \mathbb{P}$ gives

$$\mathcal{L}(\boldsymbol{\theta}(\lambda)) \leq (1-\lambda)\mathcal{L}(\boldsymbol{\theta}^A) + \lambda\mathcal{L}(\boldsymbol{\theta}^B) + L_\ell\,\mathbb{E}_{\boldsymbol{x}\sim\mathbb{P}_{\boldsymbol{x}}}\big[\|\xi_f(\lambda, \boldsymbol{x})\|_2\big]. \tag{229}$$

Finally, by Jensen's inequality,

$$\mathbb{E}_{\boldsymbol{x}\sim\mathbb{P}_{\boldsymbol{x}}}\big[\|\xi_f(\lambda, \boldsymbol{x})\|_2\big] \leq \Big(\mathbb{E}_{\boldsymbol{x}\sim\mathbb{P}_{\boldsymbol{x}}}\big[\|\xi_f(\lambda, \boldsymbol{x})\|_2^2\big]\Big)^{1/2} = \|\xi_f(\lambda, \cdot)\|_{L^2(\mathbb{P}_{\boldsymbol{x}},\mathbb{R}^{d_{\mathrm{out}}})}. \tag{230}$$

Substituting this into (229) and rearranging yields the claim. $\qquad\square$

### D.2. Chord Deviation for Smooth Activations

We also record a version of Thm. 5.1 for smooth activation functions.

**Theorem D.1** (Chord deviation in center-dominated regime, $C^2$ activations). *Let $\boldsymbol{x} \sim \mathcal{N}(0, \mathbf{\Sigma})$ with $\mathbf{\Sigma} \succeq 0$, and let $\eta \in C^2(\mathbb{R})$ with $L_\eta := \sup_{z\in\mathbb{R}}|\eta''(z)| < \infty$. Fix $\boldsymbol{W}^A, \boldsymbol{W}^B \in \mathbb{R}^{m\times d}$. Suppose there exists $\beta \in [0,1)$ such that, for all $i \in [m]$,*

$$\|\mathbf{d}_i \odot \boldsymbol{w}_i^A\|_{\mathbf{\Sigma}} \leq \beta\|\mathbf{f}_i\|_{\mathbf{\Sigma}}, \qquad \|\mathbf{d}_i \odot \boldsymbol{w}_i^B\|_{\mathbf{\Sigma}} \leq \beta\|\mathbf{f}_i\|_{\mathbf{\Sigma}}. \tag{231}$$

*Then*

$$\sup_{\lambda\in[0,1]}\big\|\xi_{\boldsymbol{H}}(\lambda;\cdot)\big\|_{L^2(\mathbb{P};\mathbb{R}^m)} \leq \frac{\sqrt{3}}{2}L_\eta\beta^2\|\mathbf{F}\mathbf{\Sigma}^{1/2}\|_F^2. \tag{232}$$

*Proof.* Fix $i \in [m]$ and write $\boldsymbol{r}_i^A := \mathbf{d}_i \odot \boldsymbol{w}_i^A$ and $\boldsymbol{r}_i^B := \mathbf{d}_i \odot \boldsymbol{w}_i^B$. Set

$$Z_i^A := (\mathbf{f}_i + \boldsymbol{r}_i^A)^\top\boldsymbol{x}, \qquad Z_i^B := (\mathbf{f}_i + \boldsymbol{r}_i^B)^\top\boldsymbol{x}, \qquad \Delta Z_i := Z_i^B - Z_i^A. \tag{233}$$

For fixed $\boldsymbol{x}$, define $h_i(s) := \eta(Z_i^A + s\Delta Z_i)$ for $s \in [0,1]$. Then

$$\xi_i(\lambda;\boldsymbol{x}) = h_i(\lambda) - \big((1-\lambda)h_i(0) + \lambda h_i(1)\big). \tag{234}$$

We obtain

$$\xi_i(\lambda; \boldsymbol{x}) \; = \; (1-\lambda) \int_0^\lambda h_i'(s) \, ds - \lambda \int_\lambda^1 h_i'(s) \, ds \tag{235}$$

$$= \; (1-\lambda) \int_0^\lambda \left( h_i'(s) - h_i'(\lambda) \right) ds \; - \; \lambda \int_\lambda^1 \left( h_i'(s) - h_i'(\lambda) \right) ds. \tag{236}$$

Moreover, $h_i''(s) = \eta''(Z_i^A + s\Delta Z_i)(\Delta Z_i)^2$, and hence $|h_i''(s)| \leq L_\eta |\Delta Z_i|^2$ for all $s \in [0,1]$. Therefore,

$$|\xi_i(\lambda; \boldsymbol{x})| \; \leq \; L_\eta |\Delta Z_i|^2 \left( (1-\lambda) \int_0^\lambda (\lambda - s) \, ds + \lambda \int_\lambda^1 (s - \lambda) \, ds \right) \tag{237}$$

$$= \; \frac{L_\eta}{2} \lambda(1-\lambda)|\Delta Z_i|^2. \tag{238}$$

Thus

$$\|\xi_i(\lambda; \cdot)\|_{L^2(\mathbb{P})}^2 \; \leq \; \frac{L_\eta^2}{4} \lambda^2 (1-\lambda)^2 \mathbb{E}[(\Delta Z_i)^4]. \tag{239}$$

Since $\Delta Z_i = (\boldsymbol{r}_i^B - \boldsymbol{r}_i^A)^\top \boldsymbol{x}$ is centered Gaussian,

$$\mathbb{E}[(\Delta Z_i)^4] \; = \; 3\|\boldsymbol{r}_i^B - \boldsymbol{r}_i^A\|_{\boldsymbol{\Sigma}}^4. \tag{240}$$

By (231),

$$\|\boldsymbol{r}_i^B - \boldsymbol{r}_i^A\|_{\boldsymbol{\Sigma}} \; \leq \; \|\boldsymbol{r}_i^B\|_{\boldsymbol{\Sigma}} + \|\boldsymbol{r}_i^A\|_{\boldsymbol{\Sigma}} \; \leq \; 2\beta\|\mathbf{f}_i\|_{\boldsymbol{\Sigma}}. \tag{241}$$

Hence

$$\|\xi_i(\lambda; \cdot)\|_{L^2(\mathbb{P})}^2 \; \leq \; 12 \, L_\eta^2 \, \lambda^2(1-\lambda)^2 \beta^4 \|\mathbf{f}_i\|_{\boldsymbol{\Sigma}}^4. \tag{242}$$

Summing over $i$ gives

$$\left\| \xi_{\boldsymbol{H}}(\lambda; \cdot) \right\|_{L^2(\mathbb{P}; \mathbb{R}^m)}^2 \; \leq \; 12 \, L_\eta^2 \, \lambda^2(1-\lambda)^2 \beta^4 \sum_{i=1}^m \|\mathbf{f}_i\|_{\boldsymbol{\Sigma}}^4. \tag{243}$$

Taking $\sqrt{\cdot}$ and using $\lambda(1-\lambda) \leq 1/4$ yields

$$\sup_{\lambda \in [0,1]} \left\| \xi_{\boldsymbol{H}}(\lambda; \cdot) \right\|_{L^2(\mathbb{P}; \mathbb{R}^m)} \; \leq \; \frac{\sqrt{3}}{2} L_\eta \beta^2 \left( \sum_{i=1}^m \|\mathbf{f}_i\|_{\boldsymbol{\Sigma}}^4 \right)^{1/2}. \tag{244}$$

Finally,

$$\left( \sum_{i=1}^m \|\mathbf{f}_i\|_{\boldsymbol{\Sigma}}^4 \right)^{1/2} \; \leq \; \sum_{i=1}^m \|\mathbf{f}_i\|_{\boldsymbol{\Sigma}}^2 \; = \; \|\mathbf{F}\boldsymbol{\Sigma}^{1/2}\|_F^2, \tag{245}$$

which proves the claim. $\qquad\qquad\qquad\qquad\qquad\qquad\qquad\qquad\qquad\qquad\qquad\qquad\qquad\qquad\square$

# E. Approximate Subspace Support and Relaxed Realization Costs

In this section, we make rigorous how the exact subspace support model (Assumptions 3.1 and 4.1) can be interpreted as a low-rank approximation of a general data distribution: Namely, after projection on the top $k$ principal components, exact realizability in the projected model implies approximate realizability on the original distribution, with the error controlled by the discarded second-moment tail energy. In this sense, the results from § 4 should not only be read as applicable to this idealized model, but as a tractable approximation to the observable realizability structure when the input distribution to a layer is only approximately low-rank. We also explain a natural relaxed variant of the realization cost framework from § 4, which does not require exact realizability on the data support, and show that the hard realization cost from (8) arises as its ridgeless limit.

## E.1. Approximate Subspace Support

Let the input $\boldsymbol{x}$ to a layer be drawn from a distribution $\mathbb{P}$ on $\mathbb{R}^d$ with finite second moment $\mathbb{E}_{\boldsymbol{x}\sim\mathbb{P}}\|\boldsymbol{x}\|_2^2 < \infty$, and define

$$\boldsymbol{\Sigma} := \mathbb{E}_{\boldsymbol{x}\sim\mathbb{P}}[\boldsymbol{x}\boldsymbol{x}^\top] \succeq 0. \tag{246}$$

Let $\boldsymbol{\Sigma} = \boldsymbol{U}\boldsymbol{\Lambda}\boldsymbol{U}^\top$ be an eigendecomposition, i.e., $\boldsymbol{\Lambda} = \mathrm{Diag}(\lambda_1,\dots,\lambda_d)$ with $\lambda_1 \geq \dots \geq \lambda_d \geq 0$ and $\boldsymbol{U}^\top\boldsymbol{U} = \boldsymbol{I}_d$. For a target rank $k$, let $\boldsymbol{U}_k \in \mathbb{R}^{d\times k}$ be the top $k$ eigenvectors and define the tail energy

$$\tau_k := \sum_{j>k} \lambda_j. \tag{247}$$

In this context, Assumption 3.1 for a given $k$ means precisely that $\tau_k = 0$ with $\mathcal{U} := \mathrm{span}\,\boldsymbol{U}_k$, and the non-degeneracy assumption (Assumption 4.1) is equivalent to $\tau_{k-1} > 0$ for injective $\eta$. More generally, one can show the following:

**Lemma E.1** (Subspace support and second-moment tail energy). *Assume* $\mathbb{E}_{\boldsymbol{x}\sim\mathbb{P}}\|\boldsymbol{x}\|_2^2 < \infty$. *Then,* $\tau_k = 0$ *iff* $\mathbb{P}(\boldsymbol{x} \in \mathrm{im}(\boldsymbol{U}_k)) = 1$. *Moreover, setting* $\boldsymbol{P}_k := \boldsymbol{U}_k\boldsymbol{U}_k^\top$, *the mean-squared residual outside the top-$k$ subspace is* $\mathbb{E}_{\boldsymbol{x}\sim\mathbb{P}}\left[\left\|\boldsymbol{x} - \boldsymbol{P}_k\boldsymbol{x}\right\|_2^2\right] = \tau_k$.

*Proof.* One can directly calculate

$$\mathbb{E}_{\boldsymbol{x}\sim\mathbb{P}}\left[\|\boldsymbol{x} - \boldsymbol{P}_k\boldsymbol{x}\|_2^2\right] = \mathbb{E}_{\boldsymbol{x}\sim\mathbb{P}}[\boldsymbol{x}^\top(\boldsymbol{I} - \boldsymbol{P}_k)\boldsymbol{x}] = \mathrm{tr}((\boldsymbol{I} - \boldsymbol{P}_k)\boldsymbol{\Sigma}) = \sum_{j>k} \lambda_j = \tau_k. \tag{248}$$

If $\tau_k = 0$, then $\|\boldsymbol{x} - \boldsymbol{P}_k\boldsymbol{x}\|_2^2 \geq 0$ has expectation 0, hence $\boldsymbol{x} = \boldsymbol{P}_k\boldsymbol{x}$ a.s., which means $\boldsymbol{x} \in \mathrm{im}(\boldsymbol{U}_k)$ a.s. Conversely, if $\boldsymbol{x} \in \mathrm{im}(\boldsymbol{U}_k)$ a.s., then $\boldsymbol{x} = \boldsymbol{P}_k\boldsymbol{x}$ a.s., so $\tau_k = 0$. $\square$

Lem. E.1 shows that $\tau_k$ is the residual incurred by projecting $\mathbb{P}$ onto its top-$k$ principal subspace. This makes the projected distribution $\mathbb{P}_k := (\boldsymbol{P}_k)_*\mathbb{P}$ a natural low-rank surrogate for the original input distribution. One may therefore ask whether coincidence of neuron functions on this projected model still implies approximate coincidence on the original distribution. The next proposition answers this affirmatively: for Lipschitz neuron functions, the resulting $L^2(\mathbb{P})$ error can be controlled by the second-moment tail energy $\tau_k$. In this sense, the subspace support model from § 4 can also be seen as a tractable approximation whenever the input distribution is only approximately low-rank.

**Proposition E.2** (Projected coincidence implies approximate coincidence). *Assume* $\mathbb{E}_{\boldsymbol{x}\sim\mathbb{P}}\|\boldsymbol{x}\|_2^2 < \infty$. *Let* $\boldsymbol{h}_1, \boldsymbol{h}_2 : \mathbb{R}^d \to \mathbb{R}$ *be* $L_1$- *and* $L_2$-*Lipschitz, respectively, i.e.,* $|\boldsymbol{h}_i(\boldsymbol{x}) - \boldsymbol{h}_i(\boldsymbol{y})| \leq L_i\|\boldsymbol{x} - \boldsymbol{y}\|_2$ *for all* $\boldsymbol{x}, \boldsymbol{y} \in \mathbb{R}^d$, $i \in \{1,2\}$. *Assume that* $\boldsymbol{h}_1$ *and* $\boldsymbol{h}_2$ *coincide on the projected distribution* $\mathbb{P}_k$, *i.e.* $\boldsymbol{h}_1(\boldsymbol{P}_k\boldsymbol{x}) = \boldsymbol{h}_2(\boldsymbol{P}_k\boldsymbol{x})$ *for* $\mathbb{P}$-*a.e.* $\boldsymbol{x}$. *Then,*

$$\mathbb{E}_{\boldsymbol{x}\sim\mathbb{P}}\left[(\boldsymbol{h}_1(\boldsymbol{x}) - \boldsymbol{h}_2(\boldsymbol{x}))^2\right] \leq 2(L_1^2 + L_2^2)\,\tau_k. \tag{249}$$

This means that if $\tau_k$ is small, any two neuron functions that agree on the top-$k$ projected model are close in $L^2(\mathbb{P})$ on the original distribution.

*Proof.* Since $\boldsymbol{h}_1(\boldsymbol{P}_k\boldsymbol{x}) = \boldsymbol{h}_2(\boldsymbol{P}_k\boldsymbol{x})$ for $\mathbb{P}$-a.e. $\boldsymbol{x}$,

$$\boldsymbol{h}_1(\boldsymbol{x}) - \boldsymbol{h}_2(\boldsymbol{x}) = \big(\boldsymbol{h}_1(\boldsymbol{x}) - \boldsymbol{h}_1(\boldsymbol{P}_k\boldsymbol{x})\big) + \big(\boldsymbol{h}_2(\boldsymbol{P}_k\boldsymbol{x}) - \boldsymbol{h}_2(\boldsymbol{x})\big). \tag{250}$$

Using $(a+b)^2 \leq 2a^2 + 2b^2$, we obtain

$$\begin{aligned}
(\boldsymbol{h}_1(\boldsymbol{x}) - \boldsymbol{h}_2(\boldsymbol{x}))^2 &\leq 2\big(\boldsymbol{h}_1(\boldsymbol{x}) - \boldsymbol{h}_1(\boldsymbol{P}_k\boldsymbol{x})\big)^2 + 2\big(\boldsymbol{h}_2(\boldsymbol{x}) - \boldsymbol{h}_2(\boldsymbol{P}_k\boldsymbol{x})\big)^2 \\
&\leq 2(L_1^2 + L_2^2)\|\boldsymbol{x} - \boldsymbol{P}_k\boldsymbol{x}\|_2^2. 
\end{aligned} \tag{251}$$

Taking expectations and using Lem. E.1,

$$\mathbb{E}_{\boldsymbol{x}\sim\mathbb{P}}\big[(\boldsymbol{h}_1(\boldsymbol{x}) - \boldsymbol{h}_2(\boldsymbol{x}))^2\big] \;\leq\; 2(L_1^2 + L_2^2)\,\mathbb{E}_{\boldsymbol{x}\sim\mathbb{P}}\|\boldsymbol{x} - \boldsymbol{P}_k\boldsymbol{x}\|_2^2 \;=\; 2(L_1^2 + L_2^2)\tau_k, \tag{252}$$

which proves (249). $\qquad\qquad\qquad\qquad\qquad\qquad\qquad\qquad\qquad\qquad\qquad\qquad\qquad\qquad\qquad\qquad\qquad\quad\square$

In particular, if $\eta$ is $L$-Lipschitz and $\boldsymbol{h}_i(\boldsymbol{x}) = \eta(\boldsymbol{w}_i^\top \boldsymbol{x})$ for $\boldsymbol{w}_i \in \mathbb{R}^d$, $i \in \{1,2\}$, then each $\boldsymbol{h}_i$ is $L\|\boldsymbol{w}_i\|_2$-Lipschitz (by Cauchy-Schwarz). Hence, Prop. E.2 yields

$$\mathbb{E}_{\boldsymbol{x}\sim\mathbb{P}}\big[(\boldsymbol{h}_1(\boldsymbol{x}) - \boldsymbol{h}_2(\boldsymbol{x}))^2\big] \;\leq\; 2L^2(\|\boldsymbol{w}_1\|_2^2 + \|\boldsymbol{w}_2\|_2^2)\tau_k \tag{253}$$

whenever $\boldsymbol{h}_1(\boldsymbol{P}_k\boldsymbol{x}) = \boldsymbol{h}_2(\boldsymbol{P}_k\boldsymbol{x})$ for $\mathbb{P}$-a.e. $\boldsymbol{x}$.

### E.2. Relaxed Realization Costs

In the previous subsection, we have seen in what sense the subspace support model can be viewed as an approximation to a general input distribution. For realization costs, a complementary question is how one should formulate the cost when exact realizability is not imposed. A natural choice is to relax the hard constraint by penalizing squared $L^2(\mathbb{P})$ approximation error and weight norm. We record it here to show that the realization cost from § 4.1 arises as its ridgeless limit.

Just as in § 4, consider a single asymmetric layer in isolation, and define $\boldsymbol{H}(\boldsymbol{W};\boldsymbol{x}) := \eta((\mathbf{F} + \mathbf{D} \odot \boldsymbol{W})\boldsymbol{x}) \in \mathbb{R}^m$, where $\boldsymbol{W} \in \mathbb{R}^{m\times d}$, $\boldsymbol{x} \in \mathbb{R}^d$, $\mathbf{F}, \mathbf{D} \in \mathbb{R}^{m\times d}$ are fixed architectural components, and $\eta : \mathbb{R} \to \mathbb{R}$ acts elementwise.

**Definition E.3** (Relaxed $L^2$ realization cost)**.** Fix neuron $i$ and $\beta > 0$. For any $\boldsymbol{h} \in L^2(\mathbb{P})$, define

$$\|\boldsymbol{h}\|_{\mathcal{H}_i(\mathbb{P}),\beta}^2 \;=\; \|\boldsymbol{h}\|_{\mathcal{H}_i,\beta}^2 \;:=\; \inf_{\boldsymbol{w}\in\mathbb{R}^d}\Big\{\beta^{-1}\mathbb{E}_{\boldsymbol{x}\sim\mathbb{P}}\big[(\boldsymbol{h}(\boldsymbol{x}) - \boldsymbol{h}_i(\boldsymbol{w};\boldsymbol{x}))^2\big] + \|\boldsymbol{w}\|_2^2\Big\}. \tag{254}$$

In cases where the distribution $\mathbb{P}$ might vary, we will make the dependence explicit through writing $\mathcal{H}_i(\mathbb{P})$. It is straightforward to see that for Lipschitz $\eta$ and any $\boldsymbol{h} \in L^2(\mathbb{P})$, $\|\boldsymbol{h}\|_{\mathcal{H}_i,\beta}^2 < \infty$ is finite. A canonical family of targets we will look at are functions of the form $\boldsymbol{h}_{\boldsymbol{u}}(\boldsymbol{x}) := \eta(\boldsymbol{u}^\top \boldsymbol{x})$, $\boldsymbol{u} \in \mathbb{R}^d$, i.e., precisely *realizable* target features, since this is precisely the setting in § 4.

**Theorem E.4** (Explicit upper bound via a quadratic surrogate)**.** *Assume $\eta$ is $L$-Lipschitz. Fix neuron $i$ and define $\mathbf{D}_i := \text{Diag}(\mathbf{d}_i) \in \mathbb{R}^{d\times d}$. For any $\boldsymbol{u} \in \mathbb{R}^d$,*

$$\|\boldsymbol{h}_{\boldsymbol{u}}\|_{\mathcal{H}_i,\beta}^2 \;\leq\; \inf_{\boldsymbol{w}\in\mathbb{R}^d}\big\{\beta^{-1}L^2\,\|\boldsymbol{u} - \mathbf{f}_i - \mathbf{D}_i\boldsymbol{w}\|_{\boldsymbol{\Sigma}}^2 + \|\boldsymbol{w}\|_2^2\big\}. \tag{255}$$

*Moreover, the infimum in (255) has a unique minimizer*

$$\boldsymbol{w}_i^\star(\boldsymbol{u}) = \Big(\beta\,\boldsymbol{I} + L^2\,\mathbf{D}_i\boldsymbol{\Sigma}\mathbf{D}_i\Big)^{-1}L^2\,\mathbf{D}_i\boldsymbol{\Sigma}(\boldsymbol{u} - \mathbf{f}_i), \tag{256}$$

*and the corresponding minimized value equals*

$$\frac{L^2}{\beta}\|\boldsymbol{u} - \mathbf{f}_i\|_{\boldsymbol{\Sigma}}^2 - \frac{L^4}{\beta^2}\,(\boldsymbol{u} - \mathbf{f}_i)^\top\boldsymbol{\Sigma}\mathbf{D}_i\Big(\boldsymbol{I} + \frac{L^2}{\beta}\mathbf{D}_i\boldsymbol{\Sigma}\mathbf{D}_i\Big)^{-1}\mathbf{D}_i\boldsymbol{\Sigma}(\boldsymbol{u} - \mathbf{f}_i). \tag{257}$$

*Proof.* Fix $\boldsymbol{w} \in \mathbb{R}^d$ and abbreviate

$$\ell_{\boldsymbol{u}}(\boldsymbol{x}) \;:=\; \boldsymbol{u}^\top \boldsymbol{x}, \qquad \ell_i(\boldsymbol{w};\boldsymbol{x}) \;:=\; (\mathbf{f}_i + \mathbf{D}_i\boldsymbol{w})^\top \boldsymbol{x}. \tag{258}$$

Since $\eta$ is $L$-Lipschitz,

$$(\boldsymbol{h}_{\boldsymbol{u}}(\boldsymbol{x}) - \boldsymbol{h}_i(\boldsymbol{w};\boldsymbol{x}))^2 \;\leq\; L^2\,(\ell_{\boldsymbol{u}}(\boldsymbol{x}) - \ell_i(\boldsymbol{w};\boldsymbol{x}))^2. \tag{259}$$

Taking expectations yields

$$\mathbb{E}[(\boldsymbol{h}_{\boldsymbol{u}}(\boldsymbol{x}) - \boldsymbol{h}_i(\boldsymbol{w};\boldsymbol{x}))^2] \;\leq\; L^2\,\|\boldsymbol{u} - \mathbf{f}_i - \mathbf{D}_i\boldsymbol{w}\|_{\boldsymbol{\Sigma}}^2. \tag{260}$$

Plugging into Def. E.3 and then taking the infimum over $\boldsymbol{w}$ proves (255). To compute the infimum, define $\alpha := L^2/\beta$ and consider the strictly convex quadratic

$$J(\boldsymbol{w}) \;:=\; \|\boldsymbol{w}\|_2^2 + \alpha\,(\boldsymbol{u} - \mathbf{f}_i - \mathbf{D}_i\boldsymbol{w})^\top\boldsymbol{\Sigma}(\boldsymbol{u} - \mathbf{f}_i - \mathbf{D}_i\boldsymbol{w}). \tag{261}$$

Differentiating gives

$$\nabla J(\boldsymbol{w}) \;=\; 2\boldsymbol{w} - 2\alpha\,\mathbf{D}_i\boldsymbol{\Sigma}(\boldsymbol{u} - \mathbf{f}_i - \mathbf{D}_i\boldsymbol{w}) \;=\; 2\Big(\boldsymbol{I} + \alpha\,\mathbf{D}_i\boldsymbol{\Sigma}\mathbf{D}_i\Big)\boldsymbol{w} - 2\alpha\,\mathbf{D}_i\boldsymbol{\Sigma}(\boldsymbol{u} - \mathbf{f}_i). \tag{262}$$

Since $\boldsymbol{I} + \alpha\, \mathbf{D}_i \boldsymbol{\Sigma} \mathbf{D}_i \succ 0$, setting $\nabla J(\boldsymbol{w}) = 0$ yields the unique minimizer (256). It remains to compute the minimum value. Write $\boldsymbol{r} := \boldsymbol{u} - \mathbf{f}_i$ and $\boldsymbol{A} := \boldsymbol{I} + \alpha\, \mathbf{D}_i \boldsymbol{\Sigma} \mathbf{D}_i \succ 0$. Expanding $J$ yields

$$J(\boldsymbol{w}) \;=\; \boldsymbol{w}^\top \boldsymbol{A} \boldsymbol{w} - 2\alpha\, \boldsymbol{w}^\top \mathbf{D}_i \boldsymbol{\Sigma} \boldsymbol{r} + \alpha\, \boldsymbol{r}^\top \boldsymbol{\Sigma} \boldsymbol{r}. \tag{263}$$

Let $\boldsymbol{c} := \alpha\, \mathbf{D}_i \boldsymbol{\Sigma} \boldsymbol{r}$. Then, $J(\boldsymbol{w}) = \boldsymbol{w}^\top \boldsymbol{A} \boldsymbol{w} - 2\boldsymbol{w}^\top \boldsymbol{c} + \alpha\, \boldsymbol{r}^\top \boldsymbol{\Sigma} \boldsymbol{r}$. Since $\boldsymbol{A} \succ 0$, we have

$$\boldsymbol{w}^\top \boldsymbol{A} \boldsymbol{w} - 2\boldsymbol{w}^\top \boldsymbol{c} \;=\; (\boldsymbol{w} - \boldsymbol{A}^{-1}\boldsymbol{c})^\top \boldsymbol{A} (\boldsymbol{w} - \boldsymbol{A}^{-1}\boldsymbol{c}) - \boldsymbol{c}^\top \boldsymbol{A}^{-1}\boldsymbol{c}, \tag{264}$$

which implies that the minimum is attained at $\boldsymbol{w}^\star := \boldsymbol{A}^{-1}\boldsymbol{c}$ (which coincides with (256)) and equals

$$\min_{\boldsymbol{w} \in \mathbb{R}^d} J(\boldsymbol{w}) \;=\; \alpha\, \boldsymbol{r}^\top \boldsymbol{\Sigma} \boldsymbol{r} - \boldsymbol{c}^\top \boldsymbol{A}^{-1}\boldsymbol{c} \;=\; \alpha\, \boldsymbol{r}^\top \boldsymbol{\Sigma} \boldsymbol{r} - \alpha^2\, \boldsymbol{r}^\top \boldsymbol{\Sigma} \mathbf{D}_i (\boldsymbol{I} + \alpha\, \mathbf{D}_i \boldsymbol{\Sigma} \mathbf{D}_i)^{-1} \mathbf{D}_i \boldsymbol{\Sigma} \boldsymbol{r}, \tag{265}$$

which is exactly (257). $\qquad \square$

If $\eta$ is bi-Lipschitz, one can prove a lower bound in a similar way. We will now show that the $\beta \to 0$ (ridgeless) limit recovers the hard realization cost from § 4.1, where non-realizable targets have infinite cost.

**Theorem E.5** (Ridgeless limit). *Assume $\eta$ is Lipschitz and $\mathbb{E}_{\boldsymbol{x} \sim \mathbb{P}} \|\boldsymbol{x}\|_2^2 < \infty$. Fix neuron $i$ and let*

$$\|\boldsymbol{h}\|_{\mathcal{H}_i(\mathbb{P})}^2 \;=\; \|\boldsymbol{h}\|_{\mathcal{H}_i}^2 \;:=\; \inf\left\{ \|\boldsymbol{w}\|_2^2 \;\middle|\; \mathbb{E}_{\boldsymbol{x} \sim \mathbb{P}}\big[(\boldsymbol{h}(\boldsymbol{x}) - \boldsymbol{h}_i(\boldsymbol{w}; \boldsymbol{x}))^2\big] = 0 \right\} \in [0, \infty] \tag{266}$$

*with the convention $\inf \varnothing := +\infty$. Then for any $\boldsymbol{h} \in L^2(\mathbb{P})$,*

$$\lim_{\beta \to 0} \|\boldsymbol{h}\|_{\mathcal{H}_i, \beta}^2 \;=\; \|\boldsymbol{h}\|_{\mathcal{H}_i}^2. \tag{267}$$

*Proof.* Write $f(\boldsymbol{w}) := \mathbb{E}[(\boldsymbol{h}(\boldsymbol{x}) - \boldsymbol{h}_i(\boldsymbol{w}; \boldsymbol{x}))^2] \geq 0$ and $\phi(\beta) := \inf_{\boldsymbol{w}}\{f(\boldsymbol{w}) + \beta\|\boldsymbol{w}\|_2^2\}$ so that $\|\boldsymbol{h}\|_{\mathcal{H}_i, \beta}^2 = \beta^{-1}\phi(\beta)$ by Def. E.3.

First, suppose $\|\boldsymbol{h}\|_{\mathcal{H}_i}^2 = +\infty$. We show $\|\boldsymbol{h}\|_{\mathcal{H}_i, \beta}^2 \to +\infty$. Assume for contradiction that there exists $0 < \beta_n \to 0$ and $C < \infty$ with $\|\boldsymbol{h}\|_{\mathcal{H}_i, \beta_n}^2 \leq C$ for all $n$. Then there exist $\boldsymbol{w}_n$ such that

$$f(\boldsymbol{w}_n) + \beta_n \|\boldsymbol{w}_n\|_2^2 \;\leq\; \beta_n(C + 1). \tag{268}$$

In particular, this implies that $\|\boldsymbol{w}_n\|_2^2 \leq C + 1$ and $f(\boldsymbol{w}_n) \to 0$ as $n \to \infty$. Since $(\boldsymbol{w}_n)_n$ is bounded, we can extract a convergent subsequence $(\boldsymbol{w}_{n_k})_k$ with $\boldsymbol{w}_{n_k} \to \boldsymbol{w}^*$ as $k \to \infty$. By Lipschitzness of $\eta$ and $\mathbb{E}\|\boldsymbol{x}\|_2^2 < \infty$, the map $\boldsymbol{w} \mapsto \boldsymbol{h}_i(\boldsymbol{w}; \cdot)$ is continuous from $(\mathbb{R}^d, \|\cdot\|_2)$ to $L^2(\mathbb{P})$, hence $f(\boldsymbol{w}_{n_k}) \to f(\boldsymbol{w}^*)$ and therefore $f(\boldsymbol{w}^*) = 0$. This contradicts $\|\boldsymbol{h}\|_{\mathcal{H}_i}^2 = +\infty$.

Assume now $\|\boldsymbol{h}\|_{\mathcal{H}_i}^2 < \infty$. Then, for every $\beta > 0$,

$$\phi(\beta) \;=\; \inf_{\boldsymbol{w}}\{f(\boldsymbol{w}) + \beta\|\boldsymbol{w}\|_2^2\} \;\leq\; \inf_{\boldsymbol{w}: f(\boldsymbol{w})=0}\{\beta\|\boldsymbol{w}\|_2^2\} \;=\; \beta\, \|\boldsymbol{h}\|_{\mathcal{H}_i}^2, \tag{269}$$

and

$$\limsup_{\beta \to 0} \beta^{-1}\phi(\beta) \;\leq\; \|\boldsymbol{h}\|_{\mathcal{H}_i}^2. \tag{270}$$

For the reverse inequality, let $0 < \beta_n \to 0$ and pick $\boldsymbol{w}_n$ such that $f(\boldsymbol{w}_n) + \beta_n \|\boldsymbol{w}_n\|_2^2 \leq \phi(\beta_n) + \beta_n/n$. By the previous upper bound, $\phi(\beta_n) \leq \beta_n(\|\boldsymbol{h}\|_{\mathcal{H}_i}^2 + 1)$ for all large enough $n$, hence $(\|\boldsymbol{w}_n\|_2)_n$ is bounded and $f(\boldsymbol{w}_n) \to 0$. Similar as above, any limit point $\boldsymbol{w}^*$ of a convergent subsequence satisfies $f(\boldsymbol{w}^*) = 0$ as $n \to \infty$. Therefore $\|\boldsymbol{w}^*\|_2^2 \geq \|\boldsymbol{h}\|_{\mathcal{H}_i}^2$ and thus $\liminf_{n \to \infty} \|\boldsymbol{w}_n\|_2^2 \geq \|\boldsymbol{h}\|_{\mathcal{H}_i}^2$. Since $\beta_n^{-1}\phi(\beta_n) \geq \|\boldsymbol{w}_n\|_2^2 - 1/n$, we can conclude

$$\liminf_{\beta \to 0} \beta^{-1}\phi(\beta) \;\geq\; \|\boldsymbol{h}\|_{\mathcal{H}_i}^2. \tag{271}$$

Combining (270) and (271) gives the claim. $\qquad \square$

When $\eta = \mathrm{id}$, the relaxed cost in Def. E.3 reduces to a strictly convex quadratic ridge objective in $\boldsymbol{w}$, so the minimizer and optimum admit explicit closed forms for every $\beta > 0$. For general nonlinear $\eta \neq \mathrm{id}$, the same objective is typically nonconvex in $\boldsymbol{w}$ and no simple closed-form solution is available, so apart from the ridgeless limit, one would have to work with tractable surrogates such as Thm. E.4. By contrast, in the ridgeless limit the problem simplifies conceptually: Thm. E.5 shows that as $\beta \to 0$, one recovers the realization cost from § 4.1, which admits the explicit closed form (10). At the same time, because the limit enforces exact realizability, it is singular under perturbations of the *support*, i.e., even a small amount of mass outside the top-$k$ subspace can destroy feasibility on the original distribution and make the hard cost jump to $+\infty$.

For fixed $\beta > 0$, however, the relaxed objective remains stable under such perturbations. The next corollary records the corresponding quantitative upper bound.

**Corollary E.6** (Projected realizability yields small relaxed cost). *Fix neuron $i$ and $\beta > 0$, and let $\boldsymbol{h} \in L^2(\mathbb{P})$. Assume that the hard realization cost of $\boldsymbol{h}$ with respect to $\mathbb{P}_k$ is finite and attained at some $\boldsymbol{w}^* \in \mathbb{R}^d$, i.e.*

$$\boldsymbol{h}(\boldsymbol{P}_k\boldsymbol{x}) = \boldsymbol{h}_i(\boldsymbol{w}^*; \boldsymbol{P}_k\boldsymbol{x}) \quad \text{for } \mathbb{P}\text{-a.e. } \boldsymbol{x}, \qquad \|\boldsymbol{w}^*\|_2 = \|\boldsymbol{h}\|_{\mathcal{H}_i(\mathbb{P}_k)}. \tag{272}$$

*Assume also that $\boldsymbol{h}$, $\boldsymbol{h}_i(\boldsymbol{w}^*; \cdot)$ are $L_{\boldsymbol{h}}$- and $L_i(\boldsymbol{w}^*)$-Lipschitz, respectively. Then,*

$$\|\boldsymbol{h}\|^2_{\mathcal{H}_i(\mathbb{P}), \beta} \;\leq\; \|\boldsymbol{h}\|^2_{\mathcal{H}_i(\mathbb{P}_k)} + 2(L_{\boldsymbol{h}}^2 + L_i(\boldsymbol{w}^*)^2)\,\beta^{-1}\tau_k. \tag{273}$$

*Proof.* By Def. E.3, evaluating the infimum at $\boldsymbol{w}^*$ and applying Prop. E.2 gives

$$\|\boldsymbol{h}\|^2_{\mathcal{H}_i(\mathbb{P}), \beta} \;\leq\; \beta^{-1}\mathbb{E}_{\boldsymbol{x}\sim\mathbb{P}}\big[(\boldsymbol{h}(\boldsymbol{x}) - \boldsymbol{h}_i(\boldsymbol{w}^*; \boldsymbol{x}))^2\big] + \|\boldsymbol{w}^*\|_2^2 \;\leq\; 2(L_{\boldsymbol{h}}^2 + L_i(\boldsymbol{w}^*)^2)\,\beta^{-1}\tau_k + \|\boldsymbol{w}^*\|_2^2, \tag{274}$$

which yields the claim since $\|\boldsymbol{w}^*\|_2 = \|\boldsymbol{h}\|_{\mathcal{H}_i(\mathbb{P}_k)}$. $\qquad\square$

Corollary E.6 links the approximate subspace support from § E.1 and the relaxed realization costs we introduced in § E.2. Namely, the first term on the r.h.s. of (273) is the hard realization cost on the projected model $\mathbb{P}_k$. The second term is the approximation error incurred when passing back to the original distribution. Thus, for fixed $\beta > 0$, the projected subspace support model yields a quantitative upper bound on the relaxed realization cost under the true distribution.

# F. Experimental Setup

## F.1. MLPs

For MLP experiments, we use MLPs with 4 layers, input dimension 784 (MNIST input dimension), hidden dimensions 512, layer norms, and GELU activation functions. For $\boldsymbol{W}$-asymmetric MLPs ($\boldsymbol{W}$-MLPs), following Lim et al. (2024b), we fix 64 entries per linear layer row (= neuron) for the first three layers, and 256 entries per row for the final layer. Fixed entries are randomly sampled, making use of a mask seed that ensures the same mask in different model instances. Whenever we perform parameter interpolation between models, the models have the same mask (which is considered a fixed part of the architecture).

For the fixed weight scale $\sigma_{\mathbf{F}}$, we use $\sigma_{\mathbf{F}} = 1$ as the default choice for effective symmetry breaking $\boldsymbol{W}$-MLPs, and $\sigma_{\mathbf{F}} = 0$ for the variant that does not exhibit effective symmetry breaking, as well as a range of values for $\sigma_{\mathbf{F}}$ between 0.0 and 5.0 for one $\sigma_{\mathbf{F}}$ sweep experiment. In contrast to Lim et al. (2024b), we always apply the same value of $\sigma_{\mathbf{F}}$ at all layers.

We train the MLPs, $\boldsymbol{W}$-MLPs, and *syre*-MLPs on MNIST for 100 epochs with AdamW (Loshchilov & Hutter, 2019), a learning rate of 0.001, and a weight decay of 0.01.

## F.2. ResNets

For our ResNet experiments, we use ResNet20 networks with $8\times$ width multiplier (following the setup of Lim et al. (2024b)), ReLU activation functions, and group norms. The architecture contains an initial convolution ($3 \rightarrow 128$ channels), three block groups ($128$, $128 \rightarrow 256$, $256 \rightarrow 512$ channels), and a final linear layer. For $\boldsymbol{W}$-asymmetric ResNets, in each convolution, a number of fixed weights are introduced per input channel (spread randomly over the output channels and kernel dimensions). The number of fixed parameters varies per block group, with the shortcut convolution in block groups 2 and 3 having a separate value for the number of fixed values. Table 3 gives the values across the network.

*Table 3.* Number of fixed parameters per layer.

| Layer | # fixed |
|---|---|
| Initial convolution | 12 |
| Block group 1 | 108 |
| Block group 2 | 162 |
| Shortcut | 18 |
| Block group 3 | 216 |
| Shortcut | 24 |
| Final linear | 24 |

The default fixed weight scale in our $\boldsymbol{W}$-ResNet experiments is $\sigma_{\mathbf{F}} = 2$ for the effective symmetry breaking setting, and $\sigma_{\mathbf{F}} = 0$ for the ineffective symmetry breaking setting. We train the ResNets on CIFAR-10 for 100 epochs using AdamW (Loshchilov & Hutter, 2019) with weight decay of 0.01, and learning rate warm up from 0.0001 to 0.01 in the first 20 epochs ($\gamma = 1.259$).

## F.3. Datasets and Augmentations

We use MNIST and CIFAR-10 datasets with a random 80%/10%/10% train/val/test split. On MNIST, we only use a normalization transform. On CIFAR-10, we additionally use a RandomHorizontalFlip transform, and a RandomCrop transform with 4 pixels of padding and a 32x32 output size.

## F.4. Activation Matching

We implement activation matching as described in § 6. For MLPs, this is straightforward: we forward the full dataset through the models and record intermediate activations $\boldsymbol{Z}^A, \boldsymbol{Z}^B$ at each layer, which we use to compute the similarity matrix $\boldsymbol{S}^{\text{act}}$. For ResNets, some subtleties are involved. The permutable dimension is the convolution channel dimension, which takes the role of the hidden representation dimension in $\boldsymbol{Z}$. We could average over the pixel dimensions to have $\boldsymbol{Z} \in \mathbb{R}^{n \times d}$. However we find it beneficial for LMC to treat each pixel as an independent data sample, producing $\boldsymbol{Z} \in \mathbb{R}^{(nk^2) \times d}$. Furthermore, in ResNets, the residual connections cause certain activation permutations to affect several layer inputs: For example, the input

to block 1 in block group 1 is also indirectly fed into block 2 via the additive residual connection, without anything yielding an independent permutation symmetry in between. Each block group exhibits one such a global permutation that affects several layer inputs, and hence should be estimated jointly, yielding $\boldsymbol{Z} \in \mathbb{R}^{(nk^2c) \times d}$ (for $c = 4$ in our case). Besides, each block admits an inner permutation symmetry between its two convolution layers, which can be estimated independently. Due to memory constraints, we perform activation matching on 10% of our CIFAR-10 training set, which is sufficient to yield an alignment of low barrier. For our plots in § 6, we average the activation matching objective across all layers.

### F.5. Coherence Experiment

We record details for the experiment isolating dependence on the coherence of the data subspace (see Fig. 6) in a teacher-student setup. The latent input variable is $(a, b) \sim \mathcal{N}(0, \boldsymbol{I}_2)$ and we aim to learn the teacher $f(a, b) := \mathrm{ReLU}(a + b)$. We sample 1024 training examples and 2048 test examples from this latent distribution, which are used across all coherence levels. The latent $(a, b)$ is embedded into the ambient space $\mathbb{R}^d$, $d = 20$, as

$$\boldsymbol{x} = \boldsymbol{U}_k(a, b)^\top, \qquad \boldsymbol{U}_k \in \mathbb{R}^{d \times 2}, \qquad \boldsymbol{U}_k^\top \boldsymbol{U}_k = \boldsymbol{I}_2, \tag{275}$$

where $k \in \{1, \ldots, 10\}$ controls the active ambient support. For the "exact-copy" family, all rows of $\boldsymbol{U}_k$ are zero except

$$U_{2r-1,1} = \frac{1}{\sqrt{k}}, \qquad U_{2r,2} = \frac{1}{\sqrt{k}}, \qquad r = 1, \ldots, k. \tag{276}$$

I.e., each latent coordinate is copied uniformly over $k$ ambient coordinates. For the "random-frame" family, we sample a permutation $\pi$ and angles $\theta_1, \ldots, \theta_{d/2} \sim \mathrm{Unif}(0, \pi)$. These sampled values are reused for all $k$. For a given $k$, all rows of $\boldsymbol{U}_k$ are zero except, for $r = 1, \ldots, k$,

$$\boldsymbol{U}_{\pi(2r-1),:} = \frac{1}{\sqrt{k}}(\cos \theta_r, \sin \theta_r), \qquad \boldsymbol{U}_{\pi(2r),:} = \frac{1}{\sqrt{k}}(-\sin \theta_r, \cos \theta_r). \tag{277}$$

The student is a one-hidden-layer asymmetric ReLU network of width 128 with $\mathbf{F} = 0$ and $\mathbf{D} \sim \mathrm{Bernoulli}(0.15)$ i.i.d. Further, the output weights are fixed to 1. Hidden weights are initialized independently from $\mathcal{N}(0, 0.05^2)$ and hidden biases are initialized to zero. Each model is trained for 100 epochs on MSE on a batch size of 512 using AdamW (Loshchilov & Hutter, 2019) (learning rate $10^{-2}$, weight decay $10^{-2}$). For each $k$ and subspace embedding, we sample 5 outer seeds for $\mathbf{D}$, and 10 initializations of student pairs for interpolation each, yielding in total 50 interpolation pairs per $k$ and subspace embedding. For each $k$ and subspace embedding, the reported midpoint barrier is the mean over the 50 pair-level midpoint barriers ± standard deviation over these 50 values.

### F.6. Gaussian Mixture Synthetic Dataset

In our experiments (**Q3** in § 6, and § G.2, § G.3 in the appendix) we use a synthetic Gaussian mixture model dataset that allows us to control the intrinsic data dimension parameter $k$. We sample class centers $\boldsymbol{\mu}_c \in \mathbb{R}^k$, $(\boldsymbol{\mu}_c)_i \sim \mathcal{N}(0, \sigma_{\mathrm{sep}}^2)$, where $\sigma_{\mathrm{sep}}$ is a class separation parameter. We generate $k$-dimensional class $k$ latent samples as

$$\boldsymbol{z}_i = \boldsymbol{Q}_c(\boldsymbol{\mu}_c + \boldsymbol{\varepsilon}_i) \in \mathbb{R}^k, \tag{278}$$

where $\boldsymbol{\varepsilon}_i \sim \mathcal{N}(0, \mathrm{Diag}(s_1^2, \ldots, s_k^2))$ is anisotropic Gaussian noise (with direction-wise scaling vector $\boldsymbol{s} \in \mathbb{R}^k$) and $\boldsymbol{Q}_c$ is a class-specific orthogonal transformation, and embed them into the ambient space as $\boldsymbol{x}_i = \boldsymbol{P}\boldsymbol{z}_i \in \mathbb{R}^d$, where $\boldsymbol{P} \in \mathbb{R}^{d \times k}$. We generate $50\,000$ training samples using five classes in an ambient space of dimension $d = 128$, varying the intrinsic dimension $k$ depending on the experiment.

### F.7. Empirical Neuron Realization and Swap Costs

We estimate neuron realization costs in trained models, and pairwise neuron swap costs derived from them (**Q3** in § 6, and § G.3 in the appendix) using two methods: Using the Mahalanobis distance of (10), and using a ridge ridge regression setup (see Def. E.3).

For the Mahalanobis distance estimation, we record activations during a forward pass on the whole dataset, and use them to estimate an input subspace for each layer as a PCA subspace that explains at least 90% of the variance. We then follow equations (6), (9) and (10) to compute per neuron the induced subspace preactivation coefficient $\boldsymbol{a}$, the Gram matrix $\mathbf{S}_i$, and finally the realization cost (Mahalanobis distance) under the subspace support model.

For the ridge regression realization cost estimation, we optimize (254) for $15\,000$ iterations using AdamW (learning rate 0.005, no weight decay). We sweep $\beta \in \{0.001, 0.01, 0.1, 1.0\}$, and report results for $\beta = 0.01$. For the ridge regression

experiments, we re-train models using batch norm instead of layer norm, consider the neuron function to include the normalization and activation function, but leave the batch norm parameters fixed. The optimization for neuron $i$'s realization cost with respect to neuron $j$'s class starts at neuron $i$'s trained weights.

# G. Additional Experimental Results

## G.1. Layerwise Activation Matching Objectives

In addition to Fig. 3, we also report activation matching objectives per layer. In Fig. 7, we plot layerwise activation matching objectives by epoch for MLPs on MNIST. This reveals that for all observed cases, the patterns are consistent across layers and, therefore, the aggregate over all layers from Fig. 3 is informative.

We also report activation matching objectives for $\sigma$-asymmetric networks (Lim et al., 2024b). This different symmetry breaking intervention operates by replacing the nonlinearity with one that does not act elementwise. This specific approach has been found to not break symmetries as effectively as $W$-asym. in the context of unaligned LMC. While $\sigma$-asym. networks do not fit into our framework of (2), one can still perform activation matching to qualitatively analyze neuron identifiability. Fig. 7 reveals that, in fact, the identity does not produce a higher objective than random permutations, and therefore, similar to $W$-asym. networks with $\sigma_{\mathbf{F}} = 0$, symmetries are not broken effectively.

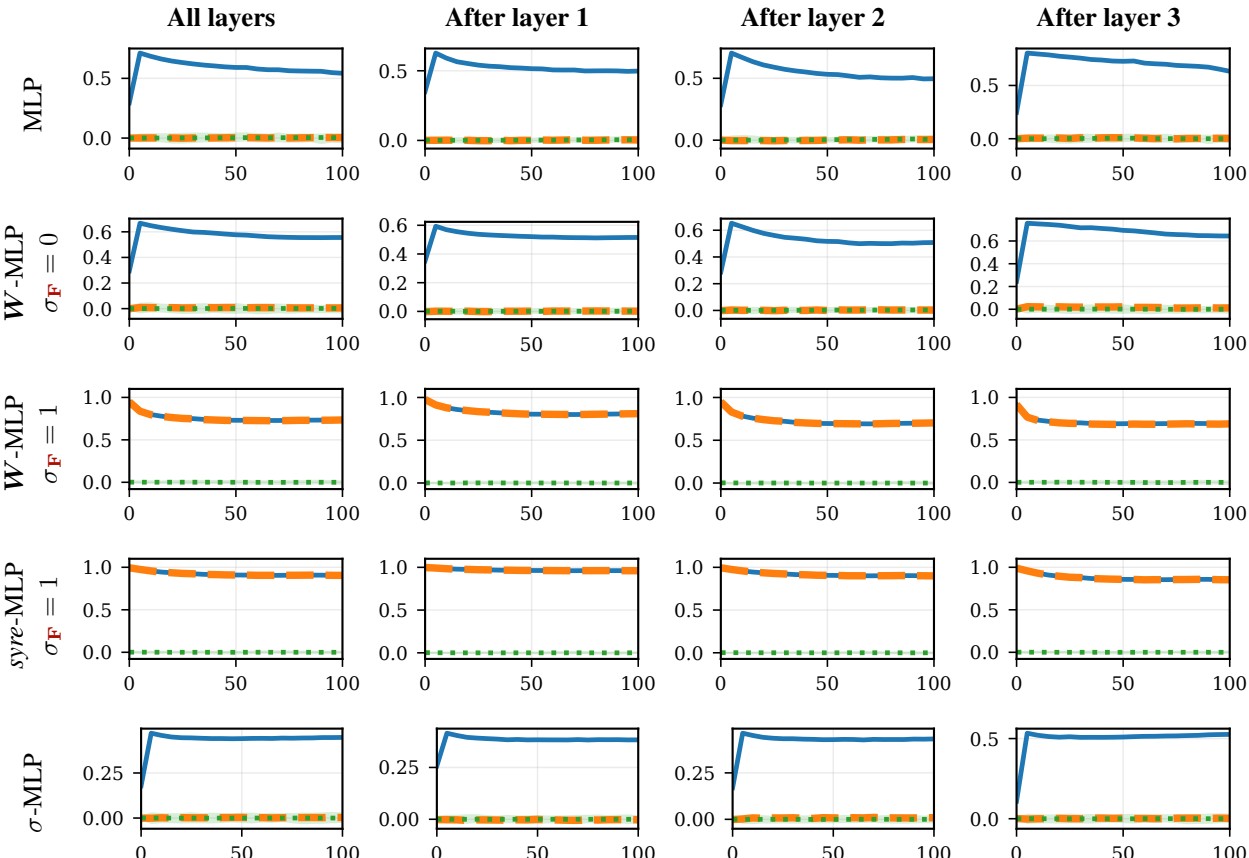

*Figure 7.* Layerwise activation matching objectives over epochs, separated by layer and by different symmetry-breaking architectures (including the $\sigma$-MLP variant, see Lim et al. (2024b)). We plot optimal, identity, and random permutations.

Further, in Figs. 8 and 9, we plot layerwise versions of the $\sigma_{\mathbf{F}}$ sweep for both MLPs on MNIST and ResNets on CIFAR-10 respectively. We also sweep the sparsity (proportion of fixed weights) in Fig. 10 to measure its effect on symmetry breaking, and find that sparsity alone without fixed weights does not yield strong neuron identifiability.

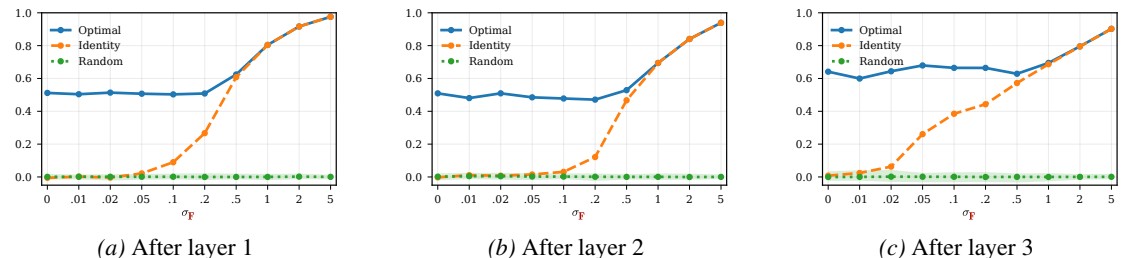

*Figure 8.* Activation matching objectives per layer ($W$-MLP on MNIST), sweeping over fixed weight scale $\sigma_{\mathbf{F}}$.

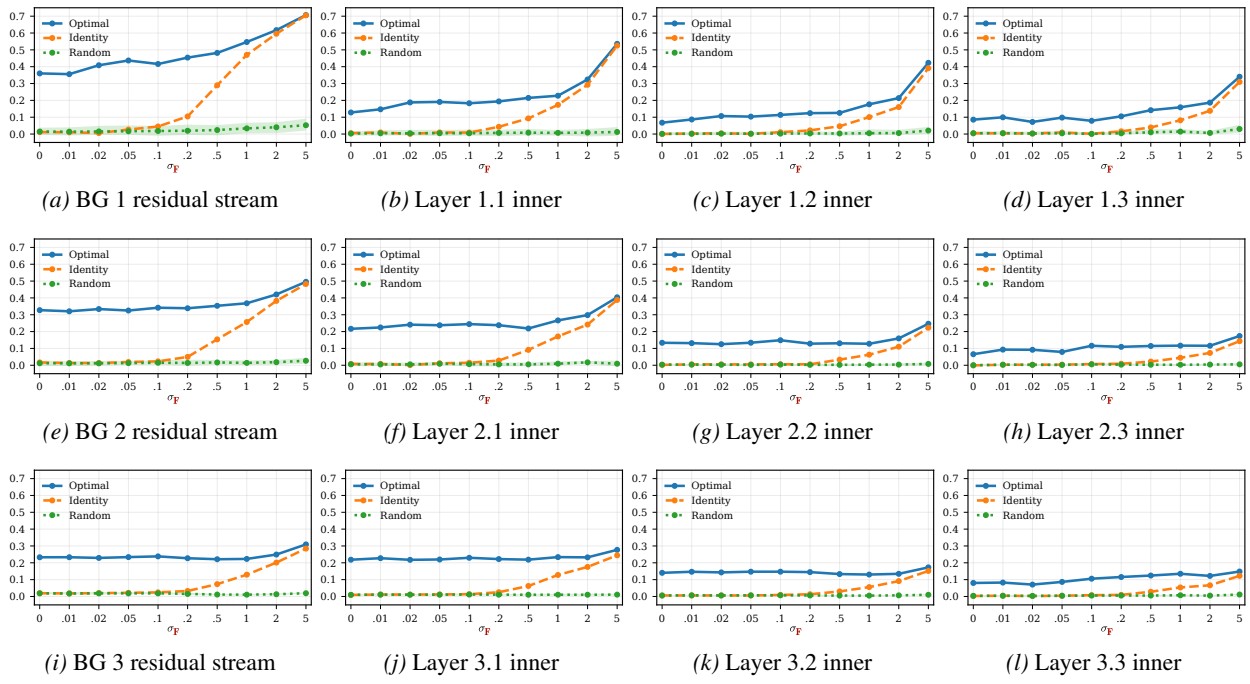

*Figure 9.* Activation matching objectives separated by layer/activation matching point ($W$-ResNet on CIFAR-10). In our $W$-ResNets, activation matching points lie within each of the three block groups' (BG) residual streams (used to estimate a global permutation for the residual stream that multiple layers write into), and between the two lin. layers of the three inner two-layer MLPs within each block group.

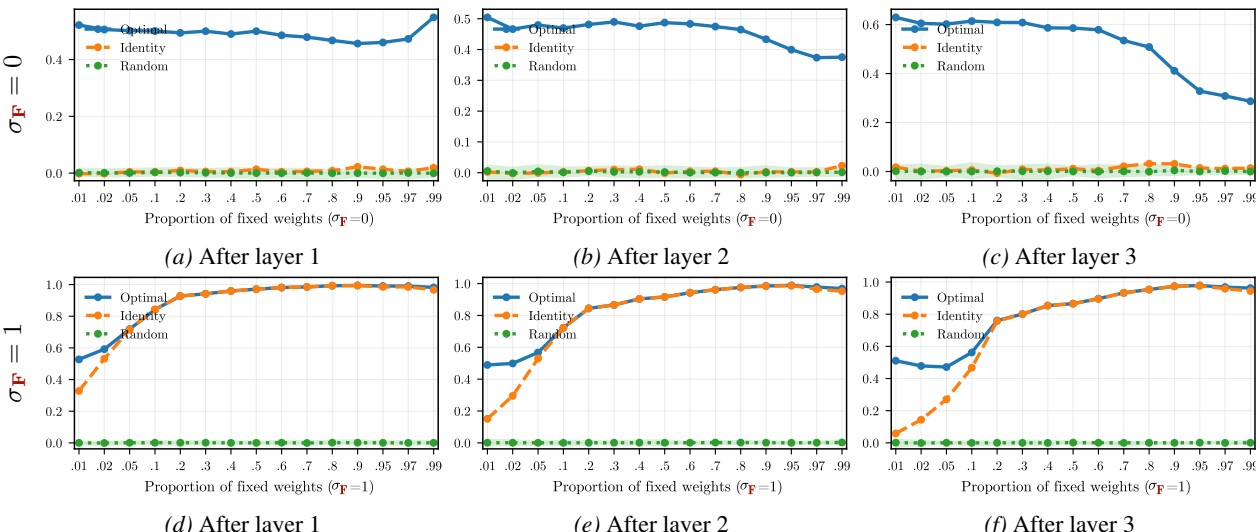

*Figure 10.* Activation matching objectives per layer ($W$-asymmetric MLPs on MNIST), sweeping over sparsity parameter (proportion of fixed weights) for both $\sigma_{\mathbf{F}} = 0$ (standard sparse training) and $\sigma_{\mathbf{F}} = 1$.

## G.2. Empirical Validation of Center Separation Rate

We also examine how the minimum distance between projected neuron centers changes with layer width $m$. For randomly sampled $\mathbf{F}$ we observe the expected scaling behavior from Thm. 4.6. To this end, we sweep the hidden dimension $m$ and intrinsic dimension $k \in \{2, 8, 32\}$ on the Gaussian mixture dataset. The constant $c_k$ in Fig. 11 is chosen per $k$ such that the rate is anchored at the respective $\gamma_{\text{out}}$ for $m = 16$. Thm. 4.6 predicts a rate of $\Theta(\sigma_{\mathbf{F}}\sqrt{k}m^{-2/k})$, where $\sigma_{\mathbf{F}}\sqrt{k}$ and further constants hidden by $\Theta$ are taken care of by the anchoring. This helps explain that neuron swaps are only plausible when projected centers are close, which is mainly the case for small intrinsic dimension or extremely wide layers. For realistic values of $k$, layers would need to be astronomically wide for such close centers to become likely.

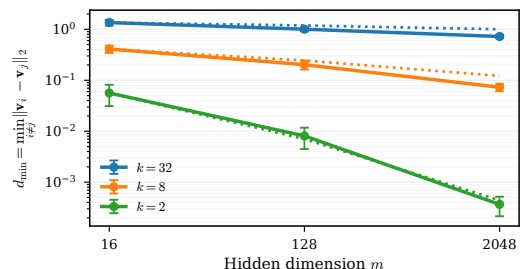

*Figure 11.* **Minimum pairwise projected center distance** $\gamma_{\text{out}}$ (solid, mean $\pm$ std) vs. **predicted asymptotic rate** $c_k m^{-2/k}$ (dotted).

## G.3. Neuron Swap Costs via Ridge Regression and Additional Pairwise Neuron Swap Costs

In § 6, we estimated the realization cost, and derived from that the neuron swap cost, via the Mahalanobis distance (10) in a PCA space with 90% explained variance. Appendix § E.1 suggests a different way to estimate realization costs, via a ridge regression objective Def. E.3. We compare the two methods in the following figures in more detail, separated by architecture, layer, and estimation method: We provide distribution box plots, comparing neuron swap costs for the four tested MLP-based architectures (MNIST) / zero fixed weights vs. center-dominated regime $\boldsymbol{W}$-asym. MLPs (Gaussian mixture data), and detailed pairwise neuron swap cost plots showing large/small/negative costs.

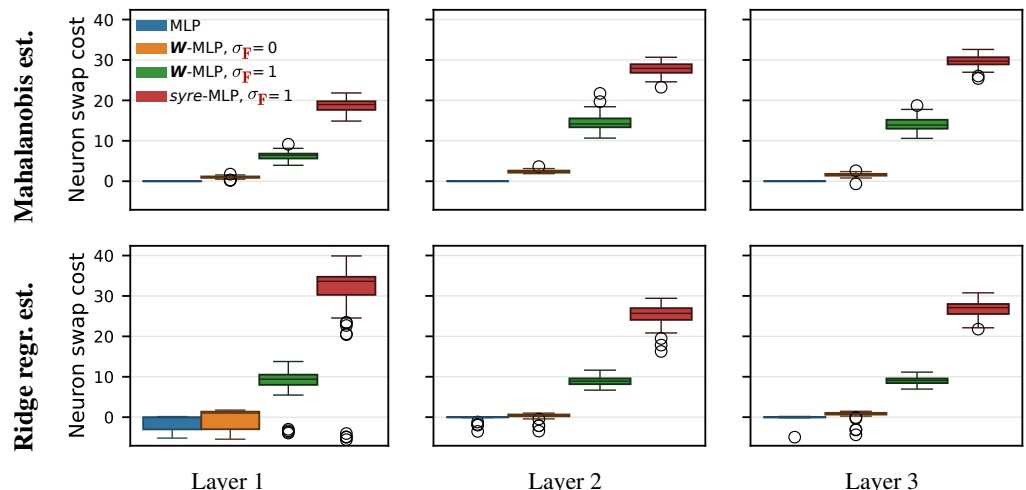

*Figure 12.* Distributions of neuron swap costs by architecture on **MNIST**. Plotted are signed sqrt. transformed $\Delta_{(ij)}^{\text{out}}$ for disjoint consecutive pairs $(i, j)$ of neurons, estimated via Mahalanobis distance (10) and ridge regression (Def. E.3, $\beta = 0.01$).

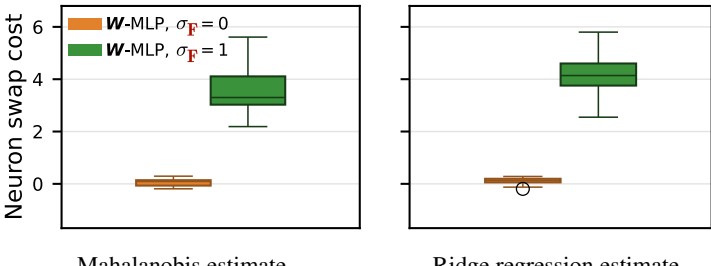

*Figure 13.* Neuron swap costs for $\boldsymbol{W}$-MLPs with $\sigma_{\mathbf{F}} \in \{0, 1\}$ on **Gaussian mixture data**. Plotted are signed sqrt.transformed $\Delta_{(ij)}^{\text{out}}$ for disj. consecutive pairs $(i, j)$ of neurons, estimated via Mahalanobis dist. (10) and ridge regression (Def. E.3, $\beta = 0.01$).

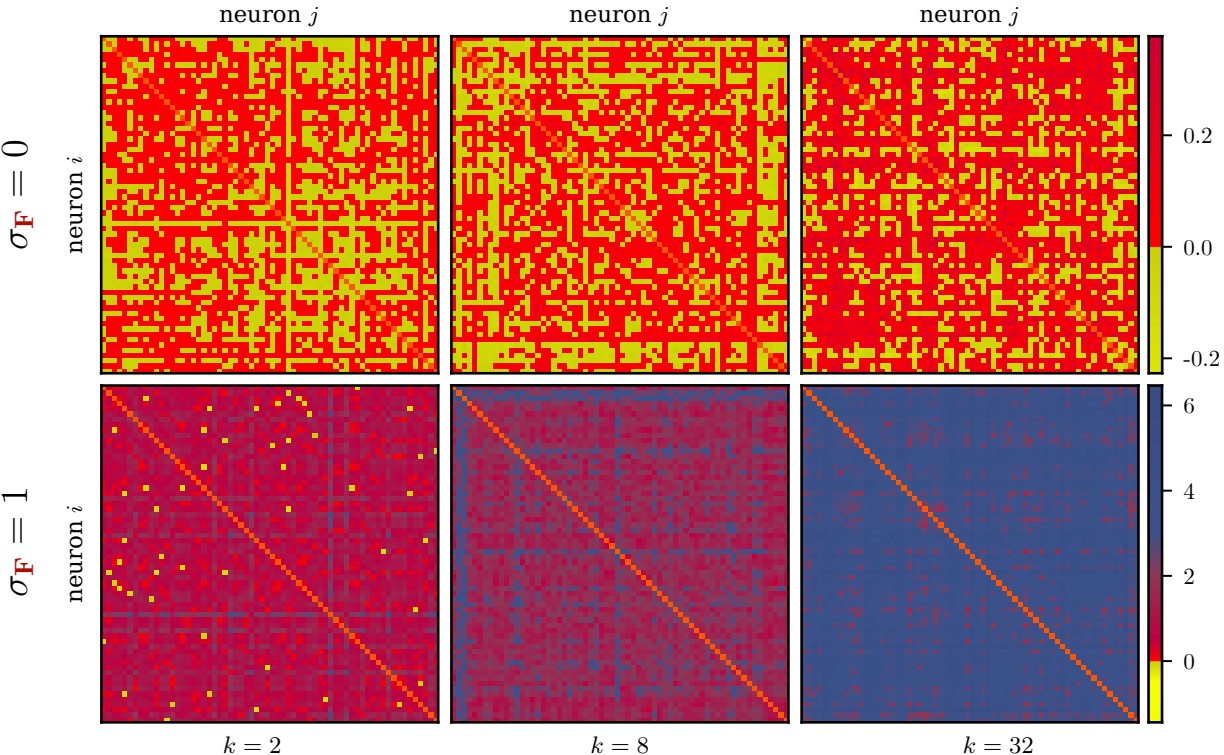

*Figure 14.* Neuron swap costs $\Delta_{(ij)}^{\mathrm{out}}$ (signed square-root transformed) estimated via **Mahalanobis distance** on **Gaussian mixture data**, for varying *intrinsic dimension k* and *fixed weight scale $\sigma_{\mathbf{F}}$* (same as Fig. 5).

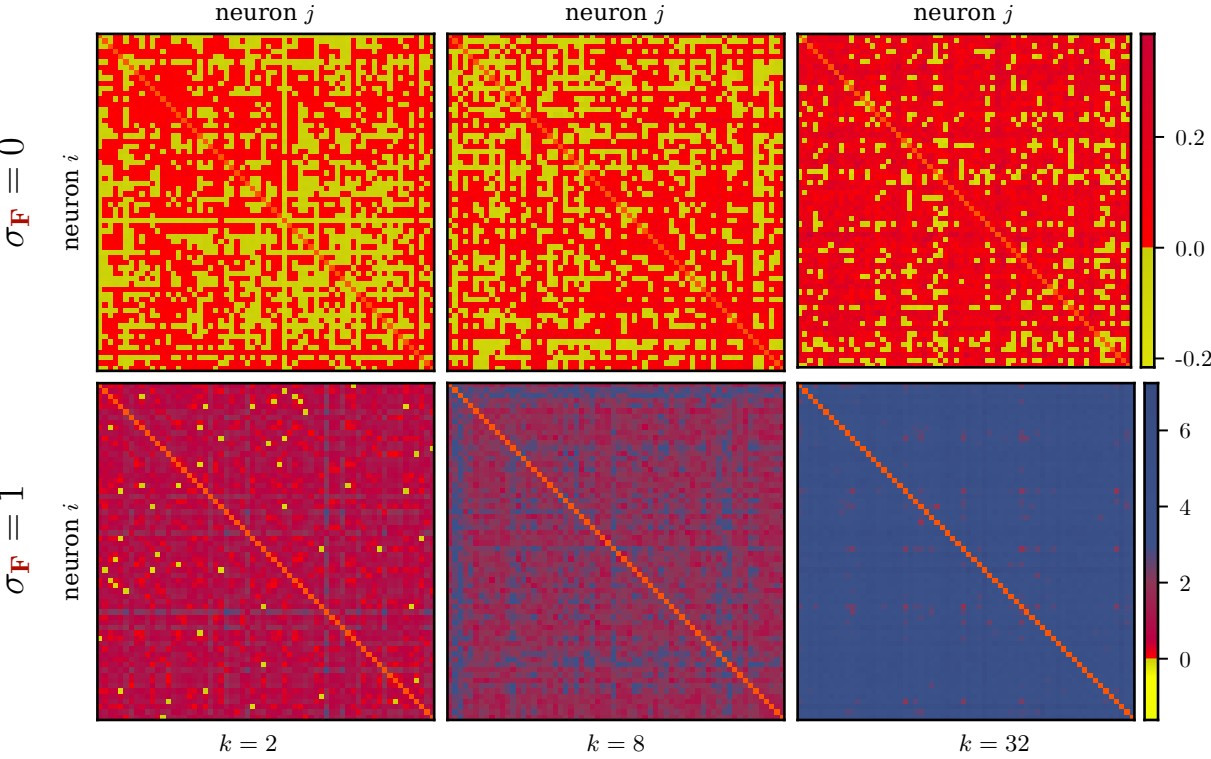

*Figure 15.* Neuron swap costs $\Delta_{(ij)}^{\mathrm{out}}$ (signed square-root transformed) estimated via **ridge regression**, $\beta = 0.01$ (see Def. E.3) on **Gaussian mixture data**, for varying *intrinsic dimension k* and *fixed weight scale $\sigma_{\mathbf{F}}$*.

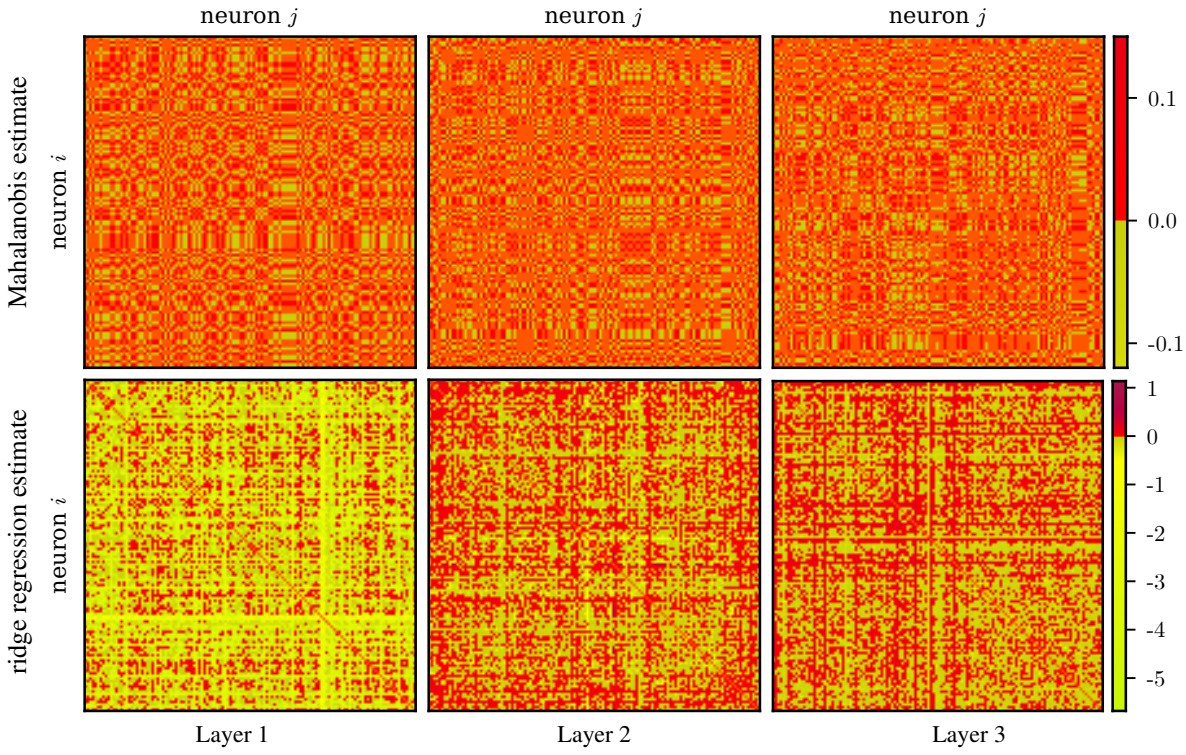

*Figure 16.* Neuron swap costs $\Delta^{\text{out}}_{(ij)}$ (signed square-root transformed) *per layer* for **standard MLP** trained on **MNIST**, estimated via *Mahalanobis distance* (10) vs. *ridge regression* (Def. E.3, $\beta = 0.01$). Layers $1 - 3$ have hidden dimension 512, shown are costs for first $128 \times 128$ neuron pairs.

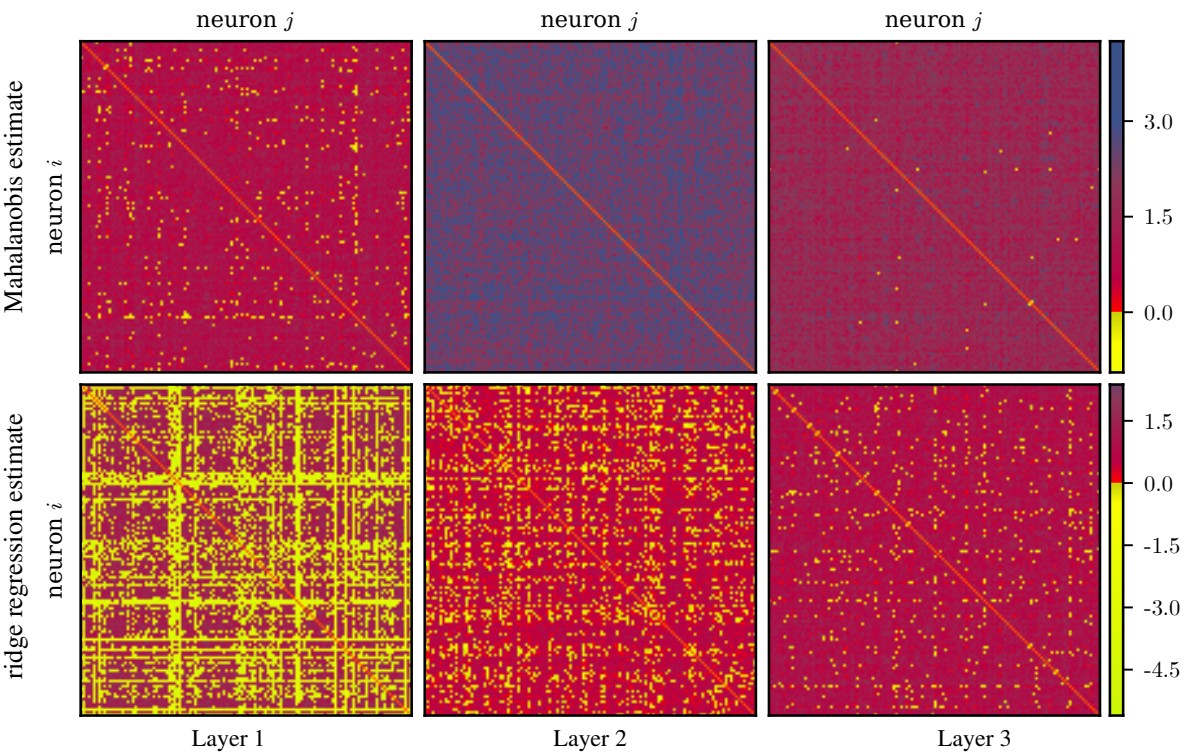

*Figure 17.* Neuron swap costs $\Delta^{\text{out}}_{(ij)}$ (signed square-root transformed) *per layer* for **W-asymmetric MLP** ($\boldsymbol{\sigma_{\mathbf{F}}} = \mathbf{0}$) trained on **MNIST**, estimated via *Mahalanobis distance* (10) and *ridge regression* (Def. E.3, $\beta = 0.01$). Layers $1 - 3$ have hidden dimension 512, shown are costs for first $128 \times 128$ neuron pairs.

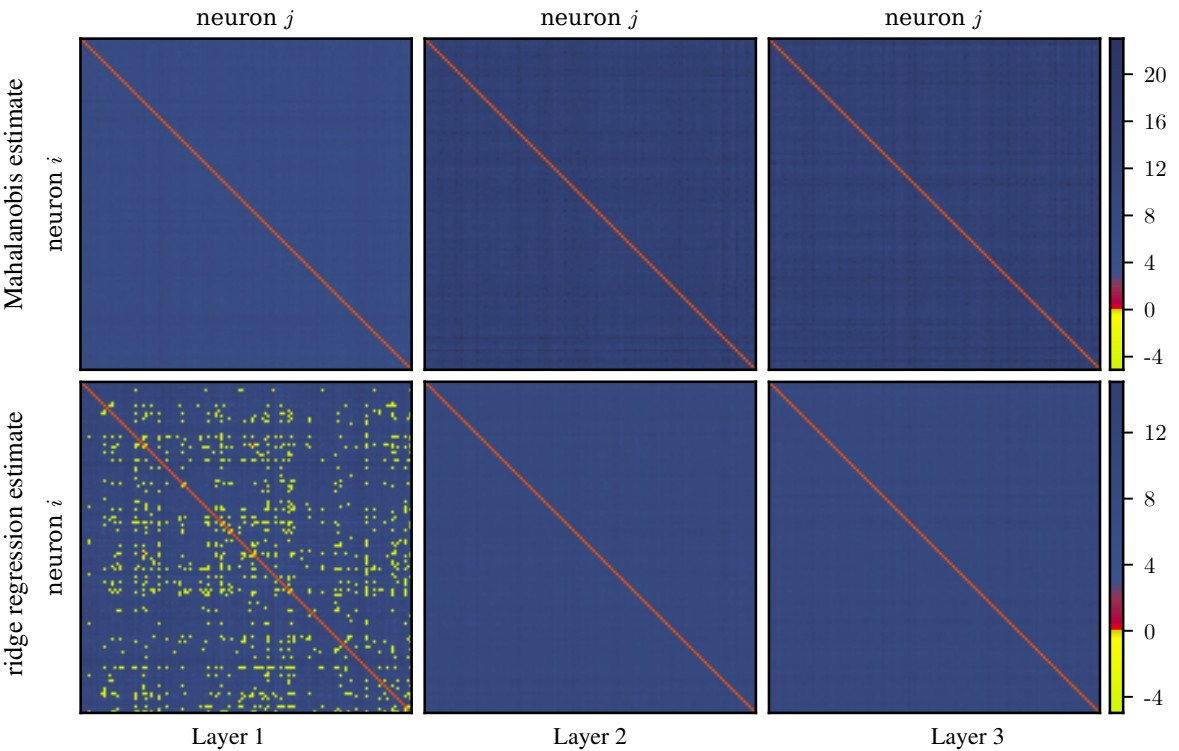

*Figure 18.* Neuron swap costs $\Delta^{\mathrm{out}}_{(ij)}$ (signed square-root transformed) *per layer* for **W-asymmetric MLP ($\sigma_{\mathbf{F}} = 1$)** trained on **MNIST**, estimated via *Mahalanobis distance* (10) and *ridge regression* (Def. E.3, $\beta = 0.01$). Layers $1-3$ have hidden dimension $512$, shown are costs for first $128 \times 128$ neuron pairs.

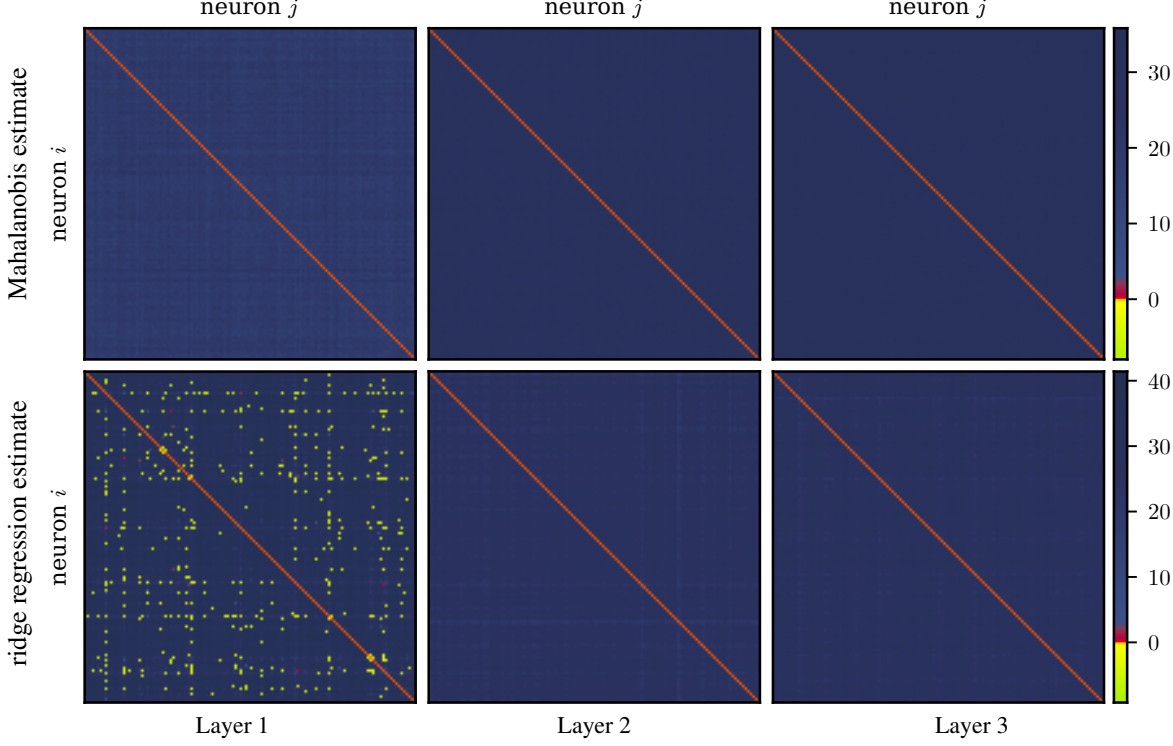

*Figure 19.* Neuron swap costs $\Delta^{\mathrm{out}}_{(ij)}$ (signed square-root transformed) *per layer* for **syre-MLP ($\sigma_{\mathbf{F}} = 1$)** trained on **MNIST**, estimated via *Mahalanobis distance* (10) and *ridge regression* (Def. E.3, $\beta = 0.01$). Layers $1-3$ have hidden dimension $512$, shown are costs for first $128 \times 128$ neuron pairs.

## G.4. Transformers

§ 6 studies the effect of neural parameter symmetries on optimization and linear mode connectivity for MLPs and ResNets. We additionally examine Vision Transformers (ViTs) (Dosovitskiy et al., 2021) on CIFAR-10 to test whether the same symmetry breaking mechanism extends to attention-based architectures. A transformer is composed of linear maps, including the query, key, value, and output projections in multi-head self-attention, the two linear layers in each feed-forward block, the patch embedding layer, and the final classification head. Transformers are an especially interesting extension because attention admits richer $GL(\mathbb{R}^n)$ symmetries beyond the permutation setting in § 6. We note that our goal is not to introduce a highly tuned state-of-the-art ViT, but to use a trainable transformer baseline whose training accuracy reached high performance. The goal is to investigate whether the same parameter-level intervention can reduce the interpolation barrier.

The ViT setting is more challenging for several reasons. First, transformer symmetries are not limited to hidden-unit or channel permutations. Multi-head attention introduces head-permutation symmetries, while the feed-forward blocks contain MLP-like hidden-unit symmetries. Second, LayerNorm, residual connections, positional embeddings, and the class token may interact with these symmetries in ways not captured by a simple linear-map masking scheme. Third, our implementation applies the $\boldsymbol{W}$-asymmetric parameterization directly to the transformer's linear maps. This is a natural extension of the original construction, but it may not be the optimal transformer-specific way to break symmetries. For example, breaking individual entries of $\boldsymbol{W}_Q, \boldsymbol{W}_K, \boldsymbol{W}_V, \boldsymbol{W}_O$ may not fully eliminate head-level equivalences unless the asymmetry is coordinated with the internal head structure. Similarly, asymmetrizing all linear maps may introduce additional optimization difficulty because fixed entries constrain the parameterization throughout training.

We use a CIFAR-10 ViT with patch size $4$, embedding dimension $384$, depth $8$, 6 attention heads, and MLP ratio $4$. Following the experimental setup in § 6, we apply a random 80%/10%/10% train/val/test split. For each linear map, we replace the ordinary trainable weight matrix $\boldsymbol{W}$ with a $\boldsymbol{W}$-asymmetric parameterization in the form of (2). The fixed entries are sampled once at initialization, and the number of fixed entries is $8$. This construction is a natural extension of the $\boldsymbol{W}$-asymmetric parameterization used for MLPs and ResNets, but it is not necessarily the optimal way to break transformer symmetries. For example, a more tailored approach might use head-aware masks or a design that explicitly respects the coupled structure of $\boldsymbol{W}_Q, \boldsymbol{W}_K, \boldsymbol{W}_V, \boldsymbol{W}_O$. We leave such transformer-specific variants for future work. We train the model using AdamW (Loshchilov & Hutter, 2019) with cosine learning-rate decay for 200 epochs using a batch size of 128. In our preliminary tuning, we found that larger fixed-weight scales can make transformer optimization more difficult. In particular, settings such as $\sigma_{\mathbf{F}} = 2$ were harder to train. When their scale is too large, they can dominate the trainable entries, alter the scale of attention logits and MLP activations, and reduce the flexibility of the model during optimization. Therefore, unlike in an idealized symmetry breaking setting, increasing $\sigma_{\mathbf{F}}$ should not be interpreted as a monotonic improvement in practice.

Fig. 20 reports the training accuracy along the linear interpolation path between independently trained transformer endpoints. We see that the standard transformer exhibits a pronounced interpolation barrier and the $\boldsymbol{W}$-Transformer with $\sigma_{\mathbf{F}} = 0$ closely matches this behavior, suggesting that masking alone, when the fixed entries are zero, is not sufficient to meaningfully improve unaligned LMC. In contrast, the $\boldsymbol{W}$-Transformer with $\sigma_{\mathbf{F}} = 1$ yields a noticeably flatter interpolation curve and substantially higher midpoint accuracy. However, the interpolation curve still remains well below the endpoint accuracy. The improvement at $\sigma_{\mathbf{F}} = 1$ suggests that nonzero $\boldsymbol{W}$-asymmetric fixed entries can reduce the unaligned interpolation barrier in transformers, qualitatively matching the mechanism observed for MLPs and ResNets in § 6. At the same time, the remaining barrier indicates that

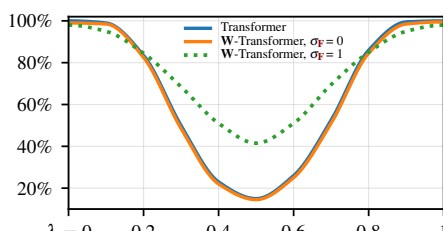

*Figure 20.* LMC of Transformer and $\boldsymbol{W}$-Transformer on CIFAR-10, measured by training accuracy along the interpolation path.

the current entrywise $\boldsymbol{W}$-asymmetric construction does not fully resolve the symmetries or optimization effects present in transformer architectures. This may be due to residual attention-head symmetries, richer $GL(\mathbb{R}^n)$ invariances, interactions with LayerNorm and residual connections, or the fact that our masks are inherited from the MLP and ResNet setting rather than designed specifically for attention. Thus, this provides evidence that the mechanism identified by our theory carries over qualitatively to transformers, while also motivating future work on transformer-specific symmetry breaking schemes.

## G.5. Subspace Coherence of MNIST Inputs and Hidden Representations

We empirically measure the subspace coherence $\nu(\mathcal{U})$ of model inputs, hidden representations and the final output for standard and asymmetric MLPs on MNIST data variants over training. Fig. 21 shows the results: in particular, variants

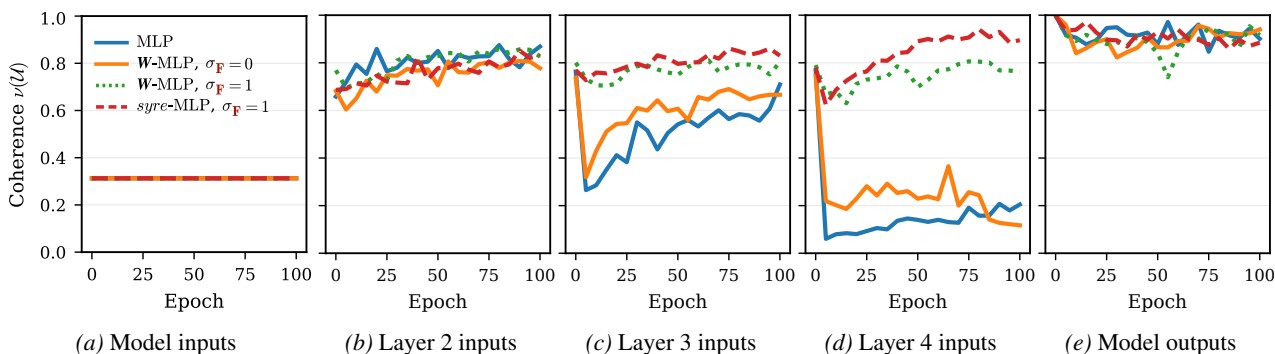

*Figure 21.* Subspace coherences $\nu(\mathcal{U})$ computed for model inputs, outputs, and hidden representations, on MNIST data, for standard and asymmetric MLP variants over training.

without effective symmetry breaking (MLP and $W$-asymmetric MLP with zero fixed weights) exhibit low subspace coherence, in particular in later layers, while $W$-asymmetric MLPs and *syre*-MLPs with large fixed weights both tend to exhibit higher subspace coherence values.

