# OpenReview forum: "Beyond Structural Symmetries: Linear Mode Connectivity via Neuron Identifiability"
_ICML.cc/2026/Conference — ICML 2026 regular_

### Official Review · Reviewer_bRWV · 2026-03-03

**Soundness:** 3
**Presentation:** 3
**Significance:** 3
**Originality:** 3
**Overall Recommendation:** 5
**Confidence:** 4

**Summary:**

This paper studies neuron identifiability in neural networks with parameter permutation symmetry and examines how such identifiability influences unaligned Linear Mode Connectivity (LMC). The authors introduce the notion of an Effective Function Class, characterizing each neuron’s realizable functions on a low-dimensional data subspace together with their associated realization costs. Based on this framework, they define Permutation Sensitivity and show that under a center-dominated regime, swapping neurons incurs a strictly positive cost, yielding a sufficient condition for functional identifiability.

The paper further links the separation of neuron centers to the intrinsic dimension of the data and establishes conditions under which linear interpolation between independently trained models admits a low-loss path without explicit alignment. Overall, the work provides a data-dependent and non-asymptotic perspective on symmetry breaking and its role in mode connectivity.

**Compliance With Llm Reviewing Policy:**

Affirmed.

**Final Justification:**

The rebuttal addressed all of my concerns, and I raise my score.

**Key Questions For Authors:**

**1. Robustness beyond strict linear subspace support.**
The theory assumes that the data distribution is supported on a fixed linear subspace and leverages global orthogonal projection structure in deriving spectral concentration and implementation cost expressions. If the data only approximately lie on a low-dimensional manifold, or follow a multi-subspace structure, do the core conclusions (e.g., center separation and permutation cost lower bound) still hold in some approximate sense? Could the framework be extended via local linearization or piecewise subspace modeling?

**2. Within-model vs. cross-model identifiability.**
Identifiability in this paper refers to functional changes induced by permuting hidden units within a fixed trained model. Does this notion imply any stability of functional roles across independently trained models with different random initializations? Or is identifiability purely an intra-model structural property?

**3. Symmetry strength and generalization.**
Under the assumption that the model successfully fits the training data, how do the authors view the relationship between effective symmetry strength and generalization performance? Could stronger symmetry reduction (higher identifiability) correlate with better or worse generalization?

**4. Symmetry structure and minima geometry.**
How do the authors interpret the relationship between network symmetry structure and the minima flatness in the loss landscape? Does the proposed framework suggest a refined view of how minima flatness affects generalization?

**Limitations:**

yes

**Strengths And Weaknesses:**

**Strengths**：

**1. Conceptually clear and structurally innovative framework.**
The introduction of the Effective Function Class provides a structured way to analyze neuron behavior under data constraints, offering a principled lens to study symmetry reduction.

**2. Clear distinction between structural symmetry and data-induced symmetry.**
The paper emphasizes that while parameter permutation groups exist at the architectural level, the effective symmetry group may be reduced by data geometry (e.g., intrinsic dimension and subspace structure), leading to functional identifiability. This distinction is conceptually meaningful and sharpens the discussion around symmetry in neural networks.

**3. A sufficient mechanism for unaligned LMC beyond empirical observation.**
By bounding interpolation-induced hidden representation deviations, the paper provides a theoretical mechanism explaining when unaligned LMC can succeed, rather than relying on empirical findings or infinite-width approximations.

**4. Non-asymptotic finite-width and finite-sample analysis.**
Several results explicitly quantify dependence on intrinsic dimension and other geometric parameters, instead of relying on NTK limits or infinite-width theory. This strengthens the practical relevance of the framework.

**Weaknesses**：

**1. Limited quantitative characterization of expressivity trade-offs.**
Although Section 4.5 discusses potential expressivity limitations under the center-dominated regime and relates them to hardness results in random feature models, the discussion remains largely structural. A more explicit quantitative analysis of the expressivity–identifiability trade-off would strengthen the theoretical impact.

**2. Lack of empirical validation for key geometric claims.**
Several theoretical relationships would benefit from experimental support. For example:
* Can synthetic datasets with controlled intrinsic dimension validate the predicted relationship between center separation scale and intrinsic dimension?
* Is there an observable correlation between the proposed identifiability measure and the success probability of unaligned LMC?

---

> ### Author Rebuttal · Authors · 2026-03-31
>
> We thank the reviewer for the constructive comments and for recognizing the relevance and originality of the problem.
>
> **Re W1 (expressivity-identifiability tradeoff):** We agree that this tradeoff can be stated more explicitly. The current paper already contains the main ingredients, but not in a single statement. In particular, Eq. 11 shows that under subspace support, the budget-$B$ slice of neuron $i$’s effective function class is an ellipsoid $\\{\\mathbf{a} \\,| \\,||\\mathbf{a}-\\mathbf{v}\_i||_{\\mathbf{S}\_i^{-1}}\le B\\}$ centered at $\mathbf{v}_i$. Under the concentration event of Eq. 19, this class lies inside a Euclidean ball of radius  $\sqrt{\mu\_d + \varepsilon}B$  around $\mathbf{v}_i$. Hence if $d\_{\min} > 2\sqrt{\mu\_d + \varepsilon}B$, the budgeted classes are already disjoint, and Thm. 4.5 yields $\min\_{i \ne j} \Delta\_{(ij)}^2(G) \gtrsim d\_{\min}^2 / \mu\_d - O(B^2)$. App. C.10 also gives a quantitative limitation: if trainable deviations remain small, the network stays close to its random-feature baseline and some target features cannot be approximated well. So the paper already implies a concrete tradeoff: shrinking the admissible deviation budget $B$ strengthens identifiability by separating the budgeted neuron classes, but if $B$ becomes too small, expressive power deteriorates. We do not claim to fully characterize the tradeoff curve, but this interpretation is also consistent with previous empirical observations on asymmetric networks ([1], fig. 6). Accordingly, we will soften the wording in the introduction and add a short corollary and discussion to make this clearer.
>
> **Re W2 (experimental validation):** Kindly refer to our response to reviewer jR4H (W2).
>
> **Re Q1 (simplifying assumptions):** We thank the reviewer for raising this point and refer to our response to Reviewer 3hQs (W2), where we explain why we focus on the subspace support model. We agree that exact subspace support is an idealization, as in practice, data / representations are better viewed as lying near manifolds or unions of subspaces. We use the subspace model as it is the simplest setting in which the realization cost becomes explicit and computable in closed form (which allows us to analyze relevant factors). Still, this abstraction is often meaningful in practice. Even nonlinear data is often well approximated by lower-dimensional subspaces. For instance, while the manifold dim. of ImageNet is ~20-50 [2], its PCA dim. is ~500 at 90% variance explained [3], still far below the ambient dim. ~150k. The subspace model is best read as a tractable approximation to the relevant data / representation geometry. We will add this to the revised paper.
>
> **Re Q2 (cross-model identifiability):** Thank you for this insightful question, this is an important distinction. We do not claim that independently trained models always converge to the exact same function. Instead, the premise of our theoretical framework (laid out in the beginning of §4), is that independently trained models often appear to realize essentially the same *set* of features in a layer, while initialization determines which neuron implements which feature. This is why post-hoc matching methods like Git Re-Basin [4] are effective in practice, and has been shown rigorously in the mean field infinite-width limit [5]. Formally, §4 analyzes permutations of a fixed realizable feature set within one model, which under our premise is a statement about cross-run stability as well. In that sense, this within-model notion also explains cross-model identifiability. This is also what our experiments test, since the predictions are evaluated on independently trained runs, and we will clarify this distinction more explicitly.
>
> **Re Q3, Q4 (generalization and geometry of minima):** Thank you for the thoughtful questions. Prior empirical work has reported a smaller generalization gap in some settings [1], so effective symmetry reduction may also affect generalization. Likewise, understanding how symmetry breaking changes minima geometry / flatness is an interesting direction, especially since this has long motivated studying parameter symmetries more broadly. As this work is motivated more by model merging / linear mode connectivity, we view both questions as beyond the current scope, but agree they are exciting future directions.
>
> [1] Lim et al. *The Empirical Impact of Neural Parameter Symmetries, or Lack Thereof.* NeurIPS 2024.
>
> [2] Ansuini et al. *Intrinsic dimension of data representations in deep neural networks.* NeurIPS 2019.
>
> [3] Bizeul. *ImageNet statistics and PCA.* https://zenodo.org/records/14589122
>
> [4] Ainsworth et al. *Git Re-Basin: Merging Models modulo Permutation Symmetries.* ICLR 2023.
>
> [5] Ferbach et al. *Proving Linear Mode Connectivity of Neural Networks via Optimal Transport.* AISTATS 2024.
>
> ---
>
> We thank the reviewer once again for their insightful comments, which helped improve the paper, and welcome any further suggestions.

---

> > ### Author Rebuttal · Reviewer_bRWV · 2026-04-02
> >
> > Thank you for your response. The authors have addressed all of my concerns, and I will raise my score.

---

### Official Review · Reviewer_4JSS · 2026-03-09

**Soundness:** 2
**Presentation:** 2
**Significance:** 3
**Originality:** 3
**Overall Recommendation:** 4
**Confidence:** 3

**Summary:**

In this work, the authors focus on  parameter symmetries, in particular permutation symmetries, that are inherent to most deep learning architectures. Due to these permutation symmetries (i.e., it is not defined a priori which neuron/filter will learn which feature, and different runs give a different result), it is often needed to align the weights of different networks before operations like model merging and similarity estimation, or to study the linear mode connectivity between models.

The authors develop a mathematical framework to characterize to which degree these permutation symmetries are broken in standard and newly defined, asymmetric architectures. The effective symmetry breaking is measured through neuron identifiablity: by the authors, this is defined as each neuron’s effective function class, i.e., the functions it can realize on the input support together with their realiza-
tion costs, measured by the minimum weight norm required to implement them.

**Compliance With Llm Reviewing Policy:**

Affirmed.

**Final Justification:**

I would like to thank the authors for their clarifications. I found the original paper quite hard to follow, but it seems the authors have a plan to make the storyline/presentation more clear. I will therefore raise my score from 3 to 4.

**Key Questions For Authors:**

Q1. Image I have a practical form of symmetry breaking (e.g., I add positional encodings). Could I always use the proposed framework to analyse neuron identifiability, or is this only possible if I can cast the symmetry breaking in the proposed shape (eq 2)?
Q2. Can you clarify in which ways your mathematical and experimental results lead to the conclusion: symmetry breaking is a functional and data-dependent phenomenon rather than purely architectural?
Q4. How would you change your manuscript to improve the presentation, if the audience is ‘ a general deep learning expert’?
Q3. Can you position your work with respect to other works that analyse/quantify symmetry breaking?

**Limitations:**

Negative societal impact: NA

limitations: limitations of used model for symmetry breaking (eq2) are unclear.

**Strengths And Weaknesses:**

Soundness

The proposed mathematical framework is seemingly sound and meticulous, in the sense that clearly a lot of effort has been put in formulating precise mathematical statements. I don’t have the background to confidently check all statements.

The analysis seems entirely based on the characteristics of the model used to obtain explicit symmetry breaking (eq 2). Although the authors describe how this model subsumes ‘several previous existing parameter breaking schemes’, it is not clear how universal this model is in representing symmetry breaking. It is, in the text, described as a ‘running example’ (see questions).

The authors also argue that one of the contributions/results of the paper are:
‘allows us to disentangle the effects of data-specific and architectural sym-
metries. ‘ (abstract)
‘In this work, we studied under which conditions parameter
symmetry breaking yields neuron identifiability [] and show that this is a functional and data-dependent phenomenon rather than purely architectural’ (conclusion).
Either I missed/did not grasp the argument, or this is not sufficienctly shown in the paper (see questions)




Presentation

The paper is hard to follow for a general deep learning expert. One reason is that the text contains a lot of terms that require more context for the reader to understand what the authors are aiming at.
E.g., in the abstract:
'we develop a theoretical framework of effective function classes defined by the neurons’ induced functions restricted to the representation subspace'
This seems an important part of the paper but it’s very vague/unclear for a general deep learning expert (perhaps not for a mathematician -it’s of course also a matter of the intended audience).

Moreover, the storyline/goal is not very clear. Is it to develop a general framework to quantify symmetry breaking through neuron identifiability? Is it to show that symmetry breaking is a functional and data-dependent phenomenon rather than a purely architectural one? Is it to develop models that don’t need to be aligned post-hoc to compare their representations/study their LMC? Perhaps it’s all these things at once, but it’s hard to follow, also because the (short) experiments cover different aspects and seemingly do not adress the stated research question:
'When does a given parameter symmetry breaking mechanism select a unique representative within each orbit across runs, and how do the data and representation distribution control its effectiveness?'
in a clear way.M ore guidance for the reader would improve the paper a lot.


Significance/originality

I believe the work might adress a relevant problem (the exact interplay between parameters, data, and representations in the context of parameter symmetries) and therefore adds to general understanding. The significance could be reduced due to the assumptions and used (not universal?) model of permutation symmetry breaking.

The work could be better situated with respect to other works that quantify symmetry breaking.

---

> ### Author Rebuttal · Authors · 2026-03-31
>
> We thank the reviewer for the thoughtful comments and for recognizing the relevance and importance of the problem we study.
>
> The main point we want to communicate more clearly is: the paper is not about proposing a new symmetry-breaking mechanism. Rather, it asks when a given architectural asymmetry actually produces stable neuron roles across runs on the data/representation support. This work argues that this is the property that matters for alignment-free comparison, merging, and LMC.
>
> **Re scope of the framework/universality of Eq. 2, Q1:** We thank the reviewer for raising this important point. Our intention is not to claim that every practical symmetry-breaking mechanism can be represented in the form of Eq. 2. Rather, Eq. 2 defines the main formal class for which our framework yields explicit finite-width identifiability guarantees. This class is broad enough to cover several recently studied interventions, e.g., $W$-asymmetric networks (Lim et al., 2024) and syre (Ziyin et al., 2025), and also includes linear residual connections and sparse networks, while the relevant quantities are explicit to support a tractable theory. However, it is not universal. Broader than Eq. 2 is the conceptual lens: compare neurons through the functions they can realize on the data support, and through the cost of realizing them.
>
> **Re support for the functional and data-dependent claim, Q2:** We thank the reviewer for this valuable comment. By functional and data-dependent we mean that within our framework, identifiability depends jointly on the architectural asymmetry and the data/representation support, not on architecture alone. An architectural asymmetry only helps for unaligned merging if it is visible on the support explored during training. If two neurons remain functionally interchangeable on that support, then they remain effectively exchangeable across runs, even if parameter permutations are no longer symmetries in the raw parameter space. §4 formalizes when the asymmetric component is strong enough to select stable neuron roles across runs, and §5 explains why this stable role assignment is the ingredient that makes unaligned merging/LMC plausible. Empirically, Figs. 3-5 track this mechanism: (1) Fig. 3 shows there are settings ($\kappa$ small) where we break structural symmetry but the neurons still do not align across models as it still leaves a substantial unaligned barrier, (2) Fig. 4 shows as $\kappa$ increases, the identity activation matching score moves from near-random to near-optimal, and (3) Fig. 5 shows neuron classes become more distinct when the architecture effectively breaks symmetries.
>
> **Re positioning relative to prior work, Q3:** We agree that the positioning should be sharpened.  Relative to most literature on parameter symmetries, our aim is to formally analyze symmetry effects that go beyond pure architecture. Relative to alignment/LMC papers, we study when post-hoc alignment becomes unnecessary. Relative to recent architectural symmetry-breaking papers, we do not propose a new intervention; instead, we provide a neuron-level analysis, via effective function classes and realization costs, for when such interventions induce identifiability. Specifically, compared with prior work that mainly offers empirical design choices or analyzes general loss symmetries, our paper gives a mechanistic, data-dependent finite-width account of effective symmetry breaking for model merging/LMC.
>
> **Re presentation for storyline and contribution clarity, Q4:** Thank you for this important comment. In the revision, we will reorganize the presentation around the paper’s central question: when does an architectural asymmetry break parameter symmetry strongly enough that corresponding neurons become identifiable across runs, reducing the need for post-hoc alignment? Specifically, we will:
>
> 1. Clarify the paper’s central question, tool, and consequence early in the introduction:
>
>     **Goal:** Characterize when explicit architectural asymmetry makes neurons identifiable across runs.
>
>     **Tool:** A new framework for modeling neuron realization cost and swapping.
>
>     **Consequence:** When that happens, one can merge representations and obtain unaligned LMC.
>
> 2. Use plainer language throughout the paper, e.g., instead of: “effective function classes defined by the neurons’ induced functions restricted to the representation subspace”, we will say: “the set of functions a neuron can realize on its input support, and the norm cost of realizing them”.
> 3. Add intuition and big picture to the theoretical results, i.e., a roadmap: architecture asymmetry → support-restricted neuron function classes → identifiability → unaligned merging and LMC.
> 4. Connect the experiments more clearly with the theory (kindly refer to our response to Reviewer jR4H (W2)).
>
> ---
>
> We thank the reviewer once again for their insightful comments, which helped us improve our paper, and hope to further discuss any ongoing suggestions.

---

> > ### Author Rebuttal · Reviewer_4JSS · 2026-04-03
> >
> > I would like to thank the authors for their clarifications. I found the original paper quite hard to follow, but it seems the authors have a plan to make the storyline/presentation more clear. I will therefore raise my score to 4.

---

### Official Review · Reviewer_jR4H · 2026-03-12

**Soundness:** 3
**Presentation:** 2
**Significance:** 3
**Originality:** 3
**Overall Recommendation:** 4
**Confidence:** 3

**Summary:**

This paper develops a theoretical framework to characterize effective symmetry breaking through neuron identifiability. At a high level, neural networks can be functionally (approximately) equivalent even if they differ in parameter space – the paper studies this by looking at the effective function classes for each neuron’s induced functions. Finally, some experimental results illustrate how effective symmetry breaking allows for representation merging.

**Compliance With Llm Reviewing Policy:**

Affirmed.

**Final Justification:**

The authors have proposed coherent plans to address most of my concerns with respect to the writing of this paper, and the additional experiments have reinforced my belief in the general validity of the theoretical results. I feel that there may still be some problems with the clarity and general density of the content of the paper, but overall, I am quite positive about the work. Based on this, I am maintaining my score.

**Key Questions For Authors:**

See Strengths and Weaknesses

**Limitations:**

Yes.

**Strengths And Weaknesses:**

*Strengths*

- The paper explores a highly relevant and timely subject – neural network parameter space symmetries are interesting, with potential for large practical impact thanks to the increasing relevance and effectiveness of model merging approaches.

- In its writing, the paper carries some ideas that bring up some interesting questions.
For instance, it is interesting to consider How one can account for “pseudo-symmetries” when studying linear mode connectivity and model merging, based on the following context:
> ... even when parameter symmetries are broken, different parameters may still implement approximately the same function...

- The assumptions made as part of the theoretical analysis seem very reasonable, relative to many theory papers where assumptions can seem overly constraining.

- The paper attempts to give intuitions for theoretical results where possible.

*Weaknesses*

- A major weakness of this work is the seeming disconnect between the theoretical framework and results, and how it connects to practice. By this, I do not necessarily mean that the setting is impractical or unrealistic; rather I struggle to see what the concrete takeaways from these theoretical results should be, and how exactly the theoretical results translate into the points mentioned in the subsection on Contributions.

- For similar reasons, this causes an apparent disconnect between the theoretical results and experiments as well, where it is hard to see why and how these experiments tell us much about how effective symmetry breaking is related to the data, for instance.

- Finally, the interpretation of experimental results could be made clearer. For instance: (i) What do the points and bars in Figure 5 indicate? Does the x-axis hold any significance? (ii) Are the “optimal” permutations in Figure 4 truly optimal, or are they permutations corresponding to activation matching?

---

> ### Author Rebuttal · Authors · 2026-03-31
>
> We thank the reviewer for the valuable comments and for recognizing the subject as relevant and timely.
>
> **Re W1 (key takeaways):** The key takeaway we will make more explicit in the revision is that breaking a parameter symmetry in raw parameter space is not, by itself, enough to make independently trained networks behave as aligned for merging or LMC. Even if “structural” weight symmetries are broken (Zhao et. al., 2025; Lim et al, 2024; Ziyin et al., 2025), the resulting networks do not necessarily behave as if “aligned” in terms of merging and LMC. What matters is whether the neurons can replace each other’s roles.
>
> This leads to three takeaways:
>
> 1. Architectural asymmetry only matters insofar as it is visible on the data support; an asymmetry that exists in parameter space but is not noticeable on that support will not reliably identify neurons across runs (Prop. 4.2, Eq. 11, Thm. 4.3 and Thm. 4.7).
> 2. If the fixed asymmetric component separates neuron roles strongly enough relative to learned deviations, then the same roles are selected across runs; if not, neurons remain effectively exchangeable (Prop. 4.4, Thm. 4.5, Thm. 4.6).
> 3. This cross-run stability makes alignment-free representation merging and unaligned LMC plausible (Thm. 5.1 and Cor. 5.2).
>
> **Re W2 (connection between theory and experiments):** We agree with the reviewer that the connection between the theory and experiments can be strengthened. The experiments in the submission focus on the effect of the center dominance (i.e., the scale of fixed weights F) and show that higher center dominance, in the form of higher scale parameter κ, yields stronger effective symmetry breaking (Figs. 4-5). This relates to the theory of §4.3, §4.4, and we will explain this connection more clearly in the revision.
>
> To further make the data dependence more explicit, we will add two synthetic experiments that isolate the effect of data-dependent quantities (subspace coherence, intrinsic dim.) that according to §4 control the effectiveness of symmetry breaking. We share the figures belonging to the following experiments in this PDF: https://anonymous.4open.science/api/repo/08DB/file/rebuttal_figures.pdf
>
> 1. In contrast to our existing finding that symmetry breaking is ineffective for F=0 (non-center-dominated), we now show that on highly coherent data, it can become effective even in this setting. Thm. 4.3/Eq. 11 predict that in this case, the subspace coherence ν(U) controls how easily neurons can take on each other's roles. We use an intrinsically two-dimensional dataset, where a, b～N(0,1) are embedded into a 20-dim. ambient space by copying each value a, b to t ≤ 10 positions, filling up the rest with zeros. Then ν(U)=1/t can be varied from minimal (t=10) to maximal (t=1) by varying t. We train a 1-hidden layer W-MLP (F=0, D～Bernoulli(0.15)) to fit the target ReLU(a+b). Our results (see PDF, §1) show LMC for high coherence, but high midpoint barrier for low coherence, and the same trend for a variant where the data is randomly rotated inside the first 2t coordinates (”random frame”).
> 2. We perform new synthetic experiments to clarify the influence of a layer’s intrinsic input dim. on the distances of projected neuron centers $\mathbf{v}\_i$ (Thm. 4.5) and neuron swapping cost. We train 1-hidden-layer W-MLPs on a Gaussian mixture classification task, where we vary the intrinsic dimension 2 ≤ k ≤ 32 and the number of neurons of the first layer 16 ≤ m ≤ 2048. We compute realization cost sensitivities (i.e., neuron swap costs) of the first layer’s neurons via Eqs. 11 and 15. The plots (see PDF, Fig. 2) show negative or near-zero swap costs both if κ=0 or the intrinsic dimension is low, as opposed to high positive swap costs for higher intrinsic dimension. This shows the dependence of effective symmetry breaking on intrinsic dim. and the scale of F.
>
>     As an additional check, we also examine how the minimum distance between projected neuron centers changes with layer width m. For randomly sampled F we observe the expected scaling behavior $\Theta(m^{-2/k})$ from Thm. 4.5 (see PDF, Fig. 3). This helps explain the swap-cost results: neuron swaps are only plausible when projected centers are close, which is mainly the case for small intrinsic dimension or extremely wide layers. For realistic values of k, layers would need to be astronomically wide for such close centers to become likely.
>
>
> We will include these experiments and a sharpened discussion in a revision.
>
> **Re W3 (presentation of experiments):** We thank the reviewer for the suggestions. We will clarify that the "optimal" in (Fig. 4) means optimal w.r.t. the activation matching objective. The plot is a box plot, and the points are outliers according to the standard IQR method. The x-axis refers to different architectures (see legend).
>
> ---
> We thank the reviewer once again for their insightful comments, which helped us improve our paper, and hope to further discuss any ongoing suggestions.

---

> > ### Author Rebuttal · Reviewer_jR4H · 2026-04-01
> >
> > I thank the authors for their responses and additional experiments, which have helped clear up some of my questions and appreciate their plans to make the draft clearer.

---

### Official Review · Reviewer_3hQs · 2026-03-13

**Soundness:** 3
**Presentation:** 3
**Significance:** 3
**Originality:** 3
**Overall Recommendation:** 5
**Confidence:** 3

**Summary:**

This paper studies neuron identifiability through the lens of the functions each neuron can realize on the data subspace and their realization costs, showing theoretically that effective symmetry breaking depends jointly on architecture and data geometry, and demonstrating empirically on MLPs and ResNets that identifiable neurons enable unaligned linear mode connectivity.

**Compliance With Llm Reviewing Policy:**

Affirmed.

**Key Questions For Authors:**

1. Theorem 5.1 introduces $t$, but the rest of the theorem does not use $t$.

2. If the proposed theory could be validated on larger-scale models, the contribution of the paper would be even stronger.

**Limitations:**

yes

**Strengths And Weaknesses:**

**Strengths**

1. The paper introduces neuron identifiability as a formal notion capturing whether neurons consistently take on the same functional roles across independent training runs, and uses it as the central concept for understanding symmetry breaking and unaligned LMC.

2. The paper further defines a realization cost, namely the minimum parameter norm required for a neuron to implement a target function.

3. The authors connect neuron identifiability to unaligned LMC by showing that when neurons have stable identities, hidden representations from independently trained models can be merged more directly, yielding smaller interpolation nonlinearity and lower loss barriers along the linear path.

**Weaknesses**

1. Although the experiments align well with the theory, the empirical validation is mainly restricted to relatively modest settings such as MLPs on MNIST and ResNets on CIFAR-10.  Are similar phenomena also observed in Transformers or large language models (LLMs)?

2. The theoretical analysis relies on several strong simplifying assumptions, such as the input distribution being exactly supported on a low-dimensional linear subspace, parts of the analysis depending on Gaussian assumptions, and many results being derived at the single-layer or local layer level.

---

> ### Author Rebuttal · Authors · 2026-03-31
>
> We thank the reviewer for their valuable comments and for recognizing the soundness of our neuron identifiability framework.
>
> **Re W1, Q2 (empirical scope):** We thank the reviewer for the insightful comment. We agree that transformers are an especially interesting extension because attention admits richer $\mathrm{GL}_n(\mathbb{R})$ symmetries beyond the permutation setting we mainly look at in this paper. We would also like to note that the intervention in Eq. 2 is not inherently permutation specific. In a linear setting, one can show that it also breaks orthogonal symmetries (i.e., restricts the relevant continuous group action). We did not include this result in the current draft due to space constraints, but we would be happy to add it in the revision. Empirically, for transformer merging, the most relevant subgroup appears to be $\mathrm{O}(n) \subset \mathrm{GL}_n(\mathbb{R})$, since scale/gauge are mainly pinned by initialization and optimization [1].
>
> Following the reviewer’s suggestion, we also tested asymmetric ViTs on CIFAR-10, which qualitatively mirror the trends shown in §6: As in Fig. 3, increasing the fixed weight scale substantially reduces the unaligned interpolation barrier, and in the regime without offsets, the identity permutation is not special and unaligned interpolation remains poor. This provides initial evidence that the mechanism identified by our theory is not specific to MLPs and ResNets, but carries over qualitatively to transformers as well. Activation matching for the ViT to replicate the results for Fig. 4 is still ongoing. We plan to present this ViT as a third model next to the MLP and ResNet in the revision.
>
> **Re W2 (simplifying assumptions):** We thank the reviewer for bringing this point up. We agree that the exact subspace support model is idealized, and we would like to explain our thought process for choosing it more explicitly. A natural starting point is to work with a relaxed ridge-like realization cost objective of the form $||g||\_{\mathcal{H}_i, \beta}^2 =  \inf_w  \\{ \beta^{-1} \mathbb{E}\_{x \sim \mathbb{P}}[(g(x)-h_i(w;x))^2] + ||w||_2^2 \\}$, in which this objective asks for the cheapest way to approximately realize the target function $g$ using neuron $i$. This formulation has the advantage that it does not require exact realizability for finite realization cost. The difficulty is that, for general nonlinear activations, this objective is nonconvex and not available in closed form. This was one of the main reasons we did not choose to make it a central object. Instead, we study its ridgeless limit by taking the limit $\beta \to 0$ which recovers our realization cost: $||g||\_{\\mathcal{H}_i}^2 = \\lim\_{\beta \to 0}||g||\_{\mathcal{H}_i,\beta}^2 = \inf\_{w} \\{ ||w||_2^2 \\:|\\: \mathbb{E}\_{x \sim \mathbb{P}}[(g(x)-h_i(w;x))^2] = 0 \\}$. This looks like an RKHS norm in spirit but differs in that it does not allow arbitrary linear combinations of $h_i(w, \cdot)$. Note that $||g||\_{\mathcal{H}_i}$ only depends on the support of $\mathbb{P}$ and not on the exact distribution, and under our assumptions, this ridgeless limit can be computed in closed form in the subspace support model (Eq. 11).
>
> This motivates the subspace-support idealization. One can think of it as replacing the true data distribution by an approximation whose support lies in a lower-dimensional subspace, for instance after a PCA truncation or frequency cutoff. We then compute the same closed-form realization cost on that approximating support. This is no longer exact for the original distribution $\mathbb{P}$, but gives a computable approximation to the observable realizability structure.
>
> We also considered a more local alternative based on linearizing around a trained model and defining realization costs through Generalized Gauss–Newton objects (see App. D.2). Our conclusion was that this substantially increases the technical overhead while adding limited insight for the claims made here. For this reason, we keep the framework at the simplest level that still captures the main phenomena we observe empirically. Finally, the formal statements are layerwise because permutation symmetries act locally at the neuron/channel level, whereas the empirical study in §6 is explicitly multi-layer.
>
> We will include this discussion in the revised paper to clarify the motivation for the subspace-support model and make the scope of the approximation more explicit.
>
> **Re Q1:** Thanks for spotting that $t$ in Thm. 5.1 is unused, we will remove it in a revision.
>
> [1] Theus et al. *Generalized Linear Mode Connectivity for Transformers.* NeurIPS 2025.
>
> ---
> We thank the reviewer once again for their insightful comments, which helped us improve our paper, and hope to further discuss any ongoing suggestions.

---

> > ### Author Rebuttal · Reviewer_3hQs · 2026-04-04
> >
> > The rebuttals have resolved my questions and I will keep the positive score.

---

### Decision · Program_Chairs · 2026-04-30

**Decision:**

Accept (regular)

**Comment:**

The reviewers are all in favor of acceptance, citing the elegant analysis of symmetries in neural networks. The main recurring complaint is the limited practical implication of this result, but it is fine for a theoretical paper.

PS: The realization cost introduced in this paper appears to be equivalent to the already existing "representation cost" (see for example https://proceedings.neurips.cc/paper/2021/hash/e22cb9d6bbb4c290a94e4fff4d68a831-Abstract.html and some of the papers cited inside) which has been studied for multiple neural architectures. If these two concepts are indeed equivalent, then the paper should be modified to use the already existing terminology and cite the existing literature.